# DECODESHARE: Tracing the Shared Subspace of LLM Decode-Time Decisions

**Zishan Shao** [1]   **Lixun Zhang** [1 2]   **Kangning Cui** [3]   **Yixiao Wang** [2]   **Ting Jiang** [1]   **Hancheng Ye** [1]   **Qinsi Wang** [1]
**Zhixu Du** [1]   **Yuzhe Fu** [1]   **Fan Yang** [3]   **Danyang Zhuo** [2]   **Yiran Chen** [1]   **Hai Helen Li** [1]

## Abstract

Large language models (LLMs) handle many tasks with one set of parameters, but under KV-cached inference it is unclear what task-general structure, if any, is used at *decode time* rather than during *prefill*. We propose DECODESHARE, a protocol that identifies a low-dimensional subspace consistently shared across tasks in decode-time hidden states, and then tests its causal role by removing that subspace only during decoding. In our experiments, disturbing the discovered shared subspace degrades decision performance far more than disturbing either a prefill-derived or random subspace under the same intervention budget. We further show this decode-shared subspace has practical consequences for activation steering: common steering directions can overlap the task-general decode channel. Projecting out this shared subspace directly separates the functional roles of the two components, while evaluating steering vectors at decode-time yields more reliable signal for downstream deployment than prefill-based proxies. Despite its compactness, the shared subspace can serve as a high-leverage causal channel at decode time. Code is available at: https://github.com/Zishan-Shao/decodeshare.git.

## 1. Introduction

Large language models (LLMs) achieve strong performance across a wide range of tasks using a single set of parameters. This multi-task generality motivates the idea of computation reuse: diverse behaviors may be mediated by a common set of internal features rather than entirely task-specific ones.

We can view this reuse as inducing a shared subspace in the model's decision-time representations–i.e., a set of directions that are consistently engaged across tasks and prompt variants. If so, it would provide a principled target for analysis and intervention that transfers across tasks and prompt variants.

Motivated by this, a growing line of work studies and steers LLM behavior by intervening on internal *activations* using low-dimensional directions or subspaces (Zou et al., 2023; Turner et al., 2023; Rimsky et al., 2024). In practice, however, activation-level interventions are often brittle: effects that appear strong under one prompt template, task setup, or implementation detail can weaken, flip, or disappear under small perturbations (Tan et al., 2024; Deng et al., 2025). This brittleness limits reproducibility and makes negative results hard to interpret.

We argue that brittleness is amplified because interventions are often not aligned with the *states* that drive next-token decisions under modern inference. *First, we rarely measure or protect task-general structure at decision time.* Most activation-steering methods estimate a small set of directions for a specific behavior or attribute under a fixed prompt distribution (Turner et al., 2023; Rimsky et al., 2024; Zou et al., 2023; Konen et al., 2024; Arditi et al., 2024; Todd et al., 2024). These methods typically do not control interactions between steering directions and task-general computation directions (Merullo et al., 2024; Kaushik et al., 2025). If next-token accuracy relies on a shared decision-time channel, then perturbing it can cause large failures and template sensitivity, consistent with observed brittleness under prompt or template shifts (Tan et al., 2024; Deng et al., 2025).

*Second, there is a mismatch between where directions are estimated and where decisions are made.* In common activation-steering pipelines, steering directions are typically estimated from *prefill* activations computed on the full prompt (Turner et al., 2023; Rimsky et al., 2024; Zou et al., 2023; Konen et al., 2024; Arditi et al., 2024; Todd et al., 2024). Under KV-cached decoding, however, each next-token decision is computed from a single-token *decode* pass. If decode-time hidden states differ from prefill activations, then prefill-estimated directions may not align

[1]Department of Electrical and Computer Engineering, Duke University, Durham, NC, USA [2]Department of Computer Science, Duke University, Durham, NC, USA [3]Department of Computer Science, Wake Forest University, Winston-Salem, NC, USA. Correspondence to: Zishan Shao <zishan.shao@duke.edu>.

*Proceedings of the 43rd International Conference on Machine Learning*, Seoul, South Korea. PMLR 306, 2026. Copyright 2026 by the author(s).

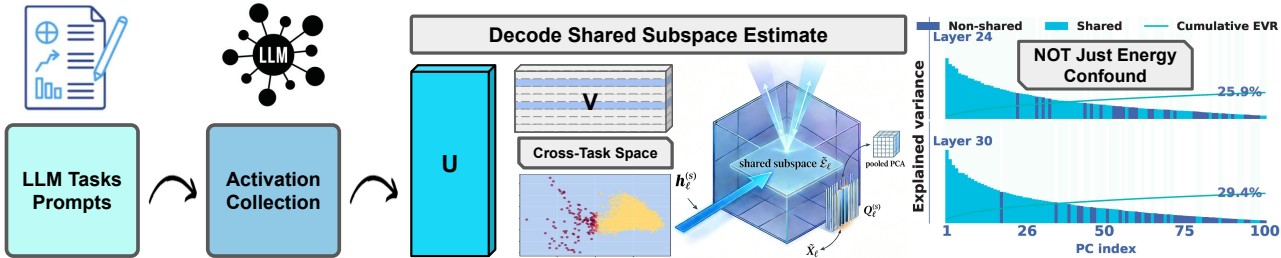

*Figure 1.* **DecodeShare Pipeline.** (1) We collect KV-cached *decode-time hidden states* from prompts across multiple tasks and pool them into a matrix. (2) A pooled PCA yields a set of orthonormal directions (columns of V matrix); we then select a subset that is consistently shared across tasks ($Q_\ell^{(s)}$), rather than simply taking the top-variance components. (3) To test causality, we intervene *only during decoding* by removing the selected component from the hidden state and compare against dimension- and *energy*-matched controls to rule out a simple high-variance/energy confound.

with the states that actually produce logits. This mismatch matters for steering selection: a vector favored by prefill-based validation may not perform best under held-out KV-cached decoding. Since steering vectors can also overlap task-general decode structure, projection is best viewed as an interference diagnostic rather than a universal repair.

We introduce DECODESHARE, a protocol that estimates candidate shared subspaces from decode-time hidden states and evaluates their relevance to decoding decisions via *decode-only* projection removal. Figure 1 previews our DECODESHARE pipeline: we collect decode-time hidden states across tasks, identify a shared subspace, and test its causal role with decode-only ablations. Accordingly, we analyze the decode-time hidden state under KV-cached inference as the representation that drives next-token logits. (We introduce formal notations and definitions in Sec 2.1.)

> Hypothesis: shared decode-time decision channel
>
> Under KV-cached decoding, next-token decisions causally depend on a *small* subspace $S$ of the decode-time hidden state that is shared across tasks.

**Key findings and implications.** Across models and benchmarks, removing the identified shared subspace causes much larger drops in decision accuracy than matched controls, supporting a high-leverage causal channel at decision time. This effect is largely *decode-specific*: under matched budgets, subspaces estimated from prefill often fail to reproduce the decode-time causal impact. Finally, we connect the shared decode-time channel to steering reliability. Common steering vectors overlap this channel; projecting out that overlap exposes task-dependent utility–robustness trade-offs rather than a universal repair, while decode-time evaluation provides a better ranking signal for held-out KV-cached decoding utility than prefill-based proxies. Together, this frames steering brittleness as partly an inference-regime alignment problem, not only a prompt-template artifact.

Our key contributions are:

1. We propose DECODESHARE, a protocol that identifies and quantifies cross-task shared subspaces *directly from KV-cached decode-time hidden states*.

2. Using decode-only interventions under matched budgets, we show the shared decode subspace is causally important for decisions and that prefill-estimated bases often fail to capture these decode-time causal effects.

3. We connect the decode-shared subspace to steering reliability; it overlaps steering vectors, and decode-time evaluation provides a better ranking signal for held-out KV-cached decoding utility than prefill-based proxies.

We position this work as a *protocol and evaluation* contribution: a practical way to find and causally test decision-relevant shared subspaces under real KV-cached decoding.

## 2. Methodology

### 2.1. Preliminary

**KV-cached inference and decode-time hidden states.** Let $f_\theta$ be a decoder-only Transformer with $L$ layers and hidden width $d$. Under KV caching, inference consists of a *prefill* pass over the full prompt, followed by iterative *decode* steps. Each decode step processes a single current token while reusing cached keys/values from previous tokens.

Fix a single layer $\ell$. Let $h_\ell^{(s)} \in \mathbb{R}^d$ denote the layer-$\ell$ hidden state of the current token at decode step $s$ under KV-cached decoding. We call $h_\ell^{(s)}$ the *decode-time hidden state*. When emphasizing that it drives the next-token logits (and thus decision accuracy), we also refer to it as the *decode-time decision state*. Unless otherwise stated, we run KV-cached decoding for up to $K$ steps and collect decode-time hidden states across steps (treating steps uniformly when pooling).

**Prefill vs. decode states.** We distinguish hidden states obtained in *prefill* pass (computed on the full prompt) from

decode-time hidden states obtained during KV-cached decoding. In all experiments, the prefill state is taken at the last prompt position to provide a single-token reference point for comparison (formal definitions in Appendix 6).

**Alignment principle.** All causal interventions in this paper act on $h_\ell^{(s)}$ *only during KV-cached decode steps* (we do not modify the prefill computation). Accordingly, we estimate candidate subspaces from decode-time hidden states and intervene on the same decode-time states, avoiding estimator-intervention mismatch.

## 2.2. Key Hypotheses and Falsification Criteria

Rather than introducing additional independent claims, we test the central hypothesis via three checks: (i) **H1** validates that shared decode-time structure exists beyond chance, (ii) **H2** tests whether the shared subspace is causally relevant at decode time, and (iii) **H3** tests whether *prefill*-estimated directions causally transfer to decode-time decisions under matched budgets.

**H1 Shared decode-time structure exists.** From decode-time hidden states, our sharedness statistic (shared set size $|S_\ell(\tau, m)|$) exceeds what is expected under matched null baselines.

**H2 Decode-time causal relevance.** Removing the estimated shared subspace from the decode-time hidden state $h_\ell^{(s)}$ *only during KV-cached decoding* harms performance more than matched non-shared controls under the same intervention budget.

**H3 Prefill-to-decode causal transfer often fails.** A basis estimated from prefill states does not reliably reproduce the decode-time causal effect of a basis estimated from decode-time hidden states under matched budgets.

**Falsification criteria.** We reject **H1** if the sharedness statistic is not significantly larger than matched null baselines. We reject **H2**–**H3** if the corresponding decode-time intervention effects are not distinguishable from matched controls under the same budget (and, for **H3**, if prefill- and decode-estimated bases yield comparable decode-time effects).

## 2.3. DECODESHARE: Constructing a decode-aligned shared subspace

**Decode-only state collection.** For each task $t \in \mathcal{T}$, we sample $N$ calibration prompts $x \sim \mathcal{D}_t$ and run KV-cached decoding for up to $K$ steps under policy $\pi$ (e.g., greedy). At each decode step $s$ (single-token forward pass), we record the layer-$\ell$ hidden state $h_\ell^{(s)} \in \mathbb{R}^d$. Stacking all recorded states yields $X_{\ell,t} \in \mathbb{R}^{n_{\ell,t} \times d}$. To balance tasks, we subsample each to $n_\ell = \min_t n_{\ell,t}$, obtaining $X'_{\ell,t} \in \mathbb{R}^{n_\ell \times d}$, and

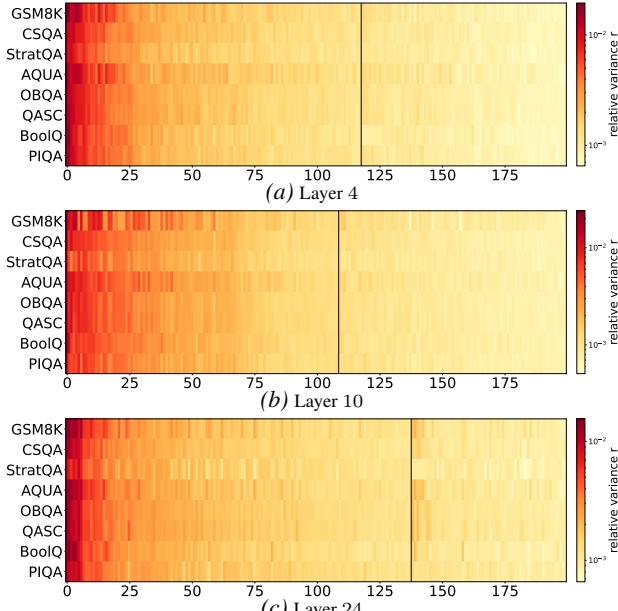

*Figure 2.* **Sharedness heatmaps across layers.** Rows are tasks and columns are pooled PCA components. Color shows the relative variance contribution $r_{\ell,t,i}$. Components are sorted by the number of tasks with $r_{\ell,t,i} \geq \tau$; the vertical line marks the boundary of the shared set $S_\ell(\tau, m)$.

task-center:

$$\tilde{X}_{\ell,t} = X'_{\ell,t} - \mathbf{1}\mu_{\ell,t}^\top, \quad \mu_{\ell,t} = \frac{1}{n_\ell} \sum_{j=1}^{n_\ell} X'_{\ell,t}[j,:].$$

**Pooled PCA basis (computed via SVD).** We pool (stack) all task-centered states:

$$\tilde{X}_\ell = \text{concat}_{t \in \mathcal{T}} \tilde{X}_{\ell,t} \in \mathbb{R}^{(|\mathcal{T}|n_\ell) \times d}.$$

We define a pooled PCA basis by running *PCA* on this pooled matrix, i.e., taking the top eigenvectors of its empirical covariance $C_\ell = \frac{1}{|\mathcal{T}|n_\ell} \tilde{X}_\ell^\top \tilde{X}_\ell$. In implementation, we obtain the same PCA directions by an SVD of the pooled matrix:

$$\tilde{X}_\ell = U \, \text{diag}(\sigma_1, \dots, \sigma_d) \, V^\top,$$

where $\sigma_1 \geq \sigma_2 \geq \cdots \geq 0$ are the singular values. For centered data, the PCA directions are the columns of $V$, and the variance captured by the $i$-th direction is proportional to $\sigma_i^2$. We choose the smallest $k$ that retains at least a fraction $\rho$ of total variance:

$$k = \min \left\{ m : \frac{\sum_{i=1}^m \sigma_i^2}{\sum_{i=1}^d \sigma_i^2} \geq \rho \right\}, \qquad Q_\ell = V_{:,1:k} \in \mathbb{R}^{d \times k}.$$

This pooled PCA basis provides a single cross-task basis set, avoiding per-task PCA basis misalignment. Empirically, the recovered shared core is stable across a wide range of $\rho$ (e.g., $\rho \in [0.9, 0.99]$; Table 10a).

**Identifying shared directions via usage scores and thresholding.** For each task $t \in \mathcal{T}$, we quantify how strongly its decode-time states vary along each pooled PCA direction. After task-centering, we project onto $Q_\ell \in \mathbb{R}^{d \times k}$,

$$Z_{\ell,t} = \tilde{X}_{\ell,t} Q_\ell \in \mathbb{R}^{n_\ell \times k},$$

and compute per-direction variance $v_{\ell,t,i} = \mathrm{Var}(Z_{\ell,t}[:,i])$ for $i \in [k]$. To compare usage patterns across tasks and remove overall scale, we normalize within the pooled subspace and define the *relative variance contribution*

$$r_{\ell,t,i} = \frac{v_{\ell,t,i}}{\sum_{j=1}^{k} v_{\ell,t,j}} \in [0,1], \qquad \sum_{i=1}^{k} r_{\ell,t,i} = 1.$$

Because $r_{\ell,t,i} \geq 0$ and $\sum_{i=1}^{k} r_{\ell,t,i} = 1$, $r_{\ell,t,i}$ is the fraction of task-$t$ variance (within the retained $k$-dimensional pooled subspace) attributed to direction $i$. We mark direction $i$ as *shared* for task $t$ if $r_{\ell,t,i} \geq \tau$. When PCA retention $\rho$ changes $k$, we report the normalized threshold $\tau k$ for comparability: $\tau k = \mathcal{O}(1)$ means the threshold is roughly constant in diffuse baseline $1/k$ (i.e., $\tau = \mathcal{O}(1/k)$).

We aggregate activity across tasks by defining the shared index set:

$$S_\ell(\tau, m) = \left\{ i \in [k] : \left| \{ t \in \mathcal{T} : r_{\ell,t,i} \geq \tau \} \right| \geq m \right\}.$$

We take the decode-aligned shared basis as

$$Q_\ell^{(S)} = Q_\ell[:, S_\ell(\tau, m)] \in \mathbb{R}^{d \times |S_\ell(\tau,m)|}.$$

We denote the corresponding shared subspace by $\mathcal{S}_\ell := \mathrm{span}(Q_\ell^{(S)})$ and its projector by $\Pi_\ell := \Pi(Q_\ell^{(S)}) = Q_\ell^{(S)}(Q_\ell^{(S)})^\top$.

### 2.4. Decode-only causal tests with matched controls

**Intervene at the decode-only boundary.** We test causal relevance by intervening on the decode-time decision state $h_\ell^{(s)}$ only during KV-cached decoding. This preserves the prefill pass and avoids conflating decision-time effects with prompt-processing dynamics.

**Remove a subspace by projection.** At decode step $s$, we ablate a subspace spanned by an orthonormal basis $Q \in \mathbb{R}^{d \times k_Q}$ by subtracting its projection from the state:

$$\tilde{h}_\ell^{(s)} = h_\ell^{(s)} - \alpha \, Q Q^\top h_\ell^{(s)}, \qquad (1)$$

with intervention strength $\alpha \geq 0$. We set $Q = Q_\ell^{(S)}$ for shared-subspace ablations, and use alternative $Q$ for controls below. Crucially, Eq. (1) is applied only on decode pass, leaving prefill untouched.

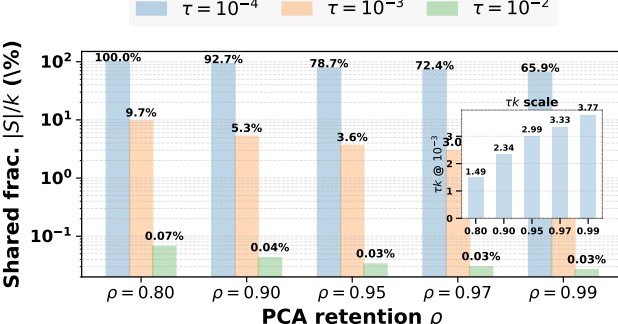

*Figure 3.* **Sensitivity to $\tau$.** Shared ratio $|S_\ell(\tau, m)|/k$ as a function of the sharedness threshold $\tau$ for several PCA retention levels $\rho$. The dashed line marks the default $\tau = 10^{-3}$; see Table 10a and §3.1 for the corresponding shared-core sizes and the $\tau k$ interpretation.

**Size- and energy-matched controls.** Decode-time representations are anisotropic, so removing high-energy subspace can be generically harmful. To isolate the effect of *which directions* are removed (sharedness) from *how much* is removed, we compare to non-shared control subspaces matched in (i) dimension $k_Q$ and (ii) removed energy under the same decode-time distribution. We estimate removed energy on calibration decode states as

$$E(Q) = \mathbb{E}_{h \sim D_{\mathrm{decode}}(\cdot, \ell)} \left[ \| Q Q^\top h \|_2^2 \right],$$

and construct controls so that either we tune $\alpha$ to match $E(Q)$ at fixed dimension, or we match $E(Q)$ by choosing $k_Q$ at fixed $\alpha$ (details in Appendix A).

**Leave-one-task-out (LOTO) to avoid leakage.** To mitigate pooled-estimation leakage concerns, we optionally estimate the shared basis using all but one task and evaluate causal effects on the held-out task using $Q_{\ell,\neg t}^{(S)}$.

**Metrics.** We report two complementary readouts. For discrete-choice tasks, our primary metric is *forced-choice accuracy* computed from teacher-forced conditional log probabilities under a fixed prompt, which isolates decision degradation from formatting/termination failures. We also report task-appropriate native generation metrics (Table 8), including exact match after task-specific extraction when applicable. Since interventions can trigger output instability, we report generation scores with diagnostics: extraction success rate, EOS rate, and average decoded length. We report paired bootstrap confidence intervals and conduct paired randomization (sign-flip) tests on per-example score differences.

## 3. Experiments

**Setup.** Our core experiments span a heterogeneous suite of benchmarks covering arithmetic, commonsense

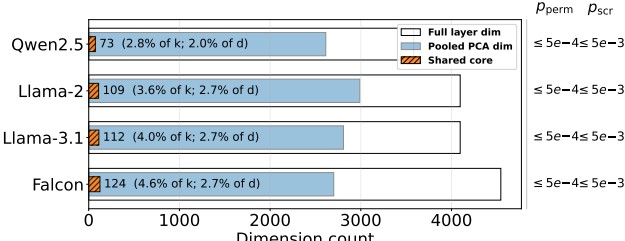

*Figure 4.* **H1: a small but statistically significant shared subspace (layer $\ell=10$).** Scale of the shared core $|\mathcal{S}_\ell(\tau, m)|$ relative to the full layer width $d$ and pooled PCA dimension $k$. Although $|\mathcal{S}_\ell|$ is only a few percent of $d$ (and of $k$), both permutation and scramble tests reject the null (Table 11).

---

**Flip Example (forced-choice; ex_id=aqua-test-9)**

**Gold:** B   **Intervention:** shared-removal at layer $\ell=10$ (decode-only).

**Prompt (truncated):**

```
Question:  A newspaper costs
$4 on Sunday ...  how many
newspapers does it buy on Monday?
Choices:  A)45 B)15 C)60 D)30
E)75
```

---

**baseline**: B (correct=true, margin=+1.08)
**shared-removed**: A (correct=false, margin=-3.30)     *(flip)*

*Figure 5.* **Flip case study (forced-choice).** The baseline forced-choice prediction matches the gold answer. Removing the shared decode subspace at layer $\ell = 10$ during cached decoding flips the predicted choice demonstrating that the shared decode pathway can be *causally necessary*.

and knowledge-intensive QA, verification, and physical reasoning (Table 8). We additionally run targeted robustness checks across multiple backbone models including `meta-llama/Llama-2-7b-chat-hf`, `Qwen2.5-7B-Instruct`, and `Falcon-7b-instruct` and at representative early/mid/late layers (e.g., $\ell \in \{4, 10, 24\}$). For each task, we estimate subspaces from 128 calibration prompts and evaluate on held-out examples under decoding, reporting paired bootstrap confidence intervals and paired sign-flip tests. Beyond the core suite, auxiliary experiments cover additional settings, including coding and language-understanding benchmarks (e.g., HumanEval, SST-2, RTE) and a lexicon-based style set. Unless otherwise noted, we report decision-level accuracy with forced-choice readout (see §2.4 and Table 8).

### 3.1. H1 Shared structure exists at decision time

We begin by testing **H1**: whether decode-time decision states exhibit cross-task shared structure beyond chance. We compute the shared-set size $T_\ell \triangleq |S_\ell(\tau, m)|$ at layer $\ell$ and compare it against the two task-preserving nulls in §2.3. We reject H1 if $T_\ell$ is not significantly larger than either null (or collapses to near-zero across layers/models).

**H1.1 Existence and significance under strong nulls.** Using the strict all-tasks sharing rule ($m_{\text{shared}} = \text{all}$) at a fixed internal layer, we consistently recover a small but nontrivial shared set across models (Table 11). Against both task-preserving null baselines (within-task permutation and orthogonal-scramble; §2.3), the null shared counts remain near-zero and never match the observed $T_\ell$, so the Monte Carlo tests reach the finite-sampling upper-bound regime reported in Table 11. Together, this rules out chance-level explanations under strong nulls and supports that the decode-time shared subspace is a robust signal. (Full per-layer sweeps are deferred to Appendix Table 12.)

**H1.2 Threshold calibration and robustness.** We next test sensitivity to PCA retention $\rho$ (which changes the re-tained dimension $k$) and the sharedness threshold $\tau$. To compare thresholds across different $k$, we interpret $\tau$ relative to the diffuse baseline $1/k$ and summarize stringency by the dimensionless scale $\tau k$ (equivalently, $\tau = \mathcal{O}(1/k)$ when $\tau k = \mathcal{O}(1)$). Sweeping $\rho$ and $\tau$ (Fig. 3) reveals a stable intermediate regime where the shared set remains compact and varies smoothly with $\rho$, whereas overly permissive/strict thresholds lead to the expected degenerate behaviors (nearly-all shared vs. near-empty). Our default $\tau = 10^{-3}$ lies in this scale-compatible regime across the sweep (Fig. 3, inset figure); additional calibrations and variants are in Appendix B.1.

**Where is sharing stronger?** As an auxiliary analysis, within-category pairs show notably stronger sharing for mathematical reasoning, while other coarse categories are close to the mixed-category baseline (Appendix B.1, Fig. 13a).

### 3.2. H2 Decode-time shared subspace is causal

We test whether the estimated shared subspace $\mathcal{S}_\ell$ is *causally necessary* for decode-time decisions under KV-cached decoding. We use three complementary checks: (i) decode-only ablation versus budget-matched non-shared controls, (ii) decision-level evaluation and leave-one-task-out (LOTO) re-estimation to rule out scoring/leakage confounds, and (iii) patchback to test sufficiency and specificity (§2.4; Appendix B.2). The formal definition of patchback and flips are available at Appendix 6.

**H2.1 Decode-only ablation vs. matched controls** Removing $\mathcal{S}_\ell$ only during KV-cached decoding causes a consistent performance collapse across tasks and models (Ta-

ble 30; Fig. 7), whereas dimension/energy-matched non-shared controls stay near baseline. This separation indicates a direction-specific effect rather than a generic consequence of reducing activation energy or rank.

Because open-ended generation accuracy can be confounded by format/termination failures (e.g., extraction/EOS issues), we also evaluate a staged protocol that constrains only the final answer format. The same shared–control gap persists under staging (Table 32), motivating decision-level tests next.

**H2.2 Decision-level degradation** To isolate decision quality from generation formatting, we evaluate a forced-choice readout that ranks candidate completions (§2.4). Shared-subspace removal still produces a clear decision-level drop, while the matched non-shared control remains near baseline (Table 31, Part B; Fig. 7).

We further rule out task leakage in subspace construction by re-estimating $\mathcal{S}_\ell$ in a leave-one-task-out manner (LOTO). The qualitative signature is unchanged: LOTO shared removal remains harmful under generation (Table 31, Part A) and under forced-choice (Table 31, Part B), while matched controls stay near baseline (Table 31). Together, these results support that $\mathcal{S}_\ell$ is causally involved in decode-time *decisions*, not merely output formatting.

**Case study (flip).** To make the decision-level effect concrete, Section 3.1 shows a baseline-correct AQuA example whose prediction flips under decode-only removal of the shared decode-time subspace $\mathcal{S}_\ell$ (at $\ell = 10$), illustrating causal necessity.

**H2.3 Patchback closes the causal loop.** Decode-only ablation establishes necessity; patchback tests whether the removed shared component is also sufficient to *rescue* the decision. We record the shared component $\Pi_\ell h_\ell^{(s)}$ from the unmodified run, rerun the ablated model, and restore *only* this shared component over either the full intervention window or a single decode step (details in §2.4).

Across models/layers, targeted patching rescues a large fraction of ablation-induced flips, while energy-matched controls (random vectors within $\mathcal{S}_\ell$, random subspaces of matched dimension, or patching outside $\mathcal{S}_\ell$) do not (Table 1). In particular, patching outside $\mathcal{S}_\ell$ yields no rescue throughout, demonstrating strong specificity.

**Case study (patchback).** Section 3.2 illustrates a representative forced-choice example: patching back only the removed *shared* component over a narrow window $W = \{0, 1\}$ restores the baseline answer, whereas energy/dimension-matched controls (random subspaces or non-shared patches) fail.

---

**Patchback example (boolq; ex_id=boolq-validation-3)**

**Dataset**: boolq
**Gold**: B
**Prompt** (truncated):

```
Question:  Passage:  Open carry
is also legal throughout North
Carolina.  In the town of Chapel
Hill, open carry is restricted to
guns of a certain minimum size ...
\nQuestion do you need a permit
to open carry in nc Choices:  A)
Yes B) No Reason step by step.
At the end, write exactly one
line:  "Final answer:  <A/B>".
```

---

**baseline**: B (correct=true, margin=+5.19)
**patchback (W={0,1})**: B (correct=true, margin=+5.19)
**rand subspace**: A (correct=false, margin=-3.66)
**time-shuf donor**: B (correct=true, margin=+2.84)
**non-shared patch**: A (correct=false, margin=-1.96)

*Figure 6.* **Patchback case study (forced-choice).** Patching back only the removed *shared* component over a narrow decode window $W = \{0, 1\}$ restores the baseline answer, while energy/dimension-matched controls do not.

Two additional diagnostics address common patchback confounds (Appendix B.2). On the AQuA flip-set, we run *transfer-donor* patching by taking the recorded shared component from a *different* baseline-correct example (optionally from a different task) and patching it into the ablated run. This rescues flips about as often as self-donor patching, so the rescue is not just copying a sample-specific answer. For open-answer patchback, we run a *time-shuffle* control by taking the shared component from a different decode step of the same example and using it as the donor. This works almost as well as `Patched(self)`, which means the shared signal in $\mathcal{S}_\ell$ changes little across decode steps and a single-step patch is often enough.

### 3.3. H3 Prefill-to-decode transfer often fails

**Setup.** We test whether the shared-subspace estimate must be aligned to the *decode-time* distribution. At the same layer $\ell$, we estimate shared bases from **decode** states versus **prefill** states with matched rank $k_{\text{match}}$, and evaluate the resulting $2 \times 2$ estimator–intervention grid under identical budgets and matched random controls (Table 7). We fix the intervention locus to decode-only and then swap the estimator (decode-est vs. prefill-est), which directly tests prefill-decode mismatch.

**Geometric mismatch.** The prefill- and decode-estimated shared subspaces are strongly misaligned even after rank

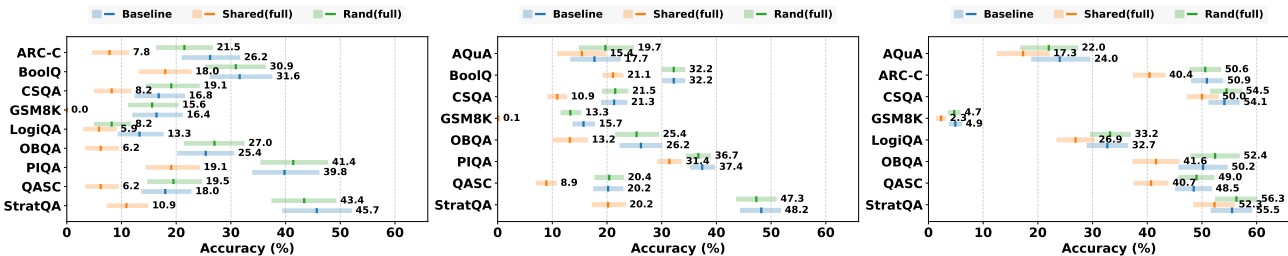

*Figure 7.* **Subspace intervention results.** From left to right: (a) All-tasks Generation, (b) LOTO Generation, (c) LOTO Forced-Choice Generation. Rectangles show 95% CIs; vertical ticks indicate means; numbers are mean accuracies. More experimental details are available in Appendix B.2, §2.4, and Table 31, 32.

*Table 1.* **Patchback on flips (multiple-choice), aggregated over tasks.** $|\mathcal{F}|$ counts baseline → ablated flips. Rescue rates are computed on $\mathcal{F}$ and aggregated across tasks by summing rescued/total counts. Green columns are targeted patching along the shared decode subspace $S_\ell$; blue columns are controls.

| Model | Layer | $|\mathcal{F}|$ | Targeted patch | | Controls | | |
|---|---|---|---|---|---|---|---|
| | | | Patched@0 | Patched@full | RandVec (shared) | Rand Subspace | Nonshared Patch |
| Llama-2-7B-Chat | 4 | 202 | 84.2% (170/202) | **100.0%** **(202/202)** | 19.3% (39/202) | 16.3% (33/202) | 0.0% (0/202) |
| Llama-2-7B-Chat | 10 | 449 | 85.7% (385/449) | **100.0%** **(449/449)** | 7.6% (34/449) | 5.3% (24/449) | 0.0% (0/449) |
| Llama-2-7B-Chat | 24 | 297 | 65.0% (193/297) | **100.0%** **(297/297)** | 8.8% (26/297) | 6.1% (18/297) | 0.0% (0/297) |
| Falcon-7B-Instruct | 4 | 91 | 100.0% (91/91) | **100.0%** **(91/91)** | 9.9% (9/91) | 0.0% (0/91) | 0.0% (0/91) |
| Falcon-7B-Instruct | 10 | 97 | 100.0% (97/97) | **100.0%** **(97/97)** | 0.0% (0/97) | 1.0% (1/97) | 0.0% (0/97) |
| Falcon-7B-Instruct | 24 | 192 | 100.0% (192/192) | **100.0%** **(192/192)** | 19.3% (37/192) | 19.8% (38/192) | 0.0% (0/192) |
| Qwen2.5-7B-Instruct | 4 | 325 | 100.0% (325/325) | **100.0%** **(325/325)** | 32.6% (106/325) | 32.9% (107/325) | 0.0% (0/325) |
| Qwen2.5-7B-Instruct | 10 | 548 | 100.0% (548/548) | **100.0%** **(548/548)** | 36.1% (198/548) | 49.8% (273/548) | 0.0% (0/548) |
| Qwen2.5-7B-Instruct | 24 | 453 | 100.0% (453/453) | **100.0%** **(453/453)** | 37.5% (170/453) | 39.3% (178/453) | 0.0% (0/453) |

matching (Table 20; layer sweep in Appendix B.4), indicating that prefill activations do not provide a reliable geometric proxy for the decode-time shared workspace.

**Causal consequence at decode.** This mismatch matters causally: under decode-only intervention, ablating the *decode-estimated* shared basis produces a large and consistent accuracy drop, while ablating the *prefill-estimated* basis largely removes the effect and behaves like matched random controls (Table 7). Interventions at prefill are negligible across conditions. Together, these results support a distribution-aligned view: to reproduce decode-locus causal effects under matched budgets, the shared subspace must be estimated from decode-time states.

**What does the shared channel encode?** While our causal interventions establish the decision-relevance of the decode-shared subspace, they do not specify its computational vocabulary. To characterize its contents, we decouple the 32D shared workspace into a 3D readout slice $Q_{\mathrm{out}}$ and its 29D residual core $Q_{\mathrm{core}}$ (Table 2).

Causal ablation reveals a clear functional divergence be-

*Table 2.* **The shared channel is not a narrow readout slice.** A 32D Llama-2-7B layer-10 workspace is split into a 3D readout slice $Q_{\mathrm{out}}$ and a 29D core $Q_{\mathrm{core}}$. Panels A and B report forced-choice accuracy (baseline: 44.1%) and held-out probe AP, respectively.

| A. Causal ablation | | | |
|---|---|---|---|
| Ablated subspace | Dim. | Acc. | $\Delta$Acc. |
| Full shared | 32 | 28.5 | $-15.6$ |
| $Q_{\mathrm{out}}$ | 3 | 44.9 | $+0.8$ |
| $Q_{\mathrm{core}}$ | 29 | 27.3 | $-16.8$ |
| **B. Held-out probe AP** | | | |
| Probe tag | $Q_{\mathrm{core}}$ | $Q_{\mathrm{out}}$ | $\Delta$ |
| Reasoning markers | 0.564 | 0.041 | $+0.523$ |
| Step markers | 0.169 | 0.029 | $+0.140$ |
| Digits | 0.966 | 0.132 | $+0.834$ |
| Equation symbols | 0.673 | 0.055 | $+0.618$ |

tween these two components. Suppressing $Q_{\mathrm{core}}$ reproduces the massive performance drop of the full subspace (over 15% loss), whereas removing $Q_{\mathrm{out}}$ leaves the model performance intact (Table 2A). Linear probing results in Panel B explain this divergence: $Q_{\mathrm{core}}$ maintains high predictive accuracy for procedural and symbolic tokens, while $Q_{\mathrm{out}}$ remains

*Table 3.* **Decode-shared directions are enriched for decision-scaffold tokens.** Vocab-alignment scores (mean overlaps) for Llama-2-7B layer 28.

| Token family | Shared | Nonshared | Prefill shared |
|---|---|---|---|
| Answer scaffold | **0.283** | 0.213 | 0.208 |
| Correctness markers | **0.207** | 0.189 | 0.182 |
| Confidence markers | **0.234** | 0.195 | 0.221 |
| Newline | **0.374** | 0.204 | 0.190 |
| Digits | **0.314** | 0.261 | 0.159 |
| Sentiment markers | 0.171 | 0.176 | **0.182** |

*Table 4.* **Decode-aligned ranking better matches held-out decode utility.** The pools contain 32 CAA contrastive vectors, 64 instruction-derived vectors, 64 SAE feature directions, and 100 diagnostic directions.

| Pool | Prefill $\rho$ | Decode $\rho$ | $\Delta$ |
|---|---|---|---|
| CAA contrastive | -0.370 | **0.700** | +1.070 |
| Instruction | 0.172 | **0.767** | +0.595 |
| SAE features | -0.064 | **0.594** | +0.659 |
| Diagnostic | 0.065 | **0.700** | +0.635 |

near baseline. Therefore, the geometric split matches the functional split: the residual core stores structured task information, while the smaller slice serves as a clean interface for reading out tokens.

Second, a logit-lens analysis shows that these decode-shared directions are uniquely enriched for answer-scaffold and formatting tokens, but not for generic sentiment markers (Table 3). This selectivity confirms that the channel does not process all semantic information indiscriminately. Instead, it operates specifically as a decode-time decision scaffold for procedural reasoning and verification.

**Scale Diagnostics.** Targeted non-7B checks further show the same qualitative pattern at 3B, 13B, and 70B; we report these robustness checks in Appendix C.4.

## 4. Downstream Utility

**Steering Robustness.** We study a drop-in repair for arbitrary steering directions $v$ under true KV-cached decoding. From neutral prompts, we estimate an orthonormal decode-time shared basis $Q$ and measure shared overlap $\mathrm{sh}(v) = \|Q^\top v\|_2/\|v\|_2$, which is consistently nontrivial across tasks (Table 6). We then repair $v$ by removing its shared component:

$$v_\alpha = (I - \alpha QQ^\top)v, \qquad \alpha \in [0,1],$$

where $\alpha = 1$ yields the fully repaired $v_{\text{fixed}} = (I - QQ^\top)v$.

Across tasks, the repaired direction remains clearly non-random under KV-cached decoding (positive mean signed shift and separation from energy-matched random controls), while often reducing template sensitivity; the magnitude of

*Table 5.* **Decode-aligned validation selects better deployed vectors.** Metrics are held-out accuracy (utility) evaluated on the real-world downstream task (REAL) for selected vectors.

| Proxy | REAL mean | REAL worst | Flip rate | Regret@1 |
|---|---|---|---|---|
| Prefill-aligned | -0.002 | -0.003 | 0.750 | 0.016 |
| Mixed Stages | -0.002 | -0.003 | 0.750 | 0.016 |
| Decode-aligned | **+0.011** | **+0.010** | **0.083** | **0.003** |

the robustness gain is task-dependent (Table 6). Overall, this supports shared-subspace interference as a concrete source of template brittleness, and motivates $v_\alpha$ as a simple offline knob that trades a small amount of average effect for robustness under KV-cached decoding.

**Stage-aligned vector selection.** Across 260 contrastive, steering, and feature vectors, decode-stage ranking aligns better with downstream decode utility than prefill-stage ranking and mixed prefill-decode ranking (Table 4). This difference directly improves deployment: evaluated on our downstream benchmark, decode-stage validation selects vectors with positive accuracy gains (utility), lower flip rates, and lower regret than prefill-stage or mixed always-on validation (Table 5). This trend occurs because prefill and mixed signals fail to isolate decode-specific token dynamics, often leading to sub-optimal vector choices. In contrast, decode-stage validation precisely identifies vectors that maintain stable steering effects during generation, thereby maximizing downstream deployment utility.

**Style case study.** We illustrate the same knob on a lightweight pirate-lexicon steering task (Appendix C.3). As the retained rank $k$ grows, the shared span captures more of $v$, while the repaired direction $v_{\text{fixed},k}$ becomes numerically orthogonal to it (Table 28). Behaviorally, moderate projection can improve both average effect and worst-case template performance, whereas overly aggressive removal can suppress the steering signal (Table 33), consistent with a denoise–excise tradeoff.

## 5. Related Works

**Causal interventions & patching.** Activation-level causal interventions, including activation patching/causal tracing, causal mediation, and interchange interventions, are widely used to localize and validate mechanisms in LLMs (Vig et al., 2020; Meng et al., 2022; Wang et al., 2023; Geiger et al., 2021; 2024). At the same time, recent analyses emphasize that patching results can be sensitive to metrics and implementation details, and that subspace patching can create interpretability illusions without appropriate controls (Zhang & Nanda, 2024; Makelov et al., 2024). Motivated by these practices, we intervene *only* on KV-cached decode-time decision states and pair shared-subspace ablations with matched nulls and budget-matched controls, then close the

*Table 6.* **Multibench behavior under KV-cache decode.** Mean signed margin shift ($\mu$), template std ($\sigma_{\text{tmpl}}$), worst-case template mean (worst; primary), and worst-case anti-steer rate (anti$_{\text{worst}}$; diagnostic, lower is better). $\text{sh}(v)$ is overlap with the decode-time shared basis. Partial projection: $v_\alpha = v - \alpha Q Q^\top v$; R is energy-matched random control. Rightmost columns report $\Delta$ between $\alpha=1$ and 0.

| Task | Cand. | sh($v$) | $\mu \uparrow$ | | | | $\sigma_{\text{tmpl}} \downarrow$ | | worst $\uparrow$ | | | | anti$_{\text{worst}} \downarrow$ | | | $\Delta$ (1−0) | |
| | | | 0 | .5 | 1 | R | 0 | 1 | 0 | .5 | 1 | R | 0 | 1 | R | $\Delta_{\text{worst}}$ | $\Delta_{\text{anti}}$ |
|---|---|---|---|---|---|---|---|---|---|---|---|---|---|---|---|---|---|
| BoolQ | Y/N | 0.405 | 0.0383 | 0.0357 | 0.0312 | −0.0100 | 0.0019 | 0.0034 | 0.0357 | 0.0321 | 0.0267 | −0.0118 | 0.3633 | 0.3633 | 0.5695 | −0.0090 | 0.0000 |
| RTE | T/F | 0.386 | 0.0173 | 0.0171 | 0.0162 | −0.0005 | 0.0095 | 0.0085 | 0.0104 | 0.0101 | 0.0089 | −0.0033 | 0.4844 | 0.4648 | 0.5195 | −0.0015 | −0.0196 |
| SST-2 | G/B | 0.391 | 0.0099 | 0.0130 | 0.0154 | 0.0012 | 0.0126 | 0.0134 | −0.0007 | 0.0011 | 0.0028 | −0.0082 | 0.4336 | 0.4648 | 0.5594 | +0.0035 | +0.0312 |

*Table 7.* **H3 $2 \times 2$ estimation-intervention grid.** Accuracy (%) under matched $k$ (here $k=48$) at layer $\ell=10$ with the same removal strength ($\alpha=1$) across conditions. Bold indicates the lowest accuracy within each row (ties broken left-to-right).

| Task | Base | Intervene: Decode | | | Intervene: Prefill | | |
| | | Dec-est | Pre-est | Rand | Dec-est | Pre-est | Rand |
|---|---|---|---|---|---|---|---|
| CSQA | 53.3 | **23.1** | 54.3 | 53.5 | 53.2 | 53.3 | 53.3 |
| StratQA | 49.6 | 52.8 | **49.6** | 49.6 | 49.8 | 49.6 | 49.6 |
| PIQA | 69.1 | **52.7** | 67.6 | 69.4 | 69.5 | 69.2 | 69.2 |
| ARC-C | 51.5 | **26.5** | 50.9 | 51.9 | 52.6 | 52.6 | 51.6 |
| OBQA | 51.2 | **27.2** | 51.8 | 51.8 | 53.8 | 52.8 | 51.2 |
| QASC | 48.9 | **13.7** | 50.3 | 49.5 | 49.0 | 49.7 | 48.8 |
| LogiQA | 32.1 | **23.2** | 32.7 | 32.1 | 32.4 | 32.9 | 32.1 |
| **Mean $\Delta$ vs. base** | – | **−19.5** | 0.2 | 0.3 | 0.7 | 0.6 | 0.0 |

loop with patchback specificity tests ( **H2** ).

**KV-cached decoding and cache management.** KV caching is a first-class serving regime for modern LLM inference (e.g., paging-based decoding) and is also central to streaming/long-context systems that explicitly manage rolling caches (Kwon et al., 2023; Xiao et al., 2024). A growing body of work studies what to retain and how to compress KV caches—including eviction/heavy-hitters, low-rank or reconstruction-based compression, and cross-layer schemes (Zhang et al., 2023; Ge et al., 2024; Saxena et al., 2024; Li et al., 2024; Cai et al., 2024; Kim et al., 2025; Chang et al., 2025; Khalaf et al., 2025). These efforts motivate our emphasis on *decode-time* measurements and interventions under true KV-cached decoding; we defer additional system background to Appendix D.1.

**Representation engineering and decision-level readouts.** Representation engineering and activation steering show that low-dimensional directions can reliably influence generations, but their behavior can degrade under context or regime shifts (Turner et al., 2023; Rimsky et al., 2024; Zou et al., 2023; Tan et al., 2024; Deng et al., 2025). To separate decision quality from formatting/termination artifacts in open-ended generation, we adopt decision-level probes based on layer-wise vocabulary readouts (lens-style methods) (Belrose et al., 2023; Geva et al., 2022), and use them to quantify regime-specific degradation under KV-cached decode-time interventions.

**Shared computation and low-dimensional structure.** Circuit-style mechanistic interpretability argues that Transformers can reuse compact computational motifs across diverse behaviors (e.g., induction-like mechanisms), motivat-

ing the search for shared structure across tasks (Elhage et al., 2021; Olsson et al., 2022; Merullo et al., 2024; Bhaskar et al., 2024). Complementarily, representation similarity tools (SVCCA/CKA) and "shared subspace" hypotheses suggest that important computation may concentrate in low-dimensional subspaces that persist across settings (Raghu et al., 2017; Kornblith et al., 2019; Kaushik et al., 2025; Wang et al., 2025). DECODESHARE operationalizes this perspective at the *decode-locus*: we estimate a cross-task shared workspace from KV-cached decision states and validate it with decode-only causal tests ( **H2** ), while explicitly probing estimator–deployment mismatch ( **H3** ).

**Differences from closely related subspace work.** Prior work studies shared/orthogonal subspaces mainly at the *parameter* level (e.g., task-update geometry or orthogonal LoRA/merging) (Arturi et al., 2025; Zhang & Zhou, 2025), or compares shared semantics across models/scales or modalities (Son et al., 2025; Whitaker et al., 2025). We instead identify an *activation-level* workspace that appears in KV-cached *decode-time decision states* within a fixed model and test it causally with decode-only interventions, matched controls, and patchback. Related projection/subspace-learning methods remove nuisance directions or learn transfer structure during tuning (Xie et al., 2022; Falissard et al., 2023); our target is a task-shared *decode-time* workspace, and we explicitly study when prefill-estimated bases fail to transfer to decode-time causal effects ( **H3** ). MSRS composes multiple subspaces for attribute control (Jiang et al., 2025); our focus is diagnostic rather than steering.

## 6. Conclusion

We propose DECODESHARE, a protocol that estimates a cross-task shared subspace from pooled decode-time activations and tests its causal role via projection interventions with budget-matched controls. The shared subspace is low-dimensional yet consistently affects decoding decisions. We also observe a regime gap: prefill-estimated directions can misalign with decode-time behavior. Crucially, our stage-aligned protocol exploits this gap to successfully optimize downstream vector selection over prefill or mixed baselines. More broadly, our results highlight decode-time state geometry as a first-class object for mechanistic analysis under KV caching, suggesting that intervention design should be evaluated in the serving regime rather than inferred from prefill statistics alone. We hope this encourages decode-centric evaluations in future LLM interpretability works.

## Acknowledgment

This work was supported by the National Science Foundation (NSF) under Grant No. IIS-2451480 and Grant No. 2112562, and the Army Research Office (ARO) under Grant No. W911NF-23-2-0224.

## Impact Statement

DecodeShare provides a protocol for exploring shared structure in model decisions during decoding. It estimates a low-dimensional subspace from KV-cached decode-time activations across tasks, then projects it out during decoding and measures output changes using matched random and energy controls. This supports causal tests of whether shared decode-time structure influences decisions and how it overlaps with steering directions studied in current works on steering vectors and style control. Potential positive impacts are primarily methodological and practical: clearer cross-task comparisons of decode-time representations, more reproducible intervention studies via matched controls, and the ability to separate the functional roles of steering directions from task-general decode channels. By shifting focus from prefill-based proxies to decode-time evaluations, the protocol also provides a more reliable signal for downstream deployment. Potential negative impacts appear limited. The protocol requires white-box access and per-model estimation, so it is not directly applicable to black-box settings. However, because the identified subspace generalizes robustly across tasks, actors with internal access could potentially exploit these universal directions for broader behavioral manipulation; we recommend conservative intervention budgets and reporting safety and stability metrics.

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

*Table 8.* Benchmark taxonomy used in our experiments. For discrete-choice tasks, we use decision-level forced-choice accuracy (FC-Acc; teacher-forced conditional logprob) as the primary readout; we also report generation-based scores (gen-acc/EM) with collapse diagnostics (extraction success rate, EOS rate, and average decoded length). For open-answer settings, GSM8K is scored by EM after extraction (gen-math) and HumanEval uses a compile-based code-generation metric; both also include pair-logprob as a decision proxy where reported.

| Category | Benchmark | I/O format | Metrics reported in paper |
|---|---|---|---|
| Arithmetic | GSM8K | open (numeric) | EM (gen-math) + Extr/EOS/Len; pair-logprob |
| Arithmetic | AQuA | MCQ | FC-Acc; gen-acc (+ Extr/EOS/Len) |
| Commonsense QA | CommonsenseQA (CSQA) | MCQ | FC-Acc; gen-acc (+ Extr/EOS/Len) |
| Physical reasoning | PIQA | 2-way MCQ | FC-Acc; gen-acc (+ Extr/EOS/Len) |
| Knowledge QA | OpenBookQA (OBQA) | MCQ | FC-Acc; gen-acc (+ Extr/EOS/Len) |
| Knowledge QA | QASC | MCQ | FC-Acc; gen-acc (+ Extr/EOS/Len) |
| Verification | BoolQ | binary | FC-Acc; gen-acc (+ Extr/EOS/Len) |
| Verification | StrategyQA | binary | FC-Acc; gen-acc (+ Extr/EOS/Len) |
| Logical/QA | ARC-Challenge (ARC-C) | MCQ | FC-Acc; gen-acc (+ Extr/EOS/Len) |
| Logical/QA | LogiQA | MCQ | FC-Acc; gen-acc (+ Extr/EOS/Len) |
| Coding (aux) | HumanEval | open (code) | gen-code-compile; pair-logprob |
| NLU (aux) | SST-2 | binary | steering utility metrics (margins); FC readout when used |
| NLU (aux) | RTE | binary | steering utility metrics (margins); FC readout when used |
| Style (aux) | Pirate lexicon set | open (style) | pirate-hit success / intensity |

**Additional Definitions.** Let $f_\theta$ be a decoder-only Transformer with $L$ layers and hidden width $d$. Under KV caching, each step is a cached decode forward pass with sequence length $= 1$. Fix a layer $\ell$ and let $h_\ell^{(s)} \in \mathbb{R}^d$ be the layer-$\ell$ hidden state of the (only) token during cached decode step $s$. We call $h_\ell^{(s)}$ the *decode-time decision state*, i.e., the layer-$\ell$ hidden state of the current token at cached decode step $s$. Unless stated otherwise, we sample $S \sim \mathrm{Unif}\{1, \ldots, K\}$ and use $h_\ell^{(S)}$. For each task $t \in \mathcal{T}$, let $D_{\mathrm{prefill}}(t, \ell)$ be the distribution of layer-$\ell$ hidden states during the prefill pass, and let $D_{\mathrm{decode}}(t, \ell; \pi, K)$ be the distribution of $\{h_\ell^{(s)}\}_{s=1}^{K}$ induced by cached decoding up to horizon $K$ under policy $\pi$. In all experiments, $D_{\mathrm{prefill}}(t, \ell)$ is formed from the hidden state at the last prompt position $i = n(u)$ to align with the sequence-length-one decode states.

**Definition of Patchback Experiments.** For a discrete-choice task, define the flip set at layer $\ell$ as

$$\mathcal{F}_\ell = \{x : \; \hat{y}_{\mathrm{base}}(x) = y(x) \; \wedge \; \hat{y}_{\mathrm{abl}}(x) \neq y(x)\}.$$

Let $\mathcal{S}_\ell$ be the shared decode-time subspace at layer $\ell$ with orthonormal basis $Q_\ell^{(S)} \in \mathbb{R}^{d \times k_S}$, and let $\Pi_\ell := Q_\ell^{(S)}\big(Q_\ell^{(S)}\big)^\top$ denote the orthogonal projector onto $\mathcal{S}_\ell$. For each $x \in \mathcal{F}_\ell$ and decode step $s$, record the baseline shared component $p_s(x) = \Pi_\ell h_{\ell,\mathrm{base}}^{(s)}(x)$.

We then re-run the ablated model and patch back only the shared component over a decode-step window $W \subseteq \{1, \ldots, K\}$ by setting

$$h_{\ell,\mathrm{patch}}^{(s)}(x) = h_{\ell,\mathrm{abl}}^{(s)}(x) + \mathbf{1}[s \in W]\Big(p_s(x) - \Pi_\ell h_{\ell,\mathrm{abl}}^{(s)}(x)\Big).$$

Equivalently, for $s \in W$ we replace the shared component $\Pi_\ell h_{\ell,\mathrm{abl}}^{(s)}(x)$ with the donor component $p_s(x)$, while keeping the orthogonal complement $(I - \Pi_\ell)h_{\ell,\mathrm{abl}}^{(s)}(x)$ unchanged.

We report the rescue rate on flips:

$$\mathrm{Rescue}(\mathcal{F}_\ell) = \frac{1}{|\mathcal{F}_\ell|} \sum_{x \in \mathcal{F}_\ell} \mathbf{1}[\hat{y}_{\mathrm{patch}}(x) = y(x)],$$

and aggregate across tasks by summing rescued/total counts.

## A. Additional Controls and Robustness Experiments

**Appendix scope.** This section reports controls and robustness checks for three confounds: (i) *distribution alignment* (all estimation, interventions, and evaluation are aligned to KV-cached, seq-len-1 decision states rather than prefill), (ii) *energy confounds* (we include energy diagnostics and energy-matched non-shared controls to rule out "stronger perturbation" explanations), and (iii) *decision-level evaluation* (we use forced-choice log-probability scoring to isolate decision degradation from EOS/format/extraction failures). Concretely, §A.1 analyzes removed-energy statistics, §A.2

presents two complementary energy-matched causal controls (k-match and $\alpha$-scaling mean-match) under strict decode-only interventions, and §B.4 reports the prefill–decode intervention grid supporting decode-specificity (H3). All appendix experiments intervene at the same decode-only boundary: for each prompt we prefill to build the KV cache, then run a seq-len-1 cached decode call at the prompt boundary, collect the layer-$\ell$ decision state, and apply the intervention at this locus, which matches both the hook site and cache-advanced forced-choice evaluation.

### A.1. Experiment Group 1: Why High Shared-Space Energy is Expected

**Motivation (energy confound as a diagnosis problem).** A natural concern about decode-time subspace removal is an *energy confound*: if the identified shared subspace carries much higher projection energy than a same-dimensional baseline, then removing it can appear disproportionately destructive simply because it removes more signal under (2). Before making any causal claim, we therefore ask a prior diagnostic question:

> *Question: Why is high decode-time energy in the estimated shared subspace expected in the first place?*

Our answer is: high shared-space energy is not, by itself, evidence of an estimator artifact (e.g., "PCA artifacts"). It is an expected consequence of (i) anisotropic decode-time activations, and (ii) the way "shared" directions are defined in our protocol (directions that consistently explain non-trivial variance across tasks). Group 2 then addresses the causal question using energy-matched controls.

**Setup and diagnostics.** We intervene on KV-cached decode last-token states $h_\ell$ via projection removal

$$\tilde{h} \; = \; h - \alpha Q Q^\top h, \tag{2}$$

where $Q \in \mathbb{R}^{d \times k}$ has orthonormal columns and $\alpha \geq 0$. For any orthonormal basis $Q$ and state $h$, define projection energy and its per-sample fraction

$$E(h; Q) \; := \; \|Q^\top h\|_2^2, \qquad r(h; Q) \; := \; \frac{\|Q^\top h\|_2^2}{\|h\|_2^2} \in [0, 1]. \tag{3}$$

Empirically we report $\mathbb{E}[r(h; Q)]$ on the prompt-boundary decode-last distribution. For theory it is also convenient to use the ratio of expectations

$$\bar{r}(Q) \; := \; \frac{\mathbb{E}[\|Q^\top h\|_2^2]}{\mathbb{E}[\|h\|_2^2]}. \tag{4}$$

These coincide for Haar-random subspaces; see Lemma A.1. We report $\mathbb{E}[r(h; Q)]$ in plots because it is an interpretable per-example fraction, but the energy matching controls use the numerator $\mathbb{E}[\|Q^\top h\|_2^2]$, since interventions scale with that quantity.

**Why high shared-space energy is expected.** Let $\Sigma := \mathbb{E}[hh^\top]$ be the second moment (with the same centering convention as in §2.3 when applicable). Then for any orthonormal $Q \in \mathbb{R}^{d \times k}$,

$$\mathbb{E}[E(h; Q)] \; = \; \mathbb{E}[\|Q^\top h\|_2^2] \; = \; \text{tr}(Q^\top \Sigma Q). \tag{5}$$

Thus high-energy subspaces are exactly those aligned with large-eigenvalue directions of $\Sigma$. The following three results formalize the expected behavior: (1) the diffuse baseline, (2) anisotropy guarantees the existence of intrinsically high-energy low-dimensional subspaces (and PCA recovers one), and (3) our *shared* selection criterion further biases toward above-diffuse energy *within the pooled span*.

**Lemma A.1** (Haar-random subspaces have exactly $k/d$ expected energy fraction). *Let $Q_{\text{rand}}$ be Haar-uniform on the Grassmannian (a random $k$-dimensional subspace), independent of $h \in \mathbb{R}^d$. Then*

$$\mathbb{E}_{Q_{\text{rand}}}\big[r(h; Q_{\text{rand}}) \mid h\big] = \frac{k}{d}, \qquad \text{and hence } \mathbb{E}[r(h; Q_{\text{rand}})] = \frac{k}{d}.$$

*Moreover, $\bar{r}(Q_{\text{rand}}) = k/d$ as well.*

*Proof.* Rotational symmetry gives $\mathbb{E}[Q_{\text{rand}} Q_{\text{rand}}^\top] = \frac{k}{d} I$. Thus $\mathbb{E}[\|Q_{\text{rand}}^\top h\|_2^2 \mid h] = h^\top \mathbb{E}[Q_{\text{rand}} Q_{\text{rand}}^\top] h = \frac{k}{d} \|h\|_2^2$, so $\mathbb{E}[r(h; Q_{\text{rand}}) \mid h] = k/d$. Taking expectation over $h$ yields $\mathbb{E}[r(h; Q_{\text{rand}})] = k/d$. Finally, $\bar{r}(Q_{\text{rand}}) = \mathbb{E}[\|Q_{\text{rand}}^\top h\|_2^2]/\mathbb{E}[\|h\|_2^2] = k/d$. $\qquad\square$

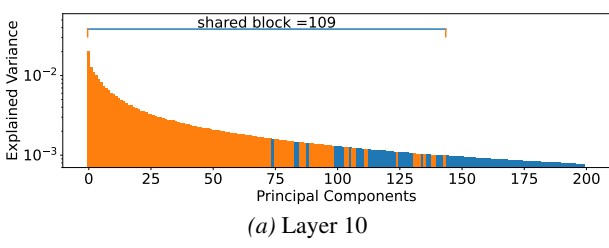
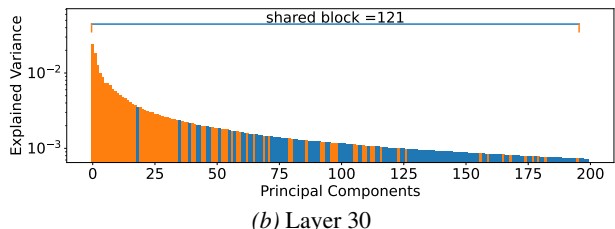

*(a)* Layer 10          *(b)* Layer 30

*Figure 8.* **Pooled PCA spectrum with shared components marked.** Explained-variance ratios (log scale) for pooled PCA components at two layers. Components belonging to the identified shared set are visually distinguished, indicating where the shared directions sit in the overall variance spectrum.

**Theorem A.2** (Top-variance subspaces exceed random energy under anisotropy)**.** *Let $h \in \mathbb{R}^d$ be zero-mean with second moment $\Sigma = \mathbb{E}[hh^\top] \succeq 0$ and eigenvalues $\lambda_1 \geq \cdots \geq \lambda_d$. For any orthonormal $Q \in \mathbb{R}^{d \times k}$,*

$$\mathbb{E}\big[\|Q^\top h\|_2^2\big] \; = \; \mathrm{tr}(Q^\top \Sigma Q).$$

*Let $Q_{\mathrm{top}}$ span the top-$k$ eigenspace of $\Sigma$. Then*

$$\mathbb{E}\big[\|Q_{\mathrm{top}}^\top h\|_2^2\big] = \sum_{i=1}^{k} \lambda_i \; \geq \; \frac{k}{d} \sum_{i=1}^{d} \lambda_i = \frac{k}{d} \mathbb{E}[\|h\|_2^2],$$

*with equality iff $\lambda_1 = \cdots = \lambda_d$ (isotropy). Equivalently, $\bar{r}(Q_{\mathrm{top}}) \geq k/d$, strictly if $\Sigma$ is anisotropic.*

*Proof.* The identity is (5).

Ky Fan's maximum principle yields $\max_{Q^\top Q = I_k} \mathrm{tr}(Q^\top \Sigma Q) = \sum_{i=1}^{k} \lambda_i$. Since the mean of the top-$k$ eigenvalues is at least the overall mean, $\frac{1}{k} \sum_{i=1}^{k} \lambda_i \geq \frac{1}{d} \sum_{i=1}^{d} \lambda_i$, with equality iff all $\lambda_i$ are equal. $\qquad\square$

A concrete formalization of "low-rank but high-energy" is the factor-plus-noise model $h = Uz + \varepsilon$ with $U^\top U = I_r$, $\mathrm{Cov}(z) = \Lambda \succeq 0$, $\mathrm{Cov}(\varepsilon) = \sigma^2 I$, for which $\Sigma = U\Lambda U^\top + \sigma^2 I$ and $\bar{r}(U) = \frac{\mathrm{tr}(\Lambda) + r\sigma^2}{\mathrm{tr}(\Lambda) + d\sigma^2}$. Thus a small $r$ can still capture a large energy fraction when $\mathrm{tr}(\Lambda) \gg d\sigma^2$.

**Lemma A.3** (Shareness threshold implies above-diffuse energy *within* the pooled PCA span)**.** *Fix a layer $\ell$ and the pooled PCA basis $Q_\ell = [q_{\ell,1}, \ldots, q_{\ell,k}] \in \mathbb{R}^{d \times k}$ from §3.3. For each task $t \in \mathcal{T}$, let $h \sim D_{\mathrm{decode}}(t, \ell)$ denote a task-centered decode-last state (as in §3.3), and define*

$$v_{\ell,t,i} \; := \; \mathrm{Var}\big(q_{\ell,i}^\top h\big) \; = \; \mathbb{E}\big[(q_{\ell,i}^\top h)^2\big], \qquad V_{\ell,t} \; := \; \sum_{j=1}^{k} v_{\ell,t,j} \; = \; \mathbb{E}\big[\|Q_\ell^\top h\|_2^2\big].$$

*Define the within-span variance share $r_{\ell,t,i} := \frac{v_{\ell,t,i}}{V_{\ell,t}} \in [0,1]$, so that $\sum_{i=1}^{k} r_{\ell,t,i} = 1$. Let $S_\ell(\tau, m)$ be the set of indices $i$ such that $r_{\ell,t,i} \geq \tau$ holds for at least $m$ tasks, and let $Q_\ell^{(S)} = Q_\ell[:, S_\ell(\tau, m)]$.*

***(i)*** *For every task $t$,*

$$\frac{\mathbb{E}\Big[\|(Q_\ell^{(S)})^\top h\|_2^2\Big]}{\mathbb{E}\big[\|Q_\ell^\top h\|_2^2\big]} \; = \; \sum_{i \in S_\ell(\tau,m)} r_{\ell,t,i}.$$

***(ii)*** *Averaging over tasks,*

$$\frac{1}{|\mathcal{T}|} \sum_{t \in \mathcal{T}} \sum_{i \in S_\ell(\tau,m)} r_{\ell,t,i} \; \geq \; \frac{m\tau}{|\mathcal{T}|} |S_\ell(\tau, m)|.$$

*(Consequently $|S_\ell(\tau, m)| \leq |\mathcal{T}|/(m\tau)$.)*

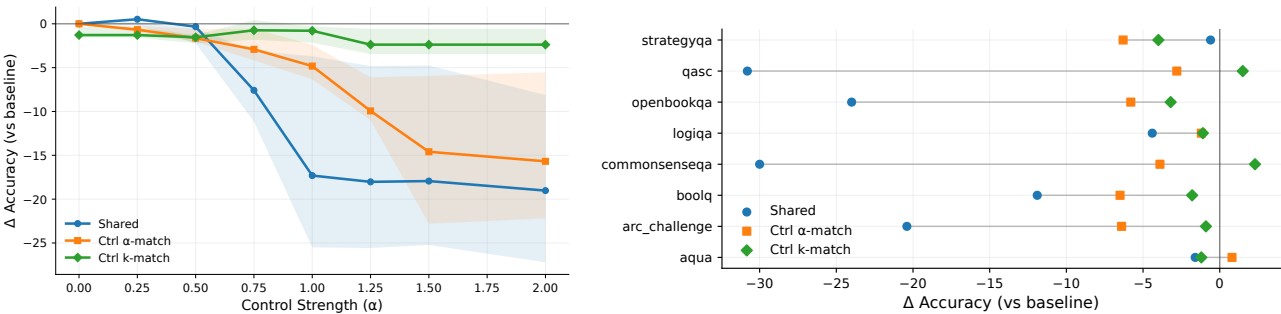

*(a)* **Task-aggregate $\Delta$ accuracy vs. $\alpha$.** Shared removal di- *(b)* **Per-task $\Delta$ accuracy at $\alpha = 1$.** Shared is consistently most harmful;
verges sharply for $\alpha \gtrsim 0.75$ while energy-matched controls  k-match remains near baseline.
stay closer to baseline.

*Figure 9.* **Energy-matched causal controls under strict decode-only interventions (Llama-2-7B-Chat, layer 10).** We compare
removing the shared subspace against two energy-matched controls (fixed-dimension $\alpha$-match and expanded-dimension k-match),
evaluated by cache-advanced forced-choice logprob scoring on KV-cached decode steps (seq len $= 1$).

*(iii)* *If one selects a uniformly random index subset $S_{\mathrm{rand}} \subseteq [k]$ with $|S_{\mathrm{rand}}| = |S_\ell|$, then for every task $t$,*

$$\mathbb{E}_{S_{\mathrm{rand}}}\Big[\sum_{i \in S_{\mathrm{rand}}} r_{\ell,t,i}\Big] = \frac{|S_\ell|}{k}.$$

*Therefore the shared set's average within-span energy share is lower-bounded by a factor $\left(\frac{m\tau k}{|\mathcal{T}|}\right)$ relative to the diffuse
in-span baseline (clipped at $1$).*

*Proof.* (i) Orthonormality gives $\mathbb{E}[\|(Q_\ell^{(S)})^\top h\|_2^2] = \sum_{i \in S_\ell} \mathbb{E}[(q_{\ell,i}^\top h)^2] = \sum_{i \in S_\ell} v_{\ell,t,i}$ and $\mathbb{E}[\|Q_\ell^\top h\|_2^2] = \sum_{j=1}^{k} v_{\ell,t,j} = V_{\ell,t}$, hence the ratio equals $\sum_{i \in S_\ell} r_{\ell,t,i}$. (ii) For each $i \in S_\ell(\tau, m)$, there are at least $m$ tasks with $r_{\ell,t,i} \geq \tau$, so $\sum_{t \in \mathcal{T}} r_{\ell,t,i} \geq m\tau$. Summing over $i \in S_\ell$ and dividing by $|\mathcal{T}|$ yields (ii). Also, since $\sum_t \sum_{i \in S_\ell} r_{\ell,t,i} \leq \sum_t \sum_{i=1}^{k} r_{\ell,t,i} = |\mathcal{T}|$, we must have $|S_\ell| \leq |\mathcal{T}|/(m\tau)$. (iii) Since $\sum_{i=1}^{k} r_{\ell,t,i} = 1$, the expected sum over a uniformly random size-$|S_\ell|$ subset is $|S_\ell|/k$. $\square$

**Why Group-1 diagnostics alone do not resolve causality.** Group 1 explains why large decode-time projection energy in $Q_{\mathrm{shared}}$ is *expected*: anisotropy implies intrinsically high-energy subspaces (Theorem A.2), and our "shareness" selection criterion further focuses the analysis on directions with measurable overlap with the decode-time shared basis, enabling a cleaner test of shared-subspace interference (Lemma A.3). However, this diagnosis alone does not resolve causality: a skeptic can still argue that shared ablation hurts more simply because it removes more energy. Group 2 therefore uses energy-matched controls to isolate causal effects from energy removal.

### A.2. Experiment Group 2: Energy-matched causal controls under strict decode-only interventions

**Motivation.** A key empirical fact from Group 1 is that the decode-time *shared* subspace is *intrinsically high-energy*: its projection captures a disproportionate share of the activation norm (larger $\mathbb{E}[\|Q^\top h\|_2^2]$). This is informative, but not yet causal: a high-energy subspace can *mechanically* amplify any decode-only intervention, so an apparent decision-level effect could arise simply because we removed "more signal," not because we removed the *right* signal.

Concretely, under our strict decode-only removal operator (Eq. (2)), with $\delta(h; Q, \alpha) := \tilde{h} - h = -\alpha QQ^\top h$, the perturbation energy satisfies

$$\|\delta(h; Q, \alpha)\|_2^2 = \alpha^2 \|QQ^\top h\|_2^2 = \alpha^2 h^\top (QQ^\top)^\top (QQ^\top)h.$$

Since $Q^\top Q = I_k$, we have $(QQ^\top)^\top = QQ^\top$ and $(QQ^\top)^2 = QQ^\top$, so

$$\|QQ^\top h\|_2^2 = h^\top QQ^\top h = (Q^\top h)^\top (Q^\top h) = \|Q^\top h\|_2^2.$$

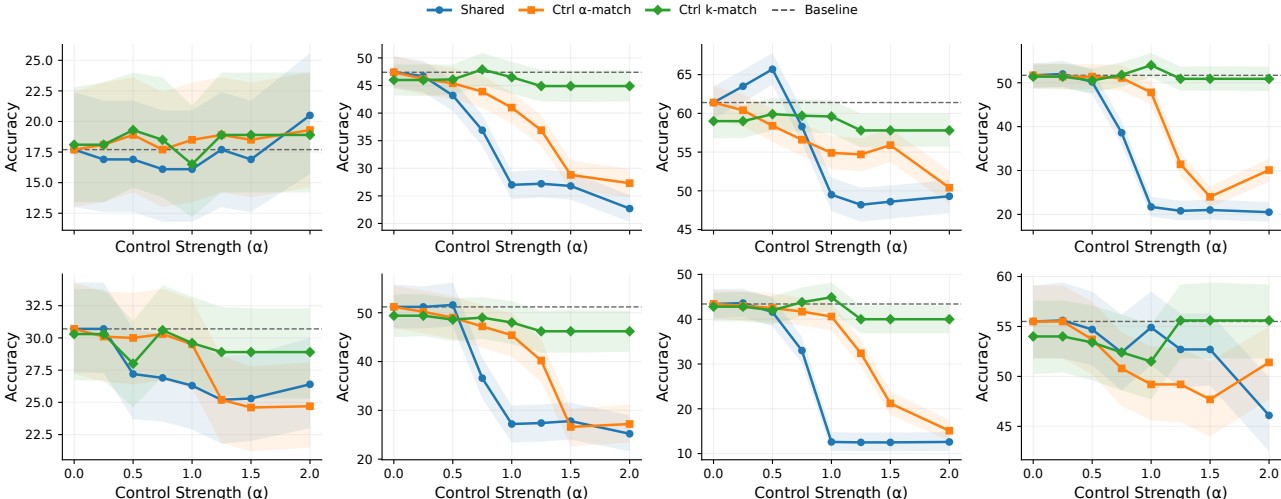

*Figure 10.* **Per-task $\alpha$ sweep (decode-only; energy-matched controls).** Panels are ordered left-to-right, top-to-bottom as: AQuA, ARC-Challenge, BoolQ, CommonsenseQA, LogiQA, OpenBookQA, QASC, StrategyQA. Dashed line: baseline; shaded regions: 95% CIs.

Therefore,

$$\|\delta(h; Q, \alpha)\|_2^2 = \alpha^2 \|Q^\top h\|_2^2.$$

In particular, even when the dimensionality $k_Q$ is fixed, a higher-energy subspace (larger $\mathbb{E}[\|Q^\top h\|_2^2]$) induces a larger expected perturbation at the same $\alpha$. This creates a natural skeptic's alternative to H2: the shared ablation could look uniquely harmful *only because* it causes a larger-energy perturbation.

To test whether decision degradation can be explained by energy alone, we introduce *energy-matched causal controls*: controls chosen (and/or scaled) so that the expected perturbation energy matches that of the shared ablation. If shared ablations remain more harmful under this match, the residual gap must be attributed to *which directions* are removed (direction-specific causal structure), rather than *how much energy* is removed.

> *Question: After matching the expected perturbation energy, do shared ablations remain uniquely harmful?*

If performance degradation is driven primarily by perturbation magnitude, then after matching $\mathbb{E}[\|\delta(h; Q, \alpha)\|_2^2]$ between the shared *removal subspace* and an energy-matched *control subspace*, the corresponding performance curves should be comparable as a function of $\alpha$ (up to noise). In particular, at a fixed layer and intervention locus, energy-matched controls should reproduce any sharp drop that would otherwise be attributed to "shareness."

**Setup.** We prefill once to build the KV cache, and intervene *only* on cached decode forwards (sequence length = 1), leaving the prefill pass untouched. Evaluation uses our decision-level scorer (forced-choice; see main text): we advance the cache under teacher forcing and accumulate conditional log-probabilities, avoiding confounds from decode-time output instability (formatting/EOS, etc.). As a sanity check, we log hook calls to confirm that prefill forwards (seq len > 1) are never modified.

**Energy matching target.** All energy matching is computed on the *same decode-last calibration states* used in the causal pipeline, so the matched quantity corresponds to the actual intervention locus. Specifically, for a basis $Q$ we define its mean projection energy on calibration states as

$$\mathcal{E}(Q) \triangleq \mathbb{E}\big[\|Q^\top h\|_2^2\big],$$

so the mean perturbation energy at strength $\alpha$ is $\alpha^2 \mathcal{E}(Q)$. We then compare $Q_S$, set of basis that span the shared subspace, against two complementary controls that match this quantity. We construct controls from a *non-shared pool of basis* $Q_{\mathrm{pool}}$ (directions explicitly outside the identified shared set). Intuitively, these are nearby directions in the same representation/activation space but *not* labeled shared by our criterion. We then match the perturbation energy in two different ways:

*Table 9.* Forced-choice accuracy (%). Each cell reports **Shared / Ctrl-$\alpha$-match / Ctrl-$k$-match**. ($k$ for $k$-match is shown in the column header; $k$-match uses $\alpha=1$.)

| Task | $N$ | Base | $\alpha$ (and $k$ for $k$-match) | | | |
|---|---|---|---|---|---|---|
| | | | 0 ($k$=126) | 0.25 ($k$=126) | 0.5 ($k$=197) | 0.75 ($k$=649) |
| commonsenseqa | 1221 | 51.7 | 51.7 / 51.7 / 51.4 | 52.0 / 51.4 / 51.4 | 50.2 / 51.4 / 50.5 | 38.6 / 51.1 / 51.8 |
| strategyqa | 687 | 55.5 | 55.5 / 55.5 / 54.0 | 55.6 / 55.5 / 54.0 | 54.7 / 53.7 / 53.4 | 52.4 / 50.8 / 52.4 |
| aqua | 254 | 17.7 | 17.7 / 17.7 / 18.1 | 16.9 / 18.1 / 18.1 | 16.9 / 18.9 / 19.3 | 16.1 / 17.7 / 18.5 |
| arc_challenge | 1172 | 47.4 | 47.4 / 47.4 / 46.0 | 46.7 / 46.2 / 46.0 | 43.2 / 45.4 / 46.1 | 36.9 / 43.9 / 47.9 |
| openbookqa | 500 | 51.2 | 51.2 / 51.2 / 49.4 | 51.2 / 50.2 / 49.4 | 51.6 / 49.0 / 48.6 | 36.6 / 47.2 / 49.0 |
| qasc | 926 | 43.4 | 43.4 / 43.4 / 42.8 | 43.6 / 43.0 / 42.8 | 41.6 / 42.5 / 42.0 | 33.0 / 41.7 / 43.8 |
| logiqa | 651 | 30.7 | 30.7 / 30.7 / 30.3 | 30.7 / 30.1 / 30.3 | 27.2 / 30.0 / 28.0 | 26.9 / 30.3 / 30.6 |
| boolq | 2048 | 61.4 | 61.4 / 61.4 / 59.0 | 63.5 / 60.4 / 59.0 | 65.7 / 58.4 / 59.9 | 58.3 / 56.6 / 59.7 |

| Task | $N$ | Base | $\alpha$ (and $k$ for $k$-match) | | | |
|---|---|---|---|---|---|---|
| | | | 1.0 ($k$=1843) | 1.25 ($k$=2705) | 1.5 ($k$=2705) | 2.0 ($k$=2705) |
| commonsenseqa | 1221 | 51.7 | 21.7 / 47.8 / 54.0 | 20.8 / 31.4 / 50.9 | 21.0 / 24.0 / 50.9 | 20.5 / 30.1 / 50.9 |
| strategyqa | 687 | 55.5 | 54.9 / 49.2 / 51.5 | 52.7 / 49.2 / 55.6 | 52.7 / 47.7 / 55.6 | 46.1 / 51.4 / 55.6 |
| aqua | 254 | 17.7 | 16.1 / 18.5 / 16.5 | 17.7 / 18.9 / 18.9 | 16.9 / 18.5 / 18.9 | 20.5 / 19.3 / 18.9 |
| arc_challenge | 1172 | 47.4 | 27.0 / 41.0 / 46.5 | 27.2 / 36.9 / 44.9 | 26.8 / 28.8 / 44.9 | 22.7 / 27.3 / 44.9 |
| openbookqa | 500 | 51.2 | 27.2 / 45.4 / 48.0 | 27.4 / 40.2 / 46.2 | 27.8 / 26.6 / 46.2 | 25.2 / 27.2 / 46.2 |
| qasc | 926 | 43.4 | 12.6 / 40.6 / 44.9 | 12.5 / 32.4 / 40.0 | 12.5 / 21.2 / 40.0 | 12.6 / 15.1 / 40.0 |
| logiqa | 651 | 30.7 | 26.3 / 29.5 / 29.6 | 25.2 / 25.2 / 28.9 | 25.3 / 24.6 / 28.9 | 26.4 / 24.7 / 28.9 |
| boolq | 2048 | 61.4 | 49.5 / 54.9 / 59.6 | 48.2 / 54.7 / 57.8 | 48.6 / 55.9 / 57.8 | 49.3 / 50.4 / 57.8 |

- **$\alpha$-match (Control-1, fixed dimension):** keep dimension fixed (typically $k_C = k_S$), but rescale the control strength $\alpha_C$ so that the *mean perturbation energy* matches:

$$\alpha_C^2 \mathbb{E}[\|Q_C^\top h\|_2^2] \;=\; \alpha_S^2 \mathbb{E}[\|Q_S^\top h\|_2^2] \quad \Rightarrow \quad \alpha_C \;=\; \alpha_S \sqrt{\frac{\mathbb{E}[\|Q_S^\top h\|_2^2]}{\mathbb{E}[\|Q_C^\top h\|_2^2]}}.$$

We sweep the shared strength $\alpha_S$ and apply the corresponding $\alpha$-match scaling. This directly answers "are we just perturbing more?" while holding $k$ fixed.

- **$k$-match (Control-2, $\alpha = 1$ pure-removal reference):** to avoid $\alpha_C > 1$ (over-subtraction), we also match energy with *unit strength* by expanding control dimension. For each swept shared strength $\alpha_S$, we set $\alpha_C = 1$ and choose the smallest $k_C$ such that the control's mean projection energy matches that of the shared subspace:

$$Q_C := Q_{\text{pool}}[:, 1:k_C] \quad \text{with} \quad \alpha_S^2 \mathbb{E}[\|Q_S^\top h\|_2^2] \approx \mathbb{E}[\|Q_C^\top h\|_2^2],$$

and then intervene with $\alpha = 1$. In practice this may require large $k_C$ (sometimes thousands of directions), which makes this control conservative: it removes *comparable energy* from a much larger set of explicitly non-shared directions.

**Results: shared directions remain uniquely causal beyond energy.** Figure 9 and Table 9 show a consistent signature: for small strengths, all methods remain close to baseline, but once $\alpha$ approaches the regime where shared removal excises a large fraction of the shared component, shared performance drops sharply while energy-matched controls stay substantially closer to baseline. This rejects the energy-only explanation in the strongest form: *even when the expected perturbation energy is matched on the same decode-last states, removing shared directions is systematically more damaging than removing non-shared directions.*

The separation is visible both in the task-aggregate curve (Fig. 9a) and per-task snapshots at $\alpha = 1$ (Fig. 9b). Moreover, the per-task sweep (Fig. 10) highlights that the effect is direction-specific and task-dependent rather than a generic brittleness of decode-only editing: the largest shared-only collapses occur on knowledge- and reasoning-heavy benchmarks (QASC/CSQA/OBQA/ARC-Challenge), while several tasks remain comparatively stable under matched controls.

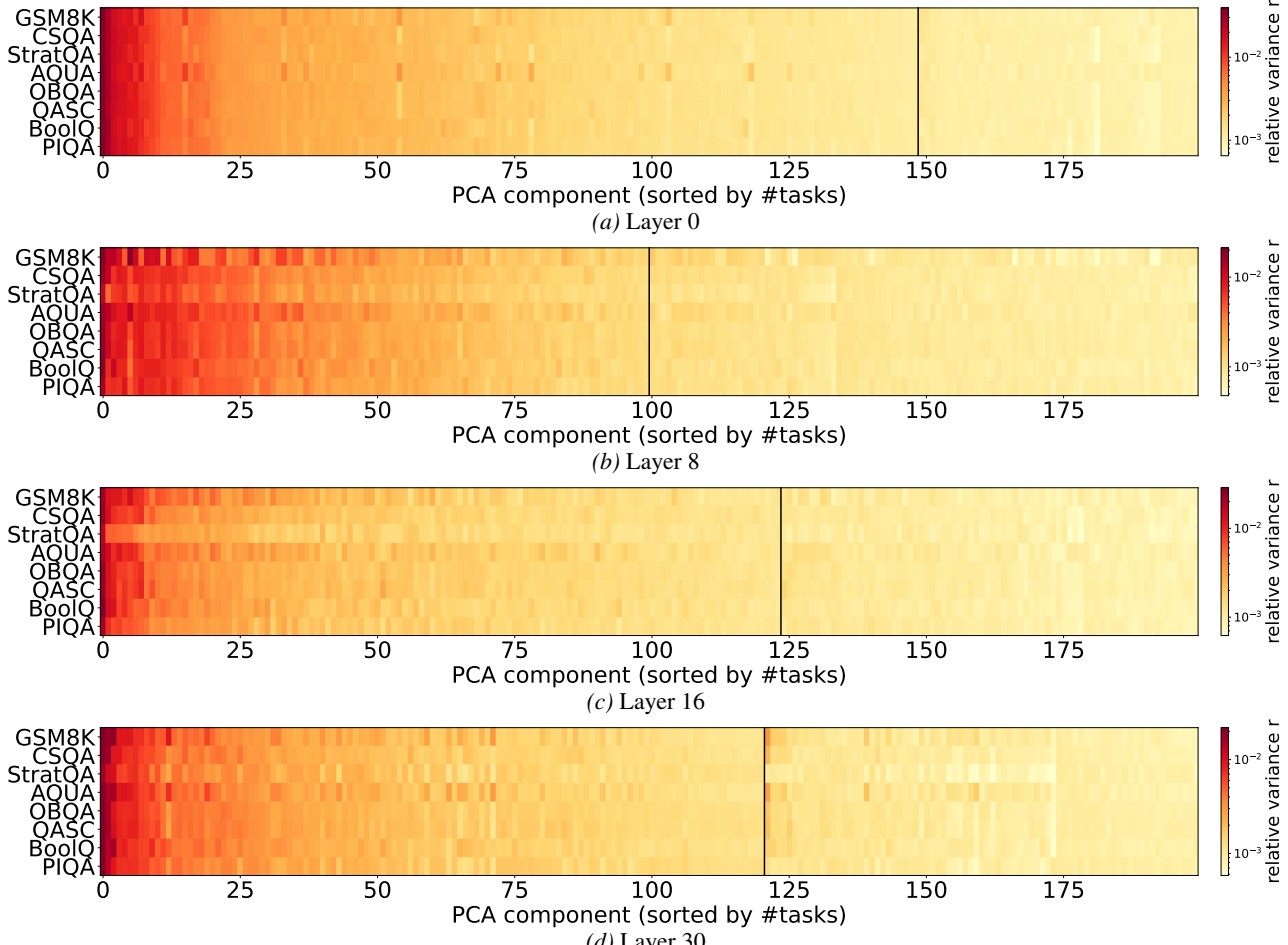

*Figure 11.* **shareness heatmaps across layers.** Rows are tasks and columns are pooled PCA components. Color shows $r_{\ell,t,i}$; columns are sorted by #tasks with $r_{\ell,t,i} \geq \tau$; vertical line marks $S_\ell(\tau, m)$.

## B. Additional Main Results.

### B.1. Additional Results for H1: Diagnostics for Shared Structure

This appendix provides additional diagnostics that complement the main H1 existence test. Throughout, we use the same notations as in §2.3: a *shared set* $S_\ell(\tau, m)$ is defined in the pooled PCA coordinate system by thresholding per-task relative variance contributions, and the *shared core size* is $T_\ell = |S_\ell(\tau, m)|$. These analyses do not repeat the null-test procedure (reported in the main paper); instead, they characterize (i) how sharing varies across coarse task groupings and (ii) how pooled subspaces behave as we vary the task pool and the inference phase.

**B.1.1 Within-category versus mixed-category sharing.** To probe whether sharing is merely a byproduct of coarse dataset categories, we compare *within-category* task pairs to a *mixed-category* baseline formed by cross-category pairs. For each pair, we compute the *shared ratio* $\text{SR} = |S_\ell(\tau, m)|/k$, where $k$ is the pooled PCA dimensionality selected by the variance retention target (default $\rho = 0.95$) and we use the same sharedness threshold as in the main paper (default $\tau = 10^{-3}$). Figure 13a summarizes the results and Table 13 reports the corresponding numbers. The math pair (GSM8K+AQuA) exhibits noticeably higher within-pair sharing than the mixed baseline, suggesting stronger reuse within mathematical reasoning. By contrast, commonsense, reasoning, and science show within-category sharing that is similar to (or slightly below) the mixed baseline, indicating that coarse category labels do not uniformly imply higher shareness under our metric. Overall, this diagnostic suggests that the decode-time shared core identified by DecodeShare is not simply a trivial artifact of coarse category membership.

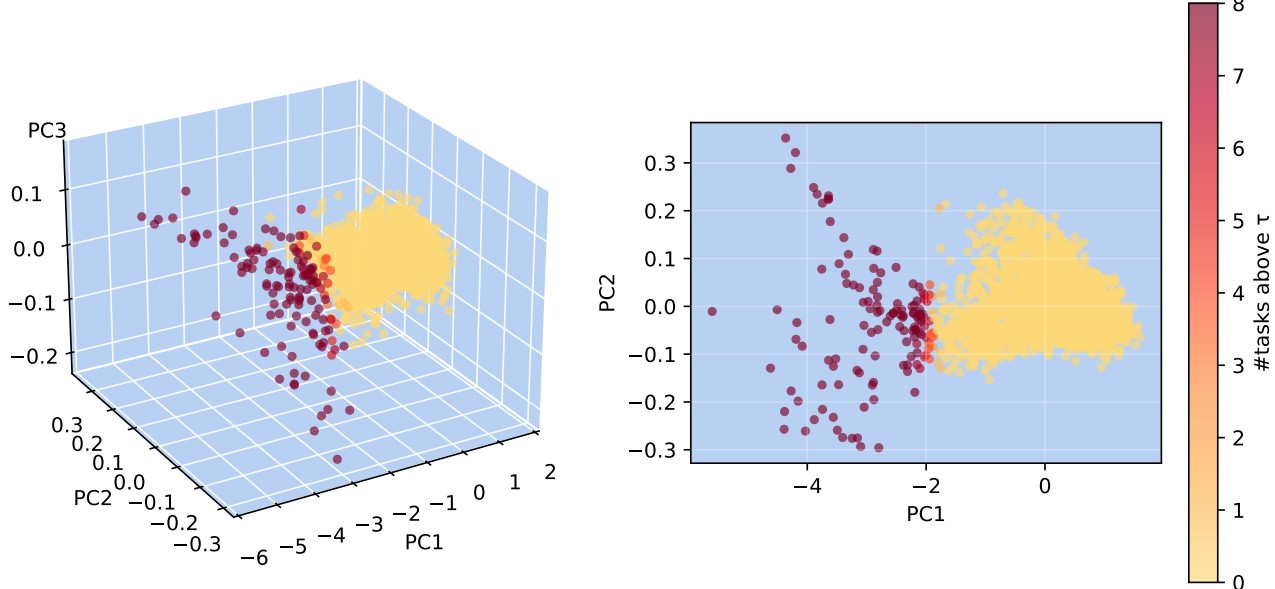

*Figure 12.* **PCA visualization of component signatures for layer 10.** *Left:* 3D projection onto the first three principal components (PC1–PC3). *Right:* top-down projection onto PC1–PC2. Each point corresponds to a component; color indicates the number of tasks whose component score exceeds the threshold $\tau$ (colorbar).

*Table 10.* H1 summary: sensitivity to $\tau$ (left) and existence test across models (right).

*(a)* Sensitivity of $|S_\ell(\tau, m)|$ to PCA retention $\rho$ (thus $k$) and threshold $\tau$. Entries show $|S|$ and $\%(|S|/k)$.

| PCA | | Shared set $|S_\ell(\tau, m)|$: $|S|$ (% of $k$) | | | |
|---|---|---|---|---|---|
| $\rho$ | $k$ | $\tau=10^{-4}$ | $\tau=10^{-3}$ | $\tau=10^{-2}$ | $\tau k$ at $10^{-3}$ |
| 0.80 | 1490 | 1490 (100.0%) | 144 (9.7%) | 1 (0.07%) | 1.49 |
| 0.90 | 2340 | 2169 (92.7%) | 123 (5.3%) | 1 (0.04%) | 2.34 |
| 0.95 | 2989 | 2353 (78.7%) | 109 (3.6%) | 1 (0.03%) | 2.99 |
| 0.97 | 3332 | 2413 (72.4%) | 100 (3.0%) | 1 (0.03%) | 3.33 |
| 0.99 | 3769 | 2484 (65.9%) | 89 (2.4%) | 1 (0.03%) | 3.77 |

*(b)* **Sharedness existence test** (layer $\ell{=}10$, $\tau{=}10^{-3}$; see Methodology). $k$: pooled cross-task PCA dim (95% var, capped by $d$). Shared: #/% of $d$. $p_{\text{perm}}$, $p_{\text{scr}}$: permutation/scramble tests (see Methodology).

| Model | $k$ | $d$ | Shared (# / %$d$) | $p_{\text{perm}}$ | $p_{\text{scr}}$ |
|---|---|---|---|---|---|
| Qwen2.5-7B-Instruct | 2614 | 3584 | 73 / 2.04 | $< 10^{-3}$ *** | 0.0196* |
| Llama-2-7B-Chat | 2957 | 4096 | 107 / 2.61 | $< 10^{-3}$ *** | 0.0196* |
| Llama-3.1-8B-Instruct | 2809 | 4096 | 112 / 2.73 | $< 10^{-3}$ *** | 0.0196* |
| Falcon-7b-instruct | 2702 | 4544 | 124 / 2.73 | $< 10^{-3}$ *** | 0.0196* |

**B.1.2. Pooled subspace convergence as we add tasks.** Next, we study whether the *pooled PCA subspace* used to define shared directions is stable as the task pool grows. Fix a layer $\ell$ and retention target (default $\rho = 0.95$). For each $n \in \{2, \ldots, T\}$, we sample $n$ tasks, estimate a pooled PCA basis $Q_n$, and compare it to the full-task reference basis $Q_T$. To quantify similarity between two orthonormal bases $Q_a \in \mathbb{R}^{d \times k_a}$ and $Q_b \in \mathbb{R}^{d \times k_b}$, we use a normalized subspace overlap score

$$\text{Overlap}(Q_a, Q_b) \triangleq \frac{\|Q_a^\top Q_b\|_F^2}{\min(k_a, k_b)} \in [0, 1],$$

which equals the mean squared cosine of principal angles (1 indicates identical subspaces). Table 14 and Fig. 13b show that the pooled subspace converges quickly: even with a small number of tasks, the overlap to the full-task reference is already high and increases monotonically as tasks are added. Meanwhile the pooled dimension $k$ (`cross_dim` in the table) grows with $n$, consistent with additional tasks contributing extra directions needed to maintain the same explained-variance target. This supports a practical point: DecodeShare's pooled coordinate system is not fragile to the exact composition of the task pool, even though the *shared set* itself depends on the sharedness threshold and the "nontrivial for many tasks" criterion.

**B.1.3. Phase-distinguished pooled geometry and the fully-shared core.** Finally, we compare pooled subspaces estimated from different inference phases: `decode-last` (KV-cached, seq len $= 1$), `prefill-last` (full-sequence prefill), and `decode-step-`$t$ (a fixed decode step). Table 15 shows that *all* phases yield stable pooled subspaces under the overlap diagnostic, but with markedly different pooled dimensions: `decode-last` produces a much larger pooled subspace (thousands of dimensions at $\rho = 0.95$), whereas `prefill-last` produces a much smaller pooled subspace (hundreds of

---

**Algorithm 1** Decode-aligned shared basis estimation at layer $\ell$ (DecodeShare)

---

**Require:** tasks $\mathcal{T}$, calibration prompts $\{\mathcal{C}_t\}$, decoding policy $\pi$, horizon $K$, PCA ratio $\rho$, threshold $(\tau, m)$
**Ensure:** joint basis $Q_\ell$ and shared basis $Q_\ell^{(S)}$

1. For each $t \in \mathcal{T}$, run cached decoding on prompts in $\mathcal{C}_t$ under policy $\pi$ up to $K$ steps.

2. Collect decode last-token states $\{h_\ell^{(s)}\}$ where seq_len $= 1$ into $X_{t,\ell} \in \mathbb{R}^{n_{t,\ell} \times d}$.

3. Task-center: $\tilde{X}_{t,\ell} \leftarrow X_{t,\ell} - \mathbf{1}\mu_{t,\ell}^\top$.

4. Balance to $n_\ell = \min_t n_{t,\ell}$ by subsampling each $\tilde{X}_{t,\ell}$.

5. Pool: $\tilde{X}_\ell \leftarrow \text{concat}(\tilde{X}_{t,\ell} : t \in \mathcal{T}) \in \mathbb{R}^{(|\mathcal{T}| n_\ell) \times d}$.

6. PCA/SVD: $\tilde{X}_\ell = U\Sigma V^\top$; choose smallest $k$ s.t. explained variance $\geq \rho$; set $Q_\ell \leftarrow V_{:,1:k}$.

7. For each $t$, project $Z_{t,\ell} \leftarrow \tilde{X}_{t,\ell}Q_\ell$, compute $r_{t,i} \leftarrow \text{Var}(Z_{t,\ell}[:, i]) / \sum_j \text{Var}(Z_{t,\ell}[:, j])$.

8. Shared set: $S_\ell(\tau, m) \leftarrow \{i : |\{t : r_{t,i} \geq \tau\}| \geq m\}$.

9. Return $Q_\ell$ and $Q_\ell^{(S)} \leftarrow Q_\ell[:, S_\ell]$.

---

**Algorithm 2** Decode-time last-token subspace removal

---

**Require:** basis $Q_\ell$ (shared or control), strength $\alpha$, stage length $K_{\text{stage}}$ (optional)

1. For decode step $s = 1, 2, \ldots$ : run cached forward pass with seq_len $= 1$ to obtain $h_\ell^{(s)}$.

2. If full removal: $\tilde{h}_\ell^{(s)} \leftarrow h_\ell^{(s)} - \alpha Q_\ell Q_\ell^\top h_\ell^{(s)}$.

3. If staged: apply the same update only when the generated-token count is $< K_{\text{stage}}$.

4. Continue to logits; choose next token (greedy/sampling); update cache.

---

dimensions). This highlights a key nuance: *subspace stability alone is not decode-specific*—prefill can also yield a stable low-rank pooled geometry.

To connect stability to shareness, we evaluate the *fully-shared core size* within each estimated basis by applying the same shareness criterion at the all-task setting $m = |T|$ (using $\tau = 10^{-3}$) and counting how many pooled directions are nontrivial for *every* task. Table 15 shows that the *absolute* fully-shared core size remains on the order of $10^2$ directions across phases and across $n$, while the *shared ratio* can vary widely because the denominator $k$ differs substantially across phases (and typically increases with $n$). In particular, `decode-last` yields a modest shared core in absolute size but a smaller shared ratio due to a much larger pooled dimension, whereas `prefill-last` yields a higher shared ratio largely because its pooled dimension is much smaller. These diagnostics sharpen the main paper's message: recovering a stable pooled geometry is not sufficient to identify a decision-time shared channel, and phase alignment matters for whether an estimated basis transfers to decode-time causal tests (see H3).

**B.1.4. All-tasks vs. majority-shared (pooled) shareness.** We distinguish two *thresholding regimes* for the same shared-set definition computed in a *single, task-aligned* coordinate system. Concretely, we first obtain a shared basis $Q_\ell$ via pooled PCA (to remove the permutation/rotation non-identifiability of per-task PCA components), and then define the shared set

$$S_\ell(\tau, m) = \left\{i : \left|\{t \in \mathcal{T} : r_{\ell,t,i} \geq \tau\}\right| \geq m\right\},$$

where $r_{\ell,t,i}$ is the task-$t$ relative-variance score of direction $i$ at layer $\ell$. The **All-tasks** setting ($m = |\mathcal{T}|$) is the strict *intersection* criterion: a direction is declared shared only if it exceeds $\tau$ for *every* task. The **majority-shared** setting (our default; e.g., $m = 8$ out of $|\mathcal{T}| = 13$) operationalizes the claim as a *compact core shared by many tasks* rather than

*Table 11.* Sharedness test at layer $\ell = 10$ with $\tau = 10^{-3}$ and 8-task $m_{\text{shared}} = 8$. $k$ is the cross-task PCA dimension (95% variance), $d$ is the hidden dimension. Permutation null uses $B_{\text{perm}} = 2000$ and scramble null uses $B_{\text{scr}} = 200$; p-values at the floor are reported as upper bounds.

| Model | $k$ | $d$ | Shared (# / %$d$) | $p_{\text{perm}}$ | $p_{\text{scr}}$ |
|---|---|---|---|---|---|
| Qwen2.5-7B-Instruct | 2614 | 3584 | 73 / 2.04 | $\leq 5.0 \times 10^{-4}$ *** | $\leq 5.0 \times 10^{-3}$ ** |
| Llama-2-7B-Chat | 2989 | 4096 | 109 / 2.66 | $\leq 5.0 \times 10^{-4}$ *** | $\leq 5.0 \times 10^{-3}$ ** |
| Llama-3.1-8B-Instruct | 2809 | 4096 | 112 / 2.73 | $\leq 5.0 \times 10^{-4}$ *** | $\leq 5.0 \times 10^{-3}$ ** |
| Falcon-7b-instruct | 2702 | 4544 | 124 / 2.73 | $\leq 5.0 \times 10^{-4}$ *** | $\leq 5.0 \times 10^{-3}$ ** |

*Table 12.* **Full sharedness results across layers.** Same setting as Table 10b, but reporting layers $\ell \in \{4, 10, 20, 24\}$ for each model.

| Model | Layer | $d$ | $d_{\text{cross}}$ | $\frac{d_{\text{cross}}}{d}$ | $T_\ell$ | $\frac{T_\ell}{d_{\text{cross}}}$ | states | $p_{\text{perm}}$ / $p_{\text{scr}}$ |
|---|---|---|---|---|---|---|---|---|
| Falcon-7B | 4 | 4544 | 2679 | 0.590 | 125 | 0.0467 | 6023 | $5.00 \times 10^{-4}$ / $4.98 \times 10^{-3}$ |
| Falcon-7B | 10 | 4544 | 2702 | 0.595 | 124 | 0.0459 | 6023 | $5.00 \times 10^{-4}$ / $4.98 \times 10^{-3}$ |
| Falcon-7B | 20 | 4544 | 2742 | 0.603 | 111 | 0.0405 | 6023 | $5.00 \times 10^{-4}$ / $4.98 \times 10^{-3}$ |
| Falcon-7B | 24 | 4544 | 2817 | 0.620 | 109 | 0.0387 | 6023 | $5.00 \times 10^{-4}$ / $4.98 \times 10^{-3}$ |
| Llama-2-7B-chat | 4 | 4096 | 2970 | 0.725 | 118 | 0.0397 | 18560 | $5.00 \times 10^{-4}$ / $4.98 \times 10^{-3}$ |
| Llama-2-7B-chat | 10 | 4096 | 2989 | 0.730 | 109 | 0.0365 | 18560 | $5.00 \times 10^{-4}$ / $4.98 \times 10^{-3}$ |
| Llama-2-7B-chat | 20 | 4096 | 3066 | 0.749 | 140 | 0.0457 | 18560 | $5.00 \times 10^{-4}$ / $4.98 \times 10^{-3}$ |
| Llama-2-7B-chat | 24 | 4096 | 3184 | 0.777 | 138 | 0.0433 | 18560 | $5.00 \times 10^{-4}$ / $4.98 \times 10^{-3}$ |
| Llama-3.1-8B | 4 | 4096 | 2739 | 0.669 | 107 | 0.0391 | 20000 | $5.00 \times 10^{-4}$ / $4.98 \times 10^{-3}$ |
| Llama-3.1-8B | 10 | 4096 | 2809 | 0.686 | 112 | 0.0399 | 20000 | $5.00 \times 10^{-4}$ / $4.98 \times 10^{-3}$ |
| Llama-3.1-8B | 20 | 4096 | 2795 | 0.682 | 94 | 0.0336 | 20000 | $5.00 \times 10^{-4}$ / $4.98 \times 10^{-3}$ |
| Llama-3.1-8B | 24 | 4096 | 2861 | 0.698 | 98 | 0.0343 | 20000 | $5.00 \times 10^{-4}$ / $4.98 \times 10^{-3}$ |
| Qwen2.5-7B | 4 | 3584 | 2621 | 0.731 | 92 | 0.0351 | 20000 | $5.00 \times 10^{-4}$ / $4.98 \times 10^{-3}$ |
| Qwen2.5-7B | 10 | 3584 | 2614 | 0.729 | 73 | 0.0279 | 20000 | $5.00 \times 10^{-4}$ / $4.98 \times 10^{-3}$ |
| Qwen2.5-7B | 20 | 3584 | 2631 | 0.734 | 87 | 0.0331 | 20000 | $5.00 \times 10^{-4}$ / $4.98 \times 10^{-3}$ |
| Qwen2.5-7B | 24 | 3584 | 2678 | 0.747 | 111 | 0.0414 | 20000 | $5.00 \times 10^{-4}$ / $4.98 \times 10^{-3}$ |

by all tasks. Importantly, both settings are evaluated under the *same* two task-preserving nulls (permutation Null-1 and orthogonal-scramble+re-estimation Null-2), so switching from $m = |\mathcal{T}|$ to $m < |\mathcal{T}|$ does not relax statistical control: a shared set must still lie in the tail of *both* null distributions, and Null-2 in particular is designed to reject spuriously inflated shareness that can arise from coordinate artifacts or over-permissive thresholds.

**Why cross all-tasks shared subspace can be empty without contradicting H1.** All-tasks shareness is a *lower bound* on cross-task sharing: by construction $S_\ell(\tau, |\mathcal{T}|) \subseteq S_\ell(\tau, m)$ for any $m < |\mathcal{T}|$. With heterogeneous tasks (here, 13 diverse benchmarks), the all-tasks intersection can become empty even when there exists a genuine shared mechanism, because it is extremely sensitive to (i) task-specific idiosyncrasies, (ii) finite-sample variability in $r_{\ell,t,i}$, and (iii) any single "outlier" task that falls just below $\tau$ on an otherwise consistently-used direction. Thus, observing a non-empty and significant majority-shared core alongside an empty all-tasks intersection should be interpreted as: there exists a statistically validated shared decision subspace used by *most* tasks, but not necessarily in a uniform-above-threshold way for *every* task. This is exactly the regime our claim targets: a *compact, decode-time shared workspace* that is broadly reused across tasks, while allowing that some tasks may rely on alternative or additional directions. Finally, we guard against the concern that pooling could induce task leakage by using the LOTO control (fit $Q_\ell$ excluding a held-out task and then evaluate shareness on that held-out task), ensuring the reported majority-shared core reflects genuine cross-task structure rather than pooled overfitting.

**B.1.5. Loosened threshold sensitivity (four-model subset).** We further test a permissive threshold setting (loosened: $\tau = 3 \times 10^{-4}$, $m = 8$) on the same 13-task pool to probe the predicted degeneracy regime. For Llama-3.1-8B and Falcon-7B, the shared set expands substantially (e.g., $|S|/\text{cross} = 0.238$ and $0.214$ respectively), but fails the stronger orthogonal-scramble null (Null-2; $p_2 = 1.0$), indicating that the inflated shared size is not distinguishable from scrambled structure. For Qwen2.5-7B and Llama-2-7B, the loosened setting remains significant under both nulls ($p_1 < 0.05$, $p_2 < 0.05$) yet yields very large shared ratios ($|S|/\text{cross} = 0.456$ and $0.573$), suggesting that overly permissive thresholds can recover a broad low-rank decision subspace rather than a compact shared core. Overall, these results are consistent with the $\tau$-stringency interpretation: when $\tau$ becomes too small relative to the diffuse-usage scale $1/k$, many low-concentration directions are labeled nontrivial, producing a large and less discriminative shared set; therefore we use $\tau = 10^{-3}$ as the conservative default and treat loosened primarily as a sensitivity/degeneracy check.

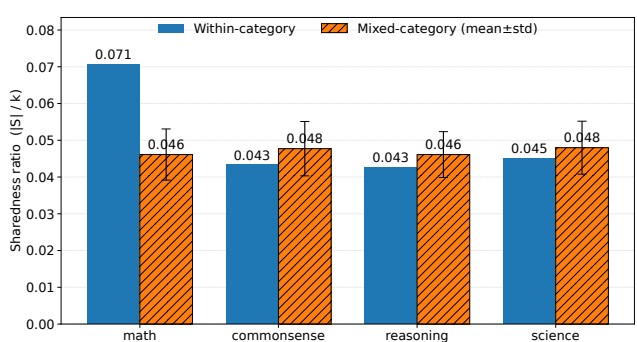

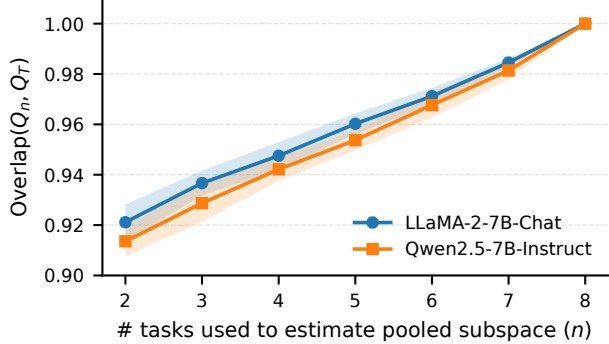

*(a)* Within-category shareness compared to a mixed-category baseline (mean±std) across four coarse categories.

*(b)* Subspace overlap to the full-task pooled subspace as a function of the number of tasks used to estimate the pooled PCA basis.

*Figure 13.* (Left) Within-category shareness vs. mixed-category baseline. (Right) Overlap to the full-task pooled subspace vs. number of tasks.

*Table 13.* Within/mixed-category diagnostic for H1. SR denotes shared ratio $|S_\ell(\tau, m)|/k$ under the default settings.

| Category | Within tasks | $k$ | Within SR | Mixed SR ($\mu \pm \sigma$) | Within / Mixed count |
|---|---|---|---|---|---|
| math | GSM8K, AQuA | 2221 | 0.0707 | 0.0461±0.0070 | 157 / 123.1±14.0 |
| commonsense | CSQA, PIQA | 2723 | 0.0433 | 0.0477±0.0074 | 118 / 126.8±14.8 |
| reasoning | StratQA, BoolQ | 2815 | 0.0426 | 0.0461±0.0062 | 120 / 123.5±12.7 |
| science | OBQA, QASC | 2600 | 0.0450 | 0.0480±0.0072 | 117 / 126.7±14.7 |

## B.2. Additional Results for **H2**: Experiments for More Models.

**B.2.1 Patchback results across models and layers.** Tables 1,18, and 19 summarize patchback behavior across three model families (Llama, Falcon, Qwen) and three representative depths (layers 4/10/24). Throughout, rescue rates are computed on the corresponding flip set $\mathcal{F}$ (examples that flip under the specified decode-only ablation), and aggregated by summing rescued/total across tasks (and across metrics where applicable).

**A consistent qualitative pattern: patching along $S_\ell$ is highly effective; nonshared controls are negligible.** Across models and layers, patching along the identified shared decode subspace $S_\ell$ is consistently effective. In multiple-choice (Table 1), the full patch (Patched@full) rescues essentially all flips, while patching a nonshared direction is exactly zero across all settings (NonsharedPatch). In open-answer (Table 18), NonsharedPatch remains small (typically ∼1–7%) and far below Patched(self), arguing against a naive "any patch works" explanation and supporting the view that a small shared decode pathway is a high-leverage causal route for last-token decision-making.

**Model- and layer-dependent separation from random controls.** While targeted patching is consistently strong, the strength of random controls varies by model and depth. In Llama, random shared-vector and random-subspace controls are comparatively small, yielding a clean mechanistic gap. In contrast, Qwen exhibits unusually strong random-subspace baselines at multiple layers, indicating that generic low-rank perturbations can often intersect decision-relevant directions. This suggests shared-subspace interference is a primary contributor to failures, but some models/layers also exhibit broader low-rank decision structure that makes random controls stronger.

**Open-answer: strong time-shuffle indicates temporal stationarity of the shared decode signal.** In open-answer patchback (Table 18), the within-example **Time-shuffle** intervention can approach Patched(self.) This is expected under our setup: the shared decode subspace $S_\ell$ is estimated from the decode-time activation distribution, so donor components drawn from other decode steps of the same example remain on-manifold for $S_\ell$. The competitiveness of Time-shuffle therefore suggests that the decision-relevant signal carried by $S_\ell$ is approximately *stationary across decode steps* (i.e., not tied to a single precise time index). Importantly, this does not collapse specificity: patching directions outside $S_\ell$ (NonsharedPatch) remains small, and random low-rank controls remain substantially below targeted patching in most settings. Nevertheless, Patched(self) remains the strongest intervention overall, and NonsharedPatch stays comparatively small, preserving a clear

*Table 14.* Pooled-subspace convergence diagnostic (decode-time states). We report mean±std over repeats.

| | LLaMA | | Qwen | |
|---|---|---|---|---|
| $n$ tasks | Overlap ↑ | cross_dim | Overlap ↑ | cross_dim |
| 2 | 0.921±0.007 | 2636.6±136.8 | 0.914±0.006 | 2258.0±80.1 |
| 3 | 0.937±0.005 | 2788.9±83.4 | 0.929±0.007 | 2419.5±48.0 |
| 4 | 0.948±0.005 | 2885.1±56.4 | 0.942±0.005 | 2490.1±38.8 |
| 5 | 0.960±0.004 | 2904.3±59.4 | 0.954±0.004 | 2521.1±38.5 |
| 6 | 0.971±0.003 | 2951.5±33.9 | 0.968±0.005 | 2569.1±24.0 |
| 7 | 0.985±0.002 | 2967.2±24.0 | 0.981±0.004 | 2592.4±17.0 |
| 8 | 1.000±0.000 | 2989.0±0.0 | 1.000±0.000 | 2614.0±0.0 |

*Table 15.* Phase-distinguished pooled-subspace convergence (LLaMA). Mean±std over repeats.

| | $n$ tasks | | | | | | |
|---|---|---|---|---|---|---|---|
| Metric / Mode | 2 | 3 | 4 | 5 | 6 | 7 | 8 |
| Overlap ↑ (decode-last) | $0.920 \pm 0.006$ | $0.933 \pm 0.005$ | $0.946 \pm 0.004$ | $0.958 \pm 0.004$ | $0.970 \pm 0.001$ | $0.984 \pm 0.002$ | $1.000 \pm 0.000$ |
| cross_dim (decode-last) | $2571.0 \pm 87.0$ | $2745.7 \pm 52.2$ | $2818.9 \pm 69.3$ | $2869.9 \pm 54.3$ | $2890.2 \pm 33.7$ | $2942.9 \pm 11.6$ | $2961.0 \pm 0.0$ |
| Overlap ↑ (prefill-last) | $0.884 \pm 0.039$ | $0.896 \pm 0.029$ | $0.910 \pm 0.026$ | $0.922 \pm 0.023$ | $0.948 \pm 0.012$ | $0.959 \pm 0.016$ | $1.000 \pm 0.000$ |
| cross_dim (prefill-last) | $160.4 \pm 3.9$ | $223.5 \pm 3.8$ | $278.1 \pm 6.1$ | $327.8 \pm 6.6$ | $377.5 \pm 5.7$ | $415.7 \pm 4.7$ | $456.0 \pm 0.0$ |
| Overlap ↑ (decode-step-$t$) | $0.946 \pm 0.011$ | $0.946 \pm 0.007$ | $0.948 \pm 0.007$ | $0.954 \pm 0.004$ | $0.961 \pm 0.004$ | $0.976 \pm 0.004$ | $1.000 \pm 0.000$ |
| cross_dim (decode-step-$t$) | $181.2 \pm 14.7$ | $271.8 \pm 15.6$ | $345.6 \pm 17.0$ | $426.2 \pm 9.0$ | $498.2 \pm 13.4$ | $577.0 \pm 8.8$ | $644.0 \pm 0.0$ |

separation between targeted repair and naive patching.

**Transfer patching can be depth-sensitive.** Flipset transfer (Table 19) reveals that robustness does not necessarily transfer uniformly across layers. For example, Qwen maintains high transfer rescue at layer10 but degrades sharply at layer24, even though self patching remains at 100%. This layer sensitivity suggests that the shared decode pathway is not monolithic across depth: deeper layers may require different donor selection or richer basis estimation to support cross-task transfer reliably.

### B.3. Additional Results for `H3`: results under generation-based evaluation

**Prefill-Decode mismatch in Generation.** Table 21 reproduces the H3 mismatch signature under a generation-based protocol. We apply the same *decode-only* intervention while swapping only the estimation distribution: a shared basis estimated from decode states (Decode-shared) versus one estimated from prefill states (Prefill-shared), with dimension matched ($k = 129$) and a matched-energy random control (Random). Across tasks, removing the *decode-estimated* shared workspace causes large degradation (e.g., GSM8K collapses to 0.0%, StrategyQA drops from 43.8% to 10.5%, ARC-C from 49.6% to 19.5%, QASC from 28.9% to 10.2%), while the *prefill-estimated* basis is much weaker and often stays near baseline within confidence intervals. The random control remains close to baseline, indicating the effect is not explained by removing arbitrary energy-matched subspaces. Most differences are statistically significant (p-values in the last column), supporting that estimator–deployment mismatch persists beyond the forced-choice setting.

We note minor task-dependent exceptions (e.g., LogiQA), but they do not change the qualitative conclusion: the strongest and most consistent causal impact at the decode locus arises only from *decode-aligned* shared bases. Together with Table 7, these results reinforce the core claim that prefill-derived shared directions are generally not a substitute for decode-time interventions.

### B.4. H3: Prefill–Decode Mismatch and the Need for Decode-Aligned Estimation

**Motivation: estimator–deployment mismatch under KV-cached decoding.** Under KV-cached inference, each next-token decision is produced by a cached forward pass with sequence length = 1. Consequently, the decode-time decision states follow a distribution $D_{\text{decode}}$ that can differ from the full-sequence prefill distribution $D_{\text{prefill}}$. This creates an estimator–deployment mismatch for many activation-space pipelines that estimate directions from prefill states but intervene at decode time. DecodeShare enforces an *alignment principle*: all causal interventions act only on cached decode calls, and subspaces are estimated from the same decode-time distribution on which we intervene.

$2 \times 2$ **estimation–intervention grid.** To isolate the causal consequence of prefill–decode mismatch, we evaluate a $2 \times 2$ grid that crosses (i) the *estimation distribution* (decode-last vs. prefill-last) and (ii) the *intervention locus* (decode-only vs.

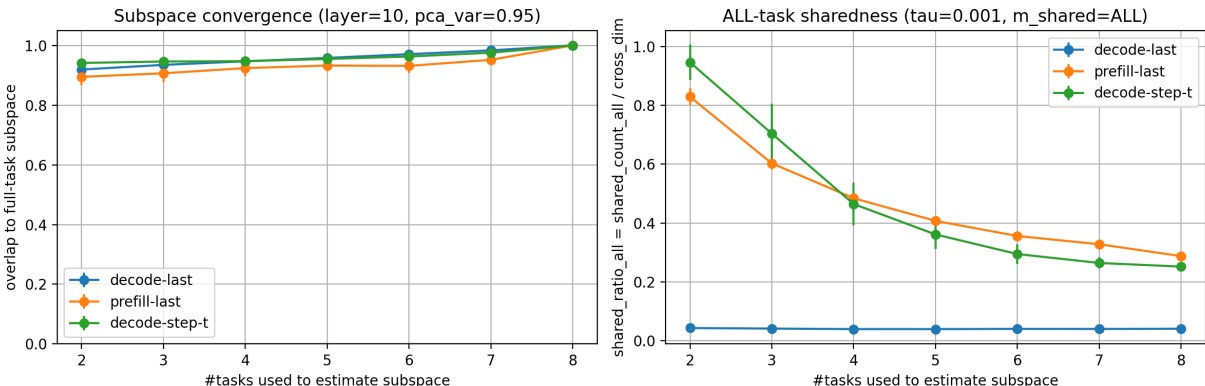

*Figure 14.* Phase-distinguished convergence trends across modes. Left: overlap to the full-task pooled subspace; right: cross dimension as the number of tasks increases.

*Table 16.* H1 shareness test on the 13-task full benchmark at layer $\ell = 10$: comparison between the strict all-tasks definition ($m = |T|$, shown as $m = 13$ or "all" depending on the run) and the majority-shared core ($m = 8$). We report cross-task pooled PCA dimension (cross), shared set size $|S|$, ratio $|S|/cross$, and null p-values $p_1$ (permutation) and $p_2$ (orthogonal scramble).

| Model | Setting | $\tau$ | $m$ | cross | $|S|$ | $|S|/cross$ | $p_1$ | $p_2$ |
|---|---|---|---|---|---|---|---|---|
| Llama-3.1-8B-Instruct | All-tasks | 0.0010 | all | 2812 | 110 | 0.039 | 0.000500 | 0.009901 |
| Llama-3.1-8B-Instruct | Pooled (majority) | 0.0010 | 8 | 2812 | 143 | 0.051 | 0.000500 | 0.009901 |
| Qwen2.5-7B-Instruct | All-tasks | 0.0010 | 13 | 2549 | 56 | 0.022 | 0.000500 | 0.009901 |
| Qwen2.5-7B-Instruct | Pooled (majority) | 0.0010 | 8 | 2549 | 85 | 0.033 | 0.000500 | 0.009901 |
| Falcon-7B-Instruct | All-tasks | 0.0010 | all | 2479 | 122 | 0.049 | 0.000500 | 0.009901 |
| Falcon-7B-Instruct | Pooled (majority) | 0.0010 | 8 | 2479 | 154 | 0.062 | 0.000500 | 0.009901 |
| Llama-2-7b-chat-hf | All-tasks | 0.0010 | 13 | 2423 | 0 | 0.000 | 1.000000 | 1.000000 |
| Llama-2-7b-chat-hf | Pooled (majority) | 0.0010 | 8 | 2423 | 35 | 0.014 | 0.000500 | 0.009901 |

prefill-only). Concretely, we estimate $Q^{(S)}_{\text{decode}}$ from decode-last states and $Q^{(S)}_{\text{prefill}}$ from prefill-last states (at the same layer, tasks, and hyperparameters), and apply projection removal $\tilde{h} = h - \alpha Q Q^\top h$ either during cached decoding (seq-len= 1) or only at prefill. Fixing the intervention locus to decode-only turns an estimator swap (decode-est vs. prefill-est) into a direct test of mismatch while keeping the intervention budget identical (Table 23). A representative small-rank setting preserves the qualitative contrast, ruling out a dimension-driven artifact (Table 22).

**Geometric mismatch.** Decode- and prefill-estimated shared subspaces are strongly misaligned even after rank matching. For example, at layer $\ell = 10$ with $k_{\text{match}} = \min(k^{\text{shared}}_{\text{decode}}, k^{\text{shared}}_{\text{prefill}})$, the principal angles between the two shared subspaces are near-orthogonal, and the decode-estimated shared dimension is substantially larger than the prefill-estimated one (Table 20). This confirms a concrete prefill–decode mismatch: a prefill-derived basis is not a faithful proxy for the decode-time workspace we causally probe.

**Causal consequence at the decode locus.** Under decode-only intervention, removing the *decode-estimated* shared basis induces large and consistent accuracy degradation across benchmarks, whereas swapping only the estimator distribution (prefill-est / decode-intervene) removes this effect and behaves comparably to matched random controls (Table 7). Interventions applied only at prefill are negligible across all conditions, ruling out explanations that operate purely through prompt-processing dynamics. Together, these results support H3: *to reproduce robust decode-time causal effects under matched budgets, estimation must be aligned to the decode-time distribution*.

**Replication under generation-based evaluation.** To verify that the mismatch signature is not an artifact of forced-choice scoring, we replicate on generation-based evaluation Table 21 and in a mixed-protocol setting Table 23. Across tasks, decode-estimated shared removal yields substantially larger degradations than prefill-estimated removal under the same decode-only intervention and matched controls, indicating that estimator–deployment mismatch persists beyond the decision-level setting.

*Table 17.* Loosened shareness setting on the same 13-task full benchmark (layer $\ell = 10$): $\tau = 3 \times 10^{-4}$ and $m = 8$. We report cross, $|S|$, $|S|/cross$, and null p-values $p_1/p_2$ using the same criteria as Table 16.

| Model | $\tau$ | $m$ | cross | $|S|$ | $|S|/cross$ | $p_1$ | $p_2$ |
|---|---|---|---|---|---|---|---|
| meta-llama/Llama-3.1-8B-Instruct | 0.0003 | 8 | 2812 | 670 | 0.238 | 0.000500 | 1.000000 |
| Qwen/Qwen2.5-7B-Instruct | 0.0003 | 8 | 2549 | 1163 | 0.456 | 0.000500 | 0.009901 |
| tiiuae/falcon-7b-instruct | 0.0003 | 8 | 2479 | 530 | 0.214 | 0.000500 | 1.000000 |
| meta-llama/Llama-2-7b-chat-hf | 0.0003 | 8 | 2423 | 1389 | 0.573 | 0.000500 | 0.009901 |

*Table 18.* **Open-answer patchback, aggregated over tasks/metrics.** Aggregated over GSM8K (gen-math + pair-logprob) and HumanEval (pair-logprob + gen-code-compile). $|\mathcal{F}_{\text{agg}}|$ is the aggregated denominator (summed across included tasks/metrics). Green column is targeted patching along $S_\ell$; blue columns are controls. Time-shuffle is a within-example temporal diagnostic (donor taken from a different decode step of the same example), preserving the decode-time distribution used to estimate $S_\ell$.

| Model | Layer | $|\mathcal{F}_{\text{agg}}|$ | Targeted patch | Controls | | | |
|---|---|---|---|---|---|---|---|
| | | | Patched(self) | Time-shuffle | RandVec (shared) | Rand Subspace | Nonshared Patch |
| Llama-2-7B-Chat | 4 | 60 | 61.7% (37/60) | 53.3% (32/60) | 20.0% (12/60) | 11.7% (7/60) | 0.0% (0/60) |
| Llama-2-7B-Chat | 10 | 86 | 38.4% (33/86) | 36.0% (31/86) | 12.8% (11/86) | 9.3% (8/86) | 3.5% (3/86) |
| Llama-2-7B-Chat | 24 | 71 | 52.1% (37/71) | 42.3% (30/71) | 4.2% (3/71) | 2.8% (2/71) | 1.4% (1/71) |
| Falcon-7B-Instruct | 4 | 107 | 55.1% (59/107) | 48.6% (52/107) | 8.4% (9/107) | 1.9% (2/107) | 0.9% (1/107) |
| Falcon-7B-Instruct | 10 | 105 | 54.3% (57/105) | 48.6% (51/105) | 5.7% (6/105) | 3.8% (4/105) | 1.0% (1/105) |
| Falcon-7B-Instruct | 24 | 80 | 37.5% (30/80) | 31.2% (25/80) | 7.5% (6/80) | 0.0% (0/80) | 1.2% (1/80) |
| Qwen2.5-7B-Instruct | 4 | 94 | 74.5% (70/94) | 70.2% (66/94) | 19.1% (18/94) | 17.0% (16/94) | 7.4% (7/94) |
| Qwen2.5-7B-Instruct | 10 | 84 | 70.2% (59/84) | 67.9% (57/84) | 17.9% (15/84) | 19.0% (16/84) | 7.1% (6/84) |
| Qwen2.5-7B-Instruct | 24 | 54 | 46.3% (25/54) | 35.2% (19/54) | 9.3% (5/54) | 3.7% (2/54) | 1.9% (1/54) |

**Dimension matching and representative small-$k$ sanity check.** To ensure the comparison is not driven by dimensionality, we report $k$-matched bases throughout (Appendix Tables 22–23). Moreover, a representative small-$k$ setting (e.g., $k_{\text{match}} = 16$) still exhibits the same qualitative mismatch on generation-heavy benchmarks, indicating the effect is not explained by "using more dimensions".

# C. Additional Experiments

## C.1. $\alpha$-Sweep in Patch back Experiments

**$\alpha$-sweep validates ablation strength as a continuous knob and transfer donors reduce answer-copying concerns.** Table 24 shows that increasing $\alpha$ continuously increases flip-rate and collapses margins, consistently across seeds. On the same AQuA flip-set, transfer donors (same-task or cross-task) achieve rescue rates comparable to self-donor patching, supporting the interpretation that patchback restores a shared decision channel rather than copying sample-specific answers.

## C.2. Additional Downstream Utility Experiments

**Appendix C.2.1: $\beta$ as a continuous *intervention-strength* knob (utility vs. worst-case stability).** Table 26 reports a three-point sweep $\beta \in \{0, 0.5, 1\}$ under true KV-cache decoding. Here $\beta$ *does not modify the model's shared workspace*; it only scales how much the *steering direction* is prevented from injecting components along the fixed decode-time shared basis $B$ (estimated once from neutral prompts at the same layer as in our workspace analysis). Intermediate $\beta=0.5$ yields intermediate behavior across tasks, consistent with $v_\beta = v - \beta BB^\top v$ acting as a smooth control on *shared-subspace interference*. For example, on SST-2 template T0, mean increases monotonically $0.0275 \to 0.0314 \to 0.0339$ while anti decreases $0.3555 \to 0.3086 \to 0.3008$. On RTE template T0, mean slightly increases $0.0104 \to 0.0112 \to 0.0116$ and

*Table 19.* **Flipset transfer patching on AQuA (aggregated).** $|\mathcal{F}_{\mathrm{flip}}|$ is the flipset size (baseline-correct examples that flip under full ablation). Both columns patch along the shared decode subspace $S_\ell$: self-donor vs transfer donor.

| Model | Layer | $|\mathcal{F}_{\mathrm{flip}}|$ | Patched(self) | Patched(transfer) |
|---|---|---|---|---|
| Llama-2-7B-Chat | 4 | 40 | 75.0 % (30/40) | 75.0 % (30/40) |
| Llama-2-7B-Chat | 10 | 169 | 75.1 % (127/169) | 78.7 % (133/169) |
| Llama-2-7B-Chat | 24 | 66 | 57.6 % (38/66) | 36.4 % (24/66) |
| Falcon-7B-Instruct | 4 | 6 | 100.0 % (6/6) | 83.3 % (5/6) |
| Falcon-7B-Instruct | 10 | 6 | 100.0 % (6/6) | 83.3 % (5/6) |
| Falcon-7B-Instruct | 24 | 88 | 100.0 % (88/88) | 92.0 % (81/88) |
| Qwen2.5-7B-Instruct | 4 | 116 | 100.0 % (116/116) | 96.6 % (112/116) |
| Qwen2.5-7B-Instruct | 10 | 128 | 100.0 % (128/128) | 96.1 % (123/128) |
| Qwen2.5-7B-Instruct | 24 | 122 | 100.0 % (122/122) | 24.6 % (30/122) |

*Table 20.* H3 subspace statistics. We estimate shared subspaces from decode vs. prefill states under $(\tau{=}10^{-3}, m{=}\mathtt{all})$. $k_{\mathrm{match}} = \min(k_{\mathrm{shared}}^{\mathrm{dec}}, k_{\mathrm{shared}}^{\mathrm{pre}}) = 48$. Principal Ang. (deg) summarize the distribution of principal angles between the decode- and prefill-estimated shared subspaces (reported as mean/p50/p95); larger values indicate stronger misalignment.

| Setting | $k_{\mathrm{shared}}^{\mathrm{dec}}$ | $k_{\mathrm{shared}}^{\mathrm{pre}}$ | $k_{\mathrm{match}}$ | Principal Ang. (deg) | | |
|---|---|---|---|---|---|---|
| | | | | mean | p50 | p95 |
| LLaMA2-7B-Chat, layer=10 | 136 | 48 | 48 | 79.74 | 82.04 | 89.12 |

anti improves $0.4844 \to 0.4766 \to 0.4648$. Aggregating per-template effects, Table 25 shows that full projection ($\beta{=}1$) trades small changes in average efficacy ($\Delta\mu = -0.0071$ on BoolQ; $-0.0011$ on RTE; $+0.0055$ on SST-2) for stability improvements in a task-dependent way (e.g., fewer worst-case failures on RTE with $\Delta\mathrm{anti}_{\mathrm{worst}} = -0.0195$, and reduced template variability with $\Delta\sigma_{\mathrm{tmpl}} = -0.0010$). Thus, $\beta$ provides an explicit, offline-computed knob for balancing steering utility against worst-case template brittleness by attenuating interference with the decode-time shared pathway (rather than contradicting its causal relevance).

**Appendix C.2.2: Per-template breakdown pinpoints the origin of worst-case gains.** Table 26 localizes worst-case behavior to specific brittle templates, showing that improvements are not driven by averaging artifacts. On SST-2, template T2 exhibits a sign flip at $\beta{=}0$ (mean $= -7\times10^{-4}$) that disappears as shared-overlap is removed, becoming positive at $\beta{=}1$ (mean $= 2.8\times10^{-3}$), which directly accounts for the improved worst-case signed behavior in Table 25 ($\Delta\mathrm{worst} = +0.0035$). RTE shows consistent reductions in anti on failure-prone templates (e.g., T0: $0.4844 \to 0.4648$) and a modest decrease in template variability (Table 25, $\Delta\sigma_{\mathrm{tmpl}} = -0.0010$). BoolQ illustrates a mild efficacy–stability tradeoff: projection slightly reduces mean and worst (Table 25), while anti improves for some templates (e.g., T1: $0.2969 \to 0.2656$), consistent with task-dependent interaction between the steering signal and the shared decode-time pathway.

**Appendix C.2.3: Candidate calibration reduces dependence on label tokenization.** To avoid confounding steering effects with poorly calibrated answer tokens, we choose forced-choice candidate pairs by maximizing baseline forced-choice accuracy on a balanced calibration subset. Table 27 lists the selected top pairs (all single-token), which match standard labelings (e.g., Yes/No; True/False; Good/Bad) and yield reasonable baseline accuracies (e.g., BoolQ: 0.583 with Yes/No; RTE: 0.615 with True/False; SST-2: 0.688 with Good/Bad). This calibration limits sensitivity to arbitrary token choices and supports interpreting subsequent robustness trends as properties of the steering intervention (and its overlap with $B$), rather than artifacts of candidate selection.

*Table 21.* **Prefill-vs-decode alignment.** All entries are accuracy in % (top) with 95% CI (bottom). Protocol: generation-based evaluation. We estimate a shared basis from decode states (Decode-shared) or from prefill states (Prefill-shared), match dimensions ($k$=129), and apply the same *decode-only* projection removal with a matched-energy random control (Random).

| Task | Baseline | Decode-shared | Prefill-shared | Random | $p$ |
|---|---|---|---|---|---|
| GSM8K | 16.8 [12.5, 21.5] | 0.0 [0.0, 0.0] | 17.6 [13.3, 22.3] | 16.8 [12.5, 21.5] | $< 10^{-4}$ |
| CSQA | 37.9 [32.0, 44.1] | 23.4 [18.4, 28.9] | 31.2 [25.8, 36.7] | 40.6 [34.8, 46.5] | 0.038 |
| StrategyQA | 43.8 [37.9, 50.0] | 10.5 [7.0, 14.5] | 41.4 [35.5, 47.7] | 46.5 [40.2, 52.7] | $< 10^{-4}$ |
| ARC-C | 49.6 [43.8, 55.5] | 19.5 [14.8, 24.6] | 39.1 [33.2, 44.9] | 48.4 [42.6, 54.7] | $< 10^{-4}$ |
| OBQA | 48.4 [42.2, 54.7] | 25.0 [19.5, 30.5] | 41.4 [35.5, 47.3] | 45.7 [39.5, 51.6] | $< 10^{-4}$ |
| QASC | 28.9 [23.4, 34.4] | 10.2 [6.6, 13.7] | 27.7 [22.3, 33.2] | 29.3 [23.8, 34.8] | $< 10^{-4}$ |
| LogiQA | 22.7 [17.6, 27.7] | 25.8 [20.7, 31.2] | 15.2 [10.9, 19.9] | 19.5 [14.8, 24.2] | 0.0032 |
| BoolQ | 65.6 [59.8, 71.5] | 55.5 [49.2, 61.3] | 66.0 [60.2, 71.9] | 70.3 [64.8, 75.8] | 0.0029 |

*Table 22.* Same as Table 23, but with a representative small rank ($k_{\mathrm{match}} = 16$) to rule out dimension-driven artifacts.

| Task | Protocol | $n$ | Baseline | Decode-shared ($k$=16) | Prefill-shared ($k$=16) | Random ($k$=16) | $\Delta(\mathrm{Decode} - \mathrm{Prefill})$ | $p$ |
|---|---|---|---|---|---|---|---|---|
| gsm8k | generation | 1319 | 19.6 [17.5, 21.8] | 3.6 [2.6, 4.6] | 19.1 [17.0, 21.2] | 19.7 [17.5, 21.8] | -15.5 [-17.9, -13.3] | 0.0001 |
| commonsenseqa | forced_choice | 1221 | 55.6 [52.8, 58.4] | 36.4 [33.7, 39.2] | 36.0 [33.3, 38.7] | 55.4 [52.7, 58.2] | +0.4 [-1.4, +2.2] | 0.716 |
| strategyqa | forced_choice | 687 | 50.4 [46.7, 54.0] | 52.0 [48.2, 55.7] | 48.6 [44.8, 52.4] | 50.5 [46.9, 54.0] | +3.3 [+0.9, +5.8] | 0.0089 |
| aqua | forced_choice | 254 | 27.2 [21.7, 32.7] | 25.2 [20.1, 30.3] | 25.2 [20.1, 30.7] | 27.2 [22.0, 32.7] | +0.0 [-3.5, +3.5] | 1 |
| arc_challenge | forced_choice | 1172 | 53.1 [50.2, 55.9] | 33.7 [31.1, 36.5] | 33.4 [30.8, 36.2] | 53.0 [50.2, 55.9] | +0.3 [-1.5, +2.1] | 0.724 |
| openbookqa | forced_choice | 500 | 57.8 [53.6, 62.2] | 36.8 [32.6, 41.0] | 34.2 [30.0, 38.6] | 58.2 [53.8, 62.4] | +2.6 [+0.0, +5.2] | 0.0372 |
| qasc | forced_choice | 926 | 52.6 [49.4, 55.8] | 32.9 [29.9, 36.0] | 36.1 [32.9, 39.2] | 52.2 [48.9, 55.3] | -3.1 [-5.4, -0.9] | 0.0047 |
| logiqa | forced_choice | 651 | 26.0 [22.6, 29.3] | 16.7 [14.0, 19.7] | 17.2 [14.3, 20.1] | 25.8 [22.6, 29.0] | -0.5 [-1.8, +0.8] | 0.644 |
| boolq | forced_choice | 2048 | 68.0 [65.9, 70.0] | 56.2 [54.1, 58.3] | 63.5 [61.3, 65.5] | 68.0 [66.0, 70.0] | -7.2 [-9.5, -4.9] | 0.0001 |

## C.3. Additional Downstream Style Metrics and Shared-Subspace Diagnostics

**Appendix C.3.1: Pirate shareness sanity check, Table 28.** This table verifies that the projection operator behaves as intended and that $v$ is substantially aligned with the decode-trace PCA span. As $k$ increases from 16 to 128, shareness$(v) = \|B_k^\top v\| / \|v\|$ rises from 0.465 to 0.545, indicating that a large fraction of the steering direction lies in the "shared" subspace captured by high-variance decode activations. In contrast, shareness$(v_{\mathrm{fixed},k})$ remains $\approx 10^{-7}$ for all $k$, confirming that $v_{\mathrm{fixed},k} = v - B_k(B_k^\top v)$ is numerically orthogonal to the span up to floating-point tolerance. Together, these results establish that the experiment is not merely showing behavioral changes under arbitrary edits: the intervention explicitly removes the component of $v$ contained in a well-defined shared subspace, and does so accurately.

**Appendix C.3.2: Pirate-hit intensity, Table 29.** While the main table reports binary success at a threshold of `pirate_hits` $\geq 2$, Appendix Table 29 provides a more graded view of the same phenomenon by summarizing the mean and worst-case `pirate_hits` across templates. The hit statistics reinforce the tradeoff observed in success rates: moderate projection (e.g., $k = 16$–64) increases the intensity of lexical pirate markers relative to $v_{\mathrm{orig}}$ under both greedy and sampling, whereas stronger projection (k=128) substantially reduces hit intensity, particularly under greedy decoding (near-zero worst-case). This corroborates an entanglement interpretation: removing a small part of the shared space can "denoise" the steering direction so pirate tokens appear more frequently when they appear at all, but removing a larger shared span begins to excise style-relevant components and suppresses lexical realization, with the remaining signal becoming more brittle and decoding-dependent.

*Table 23.* Prefill–vs.–Decode estimator swap under a fixed *decode-only* intervention locus. All bases are dimension-matched ($k_{match} = 126$). Forced-choice: warmup $W = 0$ and prefix_mode=auto.

| Task | Protocol | $n$ | Baseline | Decode-shared ($k$=126) | Prefill-shared ($k$=126) | Random ($k$=126) | $\Delta$(Decode$-$Prefill) | $p$ |
|---|---|---|---|---|---|---|---|---|
| gsm8k | generation | 1319 | 19.6 [17.5, 21.8] | 0.3 [0.1, 0.6] | 19.1 [17.1, 21.3] | 19.4 [17.3, 21.6] | -18.8 [-21.0, -16.8] | 0.0001 |
| commonsenseqa | forced_choice | 1221 | 55.6 [52.8, 58.4] | 23.8 [21.5, 26.1] | 20.4 [18.1, 22.7] | 54.5 [51.8, 57.3] | +3.4 [+2.0, +4.8] | 0.0001 |
| strategyqa | forced_choice | 687 | 50.4 [46.7, 54.0] | 52.8 [49.1, 56.6] | 48.5 [44.8, 52.3] | 50.4 [46.6, 53.9] | +4.4 [+0.9, +7.7] | 0.0131 |
| aqua | forced_choice | 254 | 27.2 [21.7, 32.7] | 20.1 [15.4, 25.2] | 25.6 [20.5, 31.1] | 27.6 [22.0, 33.1] | -5.5 [-9.8, -1.6] | 0.0147 |
| arc_challenge | forced_choice | 1172 | 53.1 [50.2, 55.9] | 27.9 [25.3, 30.5] | 23.0 [20.6, 25.4] | 52.0 [49.2, 54.9] | +4.9 [+3.0, +6.8] | 0.0001 |
| openbookqa | forced_choice | 500 | 57.8 [53.6, 62.2] | 26.8 [23.0, 30.8] | 27.8 [23.8, 31.8] | 58.8 [54.4, 62.8] | -1.0 [-3.4, +1.2] | 0.501 |
| qasc | forced_choice | 926 | 52.6 [49.4, 55.8] | 18.5 [16.0, 21.0] | 20.7 [18.3, 23.3] | 51.8 [48.5, 55.0] | -2.3 [-4.3, -0.1] | 0.0489 |
| logiqa | forced_choice | 651 | 26.0 [22.6, 29.3] | 18.0 [15.2, 21.0] | 16.7 [13.8, 19.7] | 25.2 [22.0, 28.4] | +1.2 [-0.3, +2.8] | 0.188 |
| boolq | forced_choice | 2048 | 68.0 [65.9, 70.0] | 50.8 [48.6, 52.9] | 62.6 [60.4, 64.6] | 71.2 [69.3, 73.2] | -11.8 [-14.9, -8.8] | 0.0001 |

*Table 24.* **AQuA flip-set supplementary experiments. (A)** $\alpha$**-sweep on a fixed flip-set defined at** $\alpha$=1. We report flip_rate (still flipped on the flip-set), ablated accuracy on the flip-set, and mean $\Delta$m (ablated vs. baseline). **(B) Transfer-donor patching on the same flip-set:** donors from other examples/tasks yield comparable rescue to self-donor.

| | Seed 123 ($n$=42) | | | Seed 456 ($n$=43) | | |
|---|---|---|---|---|---|---|
| $\alpha$ | flip_rate(%) | ablt_acc(%) | mean $\Delta$m | flip_rate(%) | ablt_acc(%) | mean $\Delta$m |
| 0.00 | 0.0 | 100.0 | 0.000 | 0.0 | 100.0 | 0.000 |
| 0.50 | 28.6 | 71.4 | -0.318 | 32.6 | 67.4 | -0.388 |
| 0.75 | 71.4 | 28.6 | -1.475 | 88.4 | 11.6 | -1.920 |
| 1.00 | 100.0 | 0.0 | -3.695 | 100.0 | 0.0 | -3.899 |
| **(B) Transfer donor patching on the flip-set (rescue% / mean $\Delta$m)** | | | | | | |
| Setting | Seed | | Self-donor | | Transfer-donor | |
| Same-task donors | 123 | | 73.8 / 3.31 | | 76.2 / 3.29 | |
| Cross-task donors (MC, baseline-correct) | 123 | | 73.8 / 3.31 | | 78.6 / 3.39 | |
| Cross-task donors (MC, baseline-correct) | 456 | | 79.1 / 3.57 | | 81.4 / 3.64 | |

## C.4. Sharedness under Diverse Scales

We extend our evaluations beyond the original 7B scale to verify the generality of our findings. Rather than performing a comprehensive scaling-law study, these experiments test whether our core qualitative insights hold across smaller and larger models. Crucially, we confirm that across scales, the decode-shared subspace remains causally impactful, while prefill-based proxies consistently fail to predict decode-time behavior.

Across 3B, 13B, and 70B models, DecodeShare recovers compact but statistically significant shared decode-time subspaces (Table 34). Decode-time shared-subspace removal remains harmful on the aligned four-task slice, including a strong Llama-3.2-3B layer-27 result and a clean Llama-2-70B layer-56 result where all four tasks degrade significantly (Table 35). Finally, the prefill/decode mismatch persists across scale: decode-estimated bases cause substantially larger decode-time drops than prefill-estimated or random bases under the same intervention protocol (Table 36). These results support the limited claim that the phenomenon is not confined to 7B-class models, while leaving systematic scaling trends to future work.

## C.5. Policy Sensitivity and Computational Overhead

**Policy sensitivity.** DecodeShare is policy-aligned rather than policy-invariant: the shared subspace is estimated from the decode-time state distribution induced by the serving-time decoding policy. The main experiments use greedy decoding because it provides the cleanest controlled setting for causal tests. To test whether the existence of shared structure is an artifact of greedy decoding, we additionally estimate shared bases under sampling policies with temperatures $T \in \{0.7, 1.0, 1.3\}$ on Llama-2-7B-Chat.

Table 37 summarizes the shared ratio and the mean principal angle to the corresponding greedy-estimated basis. Across all tested layers and temperatures, the H1 sharedness test remains significant ($p < 0.05$). The learned basis rotates substantially relative to greedy decoding, as indicated by the large principal angles, but the shared set remains compact and statistically distinguishable from the null. Thus, changing the decoding policy affects the orientation of the estimated shared workspace,

---

**Representative Example: Projection removes pirate style (sample, seed=3)**

---

**Prompt:** Please answer the following question:
    Explain what climate change means and why it matters.
**Threshold:** `pirate_hits` $\geq$ 2;   **Injection:** start_step=1, first_n=24;   $\alpha = 45$.

---

**`no_steer`** (hits=0, success=0)

   Of course, I'd be happy to help! Climate change refers to the long-term warming of the planet due to an increase in the average global temperature. This warming is primarily caused by human activities that release greenhouse gases, such as carbon dioxide and methane, into the atmosphere. These gases trap heat and cause the Earth's temperature to rise, leading to a range of consequences.
   Why does climate change matter? Here are some reasons:
   (1) Rising sea levels . . . (2) Extreme weather events . . .

**`v_orig_a45`** (hits=3, success=1)

   Of course, matey! Climate change be the shifts in yer climate, arrr, caused by the warmin' o' the planet due to the release o' greenhouse gases into the atmosphere. These gases trap heat and cause the Earth's temperature to rise, which leads to changes in weather patterns, sea levels, and ecosystems.
   Why does it matter, ye ask? Well, me hearty, here be some reasons:
   (1) Rising sea levels . . . (2) Extreme weather . . .

**`v_fixed_k128_a45`** (hits=0, success=0)

   Of course, I'd be happy to help! Climate change refers to the long-term warming of the planet due to an increase in the average global temperature . . .
   Why does climate change matter? Here are some reasons:
   (1) Rising sea levels . . . (2) Extreme weather events . . .

---

*Figure 15.* **Same prompt, same decoding, different vectors.** $v_{\text{orig}}$ triggers pirate lexicon (success), while projecting out the shared PCA span at $k = 128$ suppresses the style (failure), consistent with $v$ being entangled with the shared decode subspace.

*Table 25.* **Effect of fully removing the shared component.** $\Delta$ values compare decode_fixed ($\beta=1$) vs decode_est ($\beta=0$). We emphasize changes in worst-case signed behavior ($\Delta$worst), and report $\Delta$anti$_{\text{worst}}$ as a complementary diagnostic.

| Task | $\Delta\mu$ | $\Delta\sigma_{\text{tmpl}}$ | $\Delta$worst | $\Delta$anti$_{\text{worst}}$ |
|------|------|------|------|------|
| BoolQ | -0.0071 | +0.0015 | -0.0090 | 0.0000 |
| RTE | -0.0011 | -0.0010 | -0.0015 | -0.0195 |
| SST-2 | +0.0055 | +0.0008 | +0.0035 | +0.0313 |

but does not remove the existence of a decode-time shared structure.

**Computational overhead.**   DecodeShare consists of a one-time offline basis-estimation stage and a lightweight online projection stage. The offline cost is dominated by collecting decode-time activations and computing the shared basis. On an H200 GPU with Llama-2-7B-Chat at layer $\ell = 10$, the full DecodeShare pipeline over five tasks takes 296.5 seconds, compared with 48.5 seconds for the unmodified model run; basis estimation accounts for 87.6% of the total pipeline runtime.

At deployment time, the additional cost is only the projection onto the estimated shared basis at the edited layer. This projection scales as $O(d|S_\ell|)$ per edited decode state and does not modify the KV cache. In our measurement, the online intervention changes throughput from 102.95 tokens/s to 81.10 tokens/s, corresponding to a 21.2% throughput overhead, with only 1.16 MiB additional memory. These measurements indicate that the main cost is offline estimation, while the online cost is modest for a compact shared core.

## C.6. Step-Localized Ablation Across Decode Trajectories

The main causal tests remove the shared subspace throughout the decode trajectory. To check whether the effect is merely a final-token readout artifact, we run a step-localized ablation on extended GSM8K generation trajectories with Llama-2-7B-Chat. For each generated trajectory, we partition decode steps into three contiguous windows: early, middle, and late thirds. We then apply the same shared-subspace removal only within one window, leaving the rest of the trajectory unchanged.

---

**Flip Example (forced-choice; ex_id=aqua-test-9)**

---

**Gold:** B    **Intervention:** shared-removal at layer $\ell=10$ (decode-only).

**Prompt (truncated):**

```
Question:  A newspaper costs $4 on Sunday ...  how many newspapers does it buy
on Monday?  Choices:  A)45 B)15 C)60 D)30 E)75
```

---

**baseline**: B (correct=true, margin=+1.08)
**shared-removed**: A (correct=false, margin=-3.30)    *(flip)*

---

*Figure 16.* **Flip case study (forced-choice).** The baseline forced-choice prediction matches the gold answer. Removing the shared decode subspace at layer $\ell = 10$ during cached decoding flips the predicted choice demonstrating that the shared decode pathway can be *causally necessary*.

---

**Patchback example (boolq; ex_id=boolq-validation-3)**

---

**Dataset**: boolq
**Gold**: B
**Prompt** (truncated):

```
Question:  Passage:  Open carry is also legal throughout North Carolina.  In
the town of Chapel Hill, open carry is restricted to guns of a certain minimum
size ...  \nQuestion do you need a permit to open carry in nc Choices:  A) Yes
B) No Reason step by step.  At the end, write exactly one line:  "Final answer:
<A/B>".
```

---

**baseline**: B (correct=true, margin=+5.19)
**patchback (W={0,1})**: B (correct=true, margin=+5.19)
**rand subspace**: A (correct=false, margin=-3.66)
**time-shuf donor**: B (correct=true, margin=+2.84)
**non-shared patch**: A (correct=false, margin=-1.96)

---

*Figure 17.* **Patchback case study (forced-choice).** Patching back only the removed *shared* component over a narrow decode window $W = \{0, 1\}$ restores the baseline answer, while energy/dimension-matched controls do not.

Table 38 shows that all three localized interventions reduce accuracy relative to the unmodified baseline. The middle-window intervention is the most damaging, while early- and late-window interventions also cause substantial degradation. Full-trajectory ablation collapses accuracy to 0.0%. This pattern suggests that the shared decode-time subspace is used throughout the reasoning trajectory, rather than acting only as a single final-token answer switch.

We interpret this result conservatively. It does not imply that the shared subspace has the same functional role at every decode step. Rather, it shows that the causal effect is distributed across the decode trajectory and is not confined to the final answer token.

### C.7. Readout Remapping and Verbalizer Controls

A possible concern is that shared-subspace removal might primarily disrupt a narrow answer-letter readout rather than a broader decode-time decision component. The main paper already addresses this through the $Q_{\text{out}}/Q_{\text{core}}$ split: ablating the residual core reproduces most of the full shared-subspace effect, whereas ablating the small readout slice is nearly inert. Here we add a complementary verbalizer control.

We hold the same layer-10 decode-shared basis fixed and change only the forced-choice readout protocol, without re-estimating the basis. Under the original answer-label readout, the mean shared-ablation drop is $-11.7$ points relative to baseline, while matched-energy controls remain close to zero. Under a remapped `option_label` readout, e.g., "Final answer: Option A/B/...", the drop remains $-10.9$ points, preserving 93% of the original effect. Under a stronger `option_text` readout, where choices are scored using the full option text rather than a short label token, the effect remains

*Table 26.* **Per-template breakdown (final $\lambda$).** Per-template mean correct-signed margin shift (mean) and per-template failure rate (anti) for $\beta \in \{0, 0.5, 1\}$.

| | | $\beta=0$ | | $\beta=0.5$ | | $\beta=1$ | |
|---|---|---|---|---|---|---|---|
| Task | Tmpl. | mean↑ | anti↓ | mean↑ | anti↓ | mean↑ | anti↓ |
| | T0 | 0.0401 | 0.3633 | 0.0372 | 0.3594 | 0.0323 | 0.3633 |
| BoolQ | T1 | 0.0393 | 0.2969 | 0.0380 | 0.2812 | 0.0347 | 0.2656 |
| | T2 | 0.0357 | 0.3594 | 0.0321 | 0.3594 | 0.0267 | 0.3516 |
| | T0 | 0.0104 | 0.4844 | 0.0112 | 0.4766 | 0.0116 | 0.4648 |
| RTE | T1 | 0.0307 | 0.3320 | 0.0301 | 0.3242 | 0.0281 | 0.3203 |
| | T2 | 0.0107 | 0.3789 | 0.0101 | 0.3828 | 0.0089 | 0.3750 |
| | T0 | 0.0275 | 0.3555 | 0.0314 | 0.3086 | 0.0339 | 0.3008 |
| SST-2 | T1 | 0.0030 | 0.4297 | 0.0063 | 0.4492 | 0.0095 | 0.4648 |
| | T2 | -0.0007 | 0.4336 | 0.0011 | 0.4492 | 0.0028 | 0.4219 |

*Table 27.* **Candidate calibration (top pairs).** Forced-choice candidate pairs selected by maximizing baseline forced-choice accuracy on a balanced subset. All pairs are single-token.

| Task | Candidate pair | Baseline acc |
|---|---|---|
| BoolQ | Yes/No | 0.583 |
| | True/False | 0.552 |
| RTE | True/False | 0.615 |
| | Yes/No | 0.562 |
| SST-2 | Good/Bad | 0.688 |
| | Yes/No | 0.599 |
| | True/False | 0.599 |

negative at $-7.8$ points, although the magnitude attenuates as expected for a noisier multi-token verbalizer.

These results argue against the trivial explanation that DECODESHARE only disrupts the original answer-letter surface form. We do not claim strict readout invariance under every possible verbalizer. Instead, the conservative conclusion is that the dominant causal effect is largely preserved under a simple label remapping and remains directionally consistent under a stronger full-text readout, while matched controls stay near baseline.

# D. Discussion.

## D.1. Broader positioning and additional related work

**Broader positioning.** Our results sit at the intersection of (i) mechanistic interpretability, which aims to reverse-engineer internal computations and validate hypotheses with interventions, and (ii) systems/efficiency work that treats KV cache management as the defining constraint of modern inference. (Bereska & Gavves, 2024; Elhage et al., 2021; Olsson et al., 2022; Kwon et al., 2023; Xiao et al., 2024; Shao et al., 2026; Wu et al., 2026) From the interpretability perspective, circuit-style analyses and scalable discovery methods (e.g., pruning) motivate studying reusable structure across tasks, aligning with our focus on cross-task shared subspaces. (Merullo et al., 2024; Bhaskar et al., 2024)

**Mechanistic interpretability and shared structure.** Representation similarity tools (SVCCA/CKA) have long been used to compare internal representations across networks and training regimes, and recent hypotheses argue that important structure may concentrate in shared low-dimensional subspaces. (Raghu et al., 2017; Kornblith et al., 2019; Kaushik et al., 2025; Wang et al., 2025) **KV-cache compression/eviction beyond streaming.** Beyond streaming, a growing literature compresses or restructures KV caches (including low-rank and reconstruction-based schemes), closely related to our emphasis on decode-time low-dimensional structure and regime mismatch. (Saxena et al., 2024; Li et al., 2024; Cai et al., 2024; Kim et al., 2025; Chang et al., 2025; Khalaf et al., 2025) **Steering directions and reliability.** Steering vectors and representation engineering methods (including activation addition and direction-based control) provide complementary control mechanisms; their reported brittleness and reliability analyses motivate our regime-specific, decision-level evaluations and null controls. (Turner et al., 2023; Rimsky et al., 2024; Konen et al., 2024; Todd et al., 2024; Arditi et al., 2024; Tan et al., 2024; Deng et al., 2025; Belitsky et al., 2025)

**Differences from closely related subspace work.** Arturi et al. analyze *parameter-level* shared subspaces in weight updates across tasks and connect this geometry to emergent misalignment (Arturi et al., 2025); in contrast, we study an *activation-level* shared workspace that appears specifically in *decode-time* states under true KV-cached decoding, and we validate it with decode-only causal interventions (H2) and estimator–distribution mismatch tests (H3). MSRS proposes

*Table 28.* **Appendix: Shareness sanity check.** As $k$ increases, the PCA span captures more of $v$ (sharedness rises), while $v_{\text{fixed},k}$ is numerically orthogonal to the span ($\approx 10^{-7}$).

| $k$ | sharedness$(v) = \frac{\|B_k^\top v\|}{\|v\|}$ | sharedness$(v_{\text{fixed},k}) = \frac{\|B_k^\top v_{\text{fixed},k}\|}{\|v_{\text{fixed},k}\|}$ |
|---|---|---|
| 16 | 0.4651 | $9.9 \times 10^{-8}$ |
| 32 | 0.4864 | $1.0 \times 10^{-7}$ |
| 64 | 0.5166 | $1.2 \times 10^{-7}$ |
| 128 | 0.5455 | $1.4 \times 10^{-7}$ |

*Table 29.* **Appendix: Pirate-hit intensity.** Mean±std and worst-case (minimum) across templates for `pirate_hits`. This complements Table 33 by showing how strongly the lexicon is triggered when it triggers at all.

| Method | $k$ | Greedy hits | Sample hits (seeds 1–3) |
|---|---|---|---|
| no_steer | – | $0.000 \pm 0.000 \ (0.000)$ | $0.000 \pm 0.000 \ (0.000)$ |
| rand0_a45 | – | $0.000 \pm 0.000 \ (0.000)$ | $0.000 \pm 0.000 \ (0.000)$ |
| v_orig_a45 | – | $0.090 \pm 0.136 \ (0.000)$ | $0.127 \pm 0.063 \ (0.050)$ |
| v_fixed_k16_a45 | 16 | $0.400 \pm 0.130 \ (0.150)$ | $0.360 \pm 0.034 \ (0.317)$ |
| v_fixed_k32_a45 | 32 | $0.210 \pm 0.116 \ (0.050)$ | $0.340 \pm 0.064 \ (0.267)$ |
| v_fixed_k64_a45 | 64 | $0.250 \pm 0.145 \ (0.050)$ | $0.413 \pm 0.062 \ (0.317)$ |
| v_fixed_k128_a45 | 128 | $0.040 \pm 0.037 \ (0.000)$ | $0.213 \pm 0.097 \ (0.083)$ |

*multi-subspace* representation steering to compose attribute-aligned behaviors (Jiang et al., 2025); our focus is not to design a stronger steering method, but to diagnose a regime-specific failure mode (shared-workspace interference under KV-cached decoding) and provide a simple projection-based repair knob with matched controls. Son et al. study *semantic convergence across models/scales* by comparing learned representations across different LLMs (Son et al., 2025); we instead ask whether *cross-task* shared structure exists within a fixed model at decision time, and whether it is causally necessary for decode-time decisions. Zhang and Zhou address interference in *adaptation/merging* by enforcing orthogonal *parameter* subspaces (e.g., for LoRA) (Zhang & Zhou, 2025); we do not constrain training, but identify and intervene on a decode-time activation subspace at inference. Xie et al. remove nuisance low-rank subspaces to obtain language-agnostic multilingual representations (Xie et al., 2022); similarly, we use projection operations, but our target is a task-shared decode-time workspace and we establish its relevance via matched nulls, budget-matched controls, and patchback sufficiency tests. Falissard et al. learn prefix-parameter subspaces to improve generalization during fine-tuning (Falissard et al., 2023); we do not learn subspaces, but estimate them from activations and show that prefill-estimated bases often fail to transfer to decode-time causal effects (H3). Finally, Whitaker et al. study identification of shared semantics via contrastive latent alignment across modalities (Whitaker et al., 2025); while our setting is unimodal and task-based, we share the goal of isolating compact shared structure, and we complement geometric evidence with decode-time causal validation.

*Table 30.* **All-tasks shared basis: decode-only ablation (generation).** SHARED(FULL) removes the decode-estimated shared subspace; RAND(FULL) is a dimension/energy-matched non-shared control (see §2.4). Entries report mean accuracy with 95% CIs; $\Delta$ is SHARED(FULL)−Baseline with paired test $p$. A full sweep across models and layers is provided in the Appendix.

| | All-tasks basis (Mode: All; Generation) | | | |
|---|---|---|---|---|
| Task | Baseline | Shared (full) | Rand (full) | $\Delta$(Shared−Baseline) |
| ARC-C | 26.2 [21.1, 31.6] | 7.8 [4.7, 11.3] | 21.5 [16.4, 26.6] | -18.4 [-23.8, -12.9], $p < 0.001$ |
| BoolQ | 31.6 [26.2, 37.5] | 18.0 [13.3, 22.7] | 30.9 [25.4, 36.3] | -13.7 [-19.9, -7.8], $p < 0.001$ |
| CSQA | 16.8 [12.5, 21.5] | 8.2 [5.1, 11.7] | 19.1 [14.5, 24.2] | -8.6 [-13.7, -3.5], $p = 0.001$ |
| GSM8K | 16.4 [12.1, 21.1] | 0.0 [0.0, 0.0] | 15.6 [11.3, 20.3] | -16.4 [-21.1, -12.1], $p < 0.001$ |
| LogiQA | 13.3 [9.4, 17.6] | 5.9 [3.1, 9.0] | 8.2 [5.1, 11.7] | -7.4 [-12.1, -2.7], $p = 0.002$ |
| OBQA | 25.4 [20.3, 30.5] | 6.2 [3.5, 9.4] | 27.0 [21.5, 32.4] | -19.1 [-24.6, -13.7], $p < 0.001$ |
| PIQA | 39.8 [34.0, 46.1] | 19.1 [14.5, 24.2] | 41.4 [35.5, 47.7] | -20.7 [-27.3, -14.1], $p < 0.001$ |
| QASC | 18.0 [13.7, 22.7] | 6.2 [3.5, 9.4] | 19.5 [14.8, 24.6] | -11.7 [-17.2, -6.2], $p < 0.001$ |
| StratQA | 45.7 [39.5, 52.0] | 10.9 [7.4, 14.8] | 43.4 [37.5, 49.2] | -34.8 [-41.4, -28.1], $p < 0.001$ |

*Table 31.* **LOTO results (two panels).** Cells report mean (top) and 95% CI (bottom). $p$ tests Shared(full) vs. Baseline, layer $\ell = 10$. A full sweep across models and layers is provided in the Appendix.

| | | Leave-one-task-out (LOTO) results | | | |
|---|---|---|---|---|---|
| | | Part A: Subspace Intervention Experiment (Generation) | | | |
| Task | $n$ | Baseline | Shared(full) | Rand(full) | $p$-value |
| AQuA | 254 | 17.7 [13.4, 22.4] | 15.4 [11.0, 19.7] | 19.7 [15.0, 24.8] | 0.483 |
| BoolQ | 2048 | 32.2 [30.2, 34.2] | 21.1 [19.3, 22.9] | 32.2 [30.1, 34.2] | $< 0.001$ |
| CSQA | 1221 | 21.3 [19.0, 23.6] | 10.9 [9.2, 12.6] | 21.5 [19.2, 23.8] | $< 0.001$ |
| GSM8K | 1319 | 15.7 [13.8, 17.7] | 0.1 [0.0, 0.2] | 13.3 [11.6, 15.2] | $< 0.001$ |
| OBQA | 500 | 26.2 [22.4, 30.0] | 13.2 [10.2, 16.4] | 25.4 [21.6, 29.4] | $< 0.001$ |
| PIQA | 1838 | 37.4 [35.3, 39.7] | 31.4 [29.3, 33.6] | 36.7 [34.5, 38.9] | $< 0.001$ |
| QASC | 926 | 20.2 [17.6, 22.9] | 8.9 [7.1, 10.7] | 20.4 [17.8, 23.0] | $< 0.001$ |
| StratQA | 687 | 48.2 [44.4, 51.7] | 20.2 [17.3, 23.3] | 47.3 [43.7, 50.9] | $< 0.001$ |
| **Avg (8 tasks)** | — | 27.4 | 15.1 | 27.1 | — |
| | | Part B: Subspace Intervention Experiment (Forced-Choice) | | | |
| Task | $n$ | Baseline | Shared(full) | Rand(full) | $p$-value |
| AQuA | 254 | 24.0 [18.9, 29.5] | 17.3 [12.6, 22.0] | 22.0 [16.9, 27.2] | 0.0547 |
| ARC-C | 1172 | 50.9 [48.1, 53.8] | 40.4 [37.5, 43.2] | 50.6 [47.8, 53.4] | 0.0001 |
| CSQA | 1221 | 54.1 [51.3, 56.8] | 50.0 [47.3, 52.9] | 54.5 [51.6, 57.2] | 0.0043 |
| GSM8K | 1319 | 4.9 [3.8, 6.0] | 2.3 [1.5, 3.1] | 4.7 [3.6, 5.8] | 0.0001 |
| LogiQA | 651 | 32.7 [29.0, 36.4] | 26.9 [23.5, 30.3] | 33.2 [29.6, 36.9] | 0.0147 |
| OBQA | 500 | 50.2 [45.8, 54.6] | 41.6 [37.4, 45.8] | 52.4 [48.0, 56.8] | 0.0005 |
| QASC | 926 | 48.5 [45.2, 51.7] | 40.7 [37.6, 43.8] | 49.0 [45.8, 52.2] | 0.0001 |
| StratQA | 687 | 55.5 [51.7, 59.1] | 52.3 [48.5, 55.9] | 56.3 [52.5, 60.0] | 0.0182 |

*Table 32.* **Staged Generation Evaluation.** (meta-llama/Llama-2-7b-chat-hf, fp32; decode-only; layer=10; $\tau$=0.001; mode=all). Values are means over 8 tasks. *Full* denotes standard end-to-end generation under the intervention; *Staged* constrains the final-answer format to reduce format/extraction confounds. Acc/Extr/EOS are in %, and Len is in tokens.

| | Baseline | Shared removal | | Random control | |
|---|---|---|---|---|---|
| Metric | Value | Full | Staged | Full | Staged |
| Acc (%) | 27.4 | 15.1 | 16.4 | 27.1 | 26.8 |
| Extr (%) | 70.5 | 51.5 | 52.2 | 70.0 | 70.1 |
| EOS (%) | 90.8 | 45.3 | 51.0 | 92.4 | 91.9 |
| Len (tok.) | 92.5 | 144.0 | 139.2 | 86.7 | 87.1 |

*Table 33.* **Shared-space sweep + steering efficacy (Llama-2-7B-Chat).** We estimate a pirate style direction $v$ at layer 28 (KV-aligned decode, $v_{\text{decode\_steps}} = 16$, $v_n = 16$, $\alpha = 45$, inject window: start_step=1, first_n=24) and construct a PCA basis $B_k$ from decode hidden-state traces of no-steer generations. *Sharedness* is $\|B_k^\top v\|/\|v\|$; $v_{\text{fixed},k} = v - B_k(B_k^\top v)$ has $\|B_k^\top v_{\text{fixed},k}\|/\|v_{\text{fixed},k}\| \approx 0$ for all $k$ (Appendix Table 28). Success is $\mathbb{1}[\texttt{pirate\_hits} \geq 2]$. Numbers are **per-template** mean±std, with worst-case (minimum) across templates in parentheses.

| Method | $k$ | Sharedness $(v)$ | Greedy success | Sample success (seeds 1–3) |
|---|---|---|---|---|
| No steer | – | – | $0.000 \pm 0.000\,(0.000)$ | $0.000 \pm 0.000\,(0.000)$ |
| Random direction | – | – | $0.000 \pm 0.000\,(0.000)$ | $0.000 \pm 0.000\,(0.000)$ |
| $v_{\text{orig}}$ | – | – | $0.040 \pm 0.058\,(0.000)$ | $0.033 \pm 0.024\,(0.000)$ |
| $v_{\text{fixed}}$ | 16 | 0.465 | $\mathbf{0.120} \pm 0.060\,(0.000)$ | $0.100 \pm 0.015\,(0.083)$ |
| $v_{\text{fixed}}$ | 32 | 0.486 | $0.050 \pm 0.032\,(0.000)$ | $0.097 \pm 0.029\,(0.067)$ |
| $v_{\text{fixed}}$ | 64 | 0.517 | $0.090 \pm 0.066\,(0.000)$ | $\mathbf{0.127} \pm 0.017\,(0.100)$ |
| $v_{\text{fixed}}$ | 128 | 0.545 | $0.000 \pm 0.000\,(0.000)$ | $0.063 \pm 0.027\,(0.033)$ |

*Table 34.* **H1 beyond the original 7B setting.** All runs use $\tau = 10^{-3}$ and $m_{\text{shared}} = $ all. Ratio is $|S|/\text{cross\_dim}$.

| Model | Layer | Cross-dim | $|S|$ | Ratio | $(p_{\text{perm}}, p_{\text{scram}})$ |
|---|---|---|---|---|---|
| Llama-3.2-3B | 27 | 1792 | 129 | 7.20% | (0.015, 0.048) |
| Llama-2-13B | 10 | 2427 | 88 | 3.63% | (0.015, 0.048) |
| Llama-2-13B | 28 | 2746 | 132 | 4.81% | (0.015, 0.048) |
| Llama-2-13B | 39 | 2631 | 87 | 3.31% | (0.015, 0.048) |
| Llama-2-70B | 32 | 3522 | 21 | 0.60% | (0.015, 0.048) |
| Llama-2-70B | 48 | 4018 | 2 | 0.05% | (0.015, 0.048) |
| Llama-2-70B | 56 | 4096 | 13 | 0.32% | (0.015, 0.048) |

*Table 35.* **H2 causal degradation beyond 7B.** Entries are $\Delta$(shared − baseline) in accuracy points under decode-time shared-subspace removal. Rows use the aligned four-task multiple-choice slice. $^*$ denotes $p < 0.05$.

| Model | Layer | CSQA | OBQA | QASC | ARC-C | Mean / Sig. |
|---|---|---|---|---|---|---|
| Llama-3.2-3B | 27 | $-25.0^*$ | $-10.9^*$ | $-42.2^*$ | $-9.4$ | $-21.9$ / 3 of 4 |
| Llama-2-13B | 10 | $-10.2^*$ | $-11.7^*$ | $-10.2^*$ | $-6.2^*$ | $-9.6$ / 4 of 4 |
| Llama-2-13B | 28 | $-13.3^*$ | $-10.2^*$ | $-18.0^*$ | $-6.2$ | $-11.9$ / 3 of 4 |
| Llama-2-13B | 39 | $-21.9^*$ | $-2.3$ | $-22.7^*$ | $+0.0$ | $-11.7$ / 2 of 4 |
| Llama-2-70B | 32 | $-26.6^*$ | $-14.1^*$ | $-7.8$ | $-12.5^*$ | $-15.2$ / 3 of 4 |
| Llama-2-70B | 48 | $-28.1^*$ | $-21.9^*$ | $-7.8$ | $-23.4^*$ | $-20.3$ / 3 of 4 |
| Llama-2-70B | 56 | $-21.9^*$ | $-15.6^*$ | $-17.2^*$ | $-26.6^*$ | $-20.3$ / 4 of 4 |

*Table 36.* **H3 prefill/decode mismatch beyond 7B.** Accuracy is averaged over CSQA, OBQA, QASC, and ARC-C. Parentheses report change from baseline.

| Setting | Baseline | Decode-est. | Prefill-est. | Random | Angle |
|---|---|---|---|---|---|
| 3B L27 | 69.2 | 46.3 ($-22.9$) | 64.3 ($-5.0$) | 69.2 (0.0) | $72.1°$ |
| 13B L10 | 61.7 | 29.7 ($-32.0$) | 62.1 ($+0.4$) | 61.3 ($-0.4$) | $81.2°$ |
| 70B L56 | 77.0 | 42.2 ($-34.8$) | 75.8 ($-1.2$) | 76.2 ($-0.8$) | $81.9°$ |

*Table 37.* Policy sensitivity under sampling on Llama-2-7B-Chat. Each cell reports shared ratio and mean principal angle relative to the same-layer greedy-estimated basis. All 9 settings pass the H1 sharedness test ($p < 0.05$).

| Layer | $T = 0.7$ | $T = 1.0$ | $T = 1.3$ |
|---|---|---|---|
| 10 | 0.85%, 56.82° | 2.02%, 71.43° | 4.57%, 64.74° |
| 18 | 1.93%, 60.38° | 2.74%, 66.47° | 1.49%, 66.26° |
| 28 | 0.88%, 67.42° | 0.56%, 59.50° | 0.61%, 58.57° |

*Table 38.* Step-localized shared-subspace ablation on GSM8K extended generation trajectories with Llama-2-7B-Chat. We ablate the shared subspace only within one third of the generated decode steps. Accuracy is reported in percent.

| Intervention window | Accuracy (%) |
|---|---|
| No ablation | 25.0 |
| Early third only | 14.1 |
| Middle third only | 6.2 |
| Late third only | 12.5 |
| Full trajectory | 0.0 |

*Table 39.* Readout-remap control with the same fixed layer-10 decode-shared basis. More negative $\Delta$ indicates stronger causal degradation from shared-subspace removal. Matched-energy controls remain near zero under each readout.

| Readout protocol | What changes | Mean $\Delta$ | Retained effect | Controls |
|---|---|---|---|---|
| label | Original answer-letter readout | −11.7 | 100% | near 0 |
| option_label | Remapped option label | −10.9 | 93% | near 0 |
| option_text | Full option-text readout | −7.8 | 67% | near 0 |

