# OpenReview forum: "DecodeShare: Tracing the Shared Subspace of LLM Decode-Time Decisions"
_ICML.cc/2026/Conference — ICML 2026 spotlight_

### Official Review · Reviewer_GcDp · 2026-02-26

**Soundness:** 2
**Presentation:** 3
**Significance:** 3
**Originality:** 3
**Overall Recommendation:** 4
**Confidence:** 2

**Summary:**

The paper introduces DecodeShare, a protocol that (1) identifies a low-dimensional subspace shared across tasks in KV-cached decode-time hidden states via pooled PCA with a sharedness threshold, (2) demonstrates via decode-only causal interventions (ablation + patchback) that this subspace is causally necessary for next-token decisions, (3) shows prefill-estimated subspaces fail to reproduce these causal effects, and (4) proposes projecting steering vectors away from this shared subspace to reduce template brittleness.

**Compliance With Llm Reviewing Policy:**

Affirmed.

**Final Justification:**

The rebuttal fully addressed my concerns — the 3B/70B scalability checks confirm the shared subspace pattern generalizes beyond 7B, and the overhead measurements clarify computational costs. The hypothesis-driven methodology and the prefill-decode mismatch finding remain the paper's key strengths. I maintain my score at 4.

**Key Questions For Authors:**

- How sensitive is the shared subspace to the choice of decoding policy π (greedy vs. sampling with different temperatures)? All experiments appear to use greedy decoding.
- What is the computational overhead of the full DecodeShare pipeline (subspace estimation + evaluation) relative to standard inference?

**Limitations:**

yes

**Strengths And Weaknesses:**

**Strengths**:

- Rigorous experimental methodology with systematic controls (energy-matched, LOTO, patchback with multiple control conditions) that set a high bar for causal intervention studies
- Clear hypothesis-driven structure (H1–H3) with explicit falsification criteria — a model of good scientific practice
- The prefill-decode mismatch finding (H3, Table 3) is a genuinely useful insight: prefill-estimated directions do not transfer to decode-time causal effects, which has implications for the broader steering/interpretability community

**Weaknesses**:

- All experiments limited to 7B-class models; no evidence of scalability to larger (13B, 70B) or smaller models (Section 3)
- Downstream utility (steering robustness, Section 4) is demonstrated on only 3 tasks with modest, task-dependent improvements (Table 2) — the practical impact remains unclear

---

> ### Author Rebuttal · Authors · 2026-03-31
>
> **W1.** We agree that scalability beyond 7B is an important question. Since the original submission only evaluated 7B-class models, we extended the rebuttal experiments in both directions. **Additional 3B/70B spot checks show the same qualitative pattern beyond 7B.**
>
> On **Llama-2-70B-Chat**, both the DecodeShare pattern (H1–H2) and the prefill-decode mismatch (H3) persist, with causal effects becoming even more concentrated and statistically stable in later layers.
>
> **Table R8. 70B Evidence: Geometric existence (H1) and Causal effect (H2).** $\Delta$ **denotes mean accuracy drop across CSQA, OBQA, QASC, and Arc_C. (*) indicates p < 0.05.**
>
> | Layer | \|S\| | Ratio (\|S\|/d) | H1($p_{perm}, p_{scram}$) | Mean $\Delta$ | Sig. Tasks |
> |---|---|---|--- | --- | --- |
> | 32 | 21 | 0.60% | ≤0.015,≤0.048 | -15.2 pts | 3/4 |
> | 48 | 2 | 0.05% | ≤0.015,≤0.048 | -20.3 pts | 3/4 |
> | 56 | 13 | 0.32% | ≤0.015,≤0.048 | **-20.3 pts** | **4/4** |
>
> **Table R9. 70B-H3: Prefill-Decode mismatch (Layer 56).**
>
> | **70B Mean (L56)** | **Baseline** | **Decode-est** | **Prefill-est** | **Random** | Mean Angle |
> | --- | --- | --- | --- | --- | --- |
> | **Accuracy (**$\Delta$**)** | 77.0 | 42.2 (**-34.8**)* | 75.8 (-1.2) | 76.2 (-0.8) | 81.9° |
>
> On **Llama-3.2-3B-Chat**, DecodeShare pattern (H1–H2) and the prefill-decode mismatch (H3) also persist.
>
> **Table R10. 3B Evidence of H1 and H2.**
>
> | Layer | \|S\| | Ratio (\|S\|/d) | H1($p_{perm}, p_{scram}$) | Mean $\Delta$ | Sig. Tasks |
> | --- | --- | --- | --- | --- | --- |
> | 27 | 129 | 7.2% | ≤0.015,≤0.048 | -21.9 pts | 4/4 |
>
> **Table R11. 3B-H3: Prefill-decode mismatch (Layer 27).**
>
> |3B Mean (L27)|Baseline|**Decode-est**|Prefill-est|Random|
> |---|---|---|---|---|
> |**Accuracy ($\Delta$)**|69.2|46.3 (**-22.9***)|64.3 (-4.9)|69.2 (0.0)|
>
> *These additional spot checks suggest that the main DecodeShare pattern is not confined to 7B.*
>
> ---
>
> **W2.** We agree that the gains in the main-paper Table 2 are modest and task-dependent, and we will revise the paper to make this clearer. Beyond the task-level gains in Table 2, our new analysis focuses on the underlying **deployment brittleness**: when selection relies on a prefill-involved proxy, the candidate that looks best in validation may not remain best at decode-time deployment. We therefore evaluate which proxy best predicts held-out decoding utility (**REAL**).
>
> To test this, we evaluate three heterogeneous candidate pools (CAA, SAE, and INSTR), so the result is not driven by a single intervention family. Here, **Selected Utility** denotes the held-out utility of the candidate chosen by each proxy, averaged across pools. As in Table R12, **Decode-only** rank consistently aligns with **REAL** with highest correlation, the only positive mean and worst-case selected utility, and the lowest flip rate and regret@1. Thus, DecodeShare’s practical impact **extends beyond task-specific gains**, indicating a more reliable decode-aligned ranking signal in our tested deployment setting.
>
> **Table R12. Ranking Proxies vs. Held-out Deployment Target (REAL)**
> |**Proxy Protocol**|**Spearman Corr. (w/ REAL)**|**Selected Utility (Mean)**|**Selected Utility (Worst)**|**Flip Rate**|**Regret@1**|
> |---|---|---|---|---|---|
> |**Prefill-only**|0.065|-0.0023|-0.0034|75.0%|0.0156|
> |**Mixed (P+D)**|0.454|-0.0023|-0.0034|75.0%|0.0156|
> |**Decode-only (Ours)**|**0.700**|**+0.0107**|**+0.0099**|**8.3%**|**0.0026**|
>
> ---
>
> **Q1.** We thank the reviewer for this question. DecodeShare is **policy-aligned** rather than policy-invariant: as defined in **Section 2.3, line 140-141**, the shared subspace is estimated from states collected under the specific serving-time policy $\pi$ (e.g., greedy, sampling). The main paper focuses on greedy decoding because it provides the cleanest controlled setting for the causal tests.
>
> As Table R13 shows, **H1 persists under all tested sampling policies ($T \le 1.3$)**: while the basis rotates relative to greedy, the shared structure remains compact and statistically significant. Thus, policy changes only affect the basis orientation, not the underlying existence of the shared workspace; further qualitative evidence of the subspace's behavioral impact under sampling is detailed in Appendix C.3.
>
> **Table R13. Policy sensitivity under sampling (Llama-2-7B-Chat). Entries are (shared ratio, mean angle vs. same-layer greedy); all 9 cases pass H1 (p<0.05).**
> |Layer|$T=0.7$|$T=1.0$|$T=1.3$|
> |---|---|---|---|
> |10|0.85%, 56.82°|2.02%, 71.43°|4.57%, 64.74°|
> |18|1.93%, 60.38°|2.74%, 66.47°|1.49%, 66.26°|
> |28|0.88%, 67.42°|0.56%, 59.50°|0.61%, 58.57°|
>
> ---
>
> **Q2.** We measured DecodeShare overhead on an H200 GPU. On Llama-2-7b-chat (L10), the **offline cost** is **dominated by basis estimation**: the full pipeline took 296.5s across five tasks versus 48.5s for the unmodified model, with estimation accounting for 87.6% of runtime. **The online cost is modest**, with a 21.2% throughput overhead (81.10 vs. 102.95 tok/s) and a negligible 1.16 MiB memory footprint.

---

> > ### Author Rebuttal · Reviewer_GcDp · 2026-04-03
> >
> > Thank you for the detailed rebuttal and additional experiments. The scalability checks and overhead measurements are appreciated. I retain my score.

---

> > > ### Author Response · Authors · 2026-04-03
> > >
> > > We thank the reviewer for the positive feedback. It is encouraging to hear that the scalability and overhead data provided the necessary clarity. We really appreciate your time and support for our work!

---

### Official Review · Reviewer_8cSe · 2026-03-10

**Soundness:** 3
**Presentation:** 3
**Significance:** 2
**Originality:** 3
**Overall Recommendation:** 3
**Confidence:** 4

**Summary:**

This paper proposes DecodeShare, a framework for analyzing knowledge sources and information flow during LLM decoding. By examining token-level interactions, it traces how information is shared and combined across tokens to explain generated outputs. Experiments on several models show its potential for improving generation interpretability.

**Compliance With Llm Reviewing Policy:**

Affirmed.

**Final Justification:**

I retain my score and tend to decline. This decision stems from my remaining concerns, primarily regarding the purported scientific value of H1–H3.

**Key Questions For Authors:**

- What is the formal definition of “knowledge sharing” in DecodeShare?

- How does DecodeShare differ from causal tracing or influence functions?

- Can quantitative metrics be designed to evaluate explanation accuracy or stability?

**Limitations:**

See the weaknesses.

**Strengths And Weaknesses:**

Pros:

- The paper studies where generated content in LLMs comes from and how knowledge spreads during decoding, which is relevant for interpretability, trust, and safety.

- It analyzes knowledge propagation through decoding dynamics, rather than relying on standard attention or feature attribution methods.

- DecodeShare focuses on the generation process, so it can be applied to different autoregressive language models.

- Examples show how the method can trace token-level knowledge sources during generation.

Cons:

- The paper does not clearly define `knowledge sharing` how token-level attribution maps to knowledge sources, or how DecodeShare fundamentally differs from existing attribution methods. The approach feels more heuristic than theoretically grounded.

- The paper does not systematically compare DecodeShare with existing interpretability methods such as attention attribution, gradient attribution, causal tracing, logit lens, or influence functions, making its advantages hard to evaluate.

- Experiments rely on case studies without quantitative metrics like attribution accuracy, causal consistency, or faithfulness, making it difficult to judge whether the method truly captures knowledge flow.

- If the analysis depends on attention or similar signals, results may be affected by noise or token interactions, but this issue is not examined.

- Tracking token-level information flow during decoding may be computationally expensive for long sequences or large models, yet the paper does not analyze complexity or scalability.

---

> ### Author Rebuttal · Authors · 2026-03-31
>
> We thank the reviewer for raising these questions on terminology, positioning, and evaluation.
>
> ---
>
> **W1 / Q1.** We agree this needs clearer wording. To avoid confusion, the paper does **not** use **“knowledge sharing”** as a technical term. Instead, DecodeShare targets a **cross-task shared decode-time subspace** estimated from KV-cached hidden states. Section 2.3 defines a direction as shared when its relative variance contribution $r_{\ell,t,i}$ exceeds $\tau$ for at least $m$ tasks, yielding the shared set $S_\ell(\tau,m)$ and shared basis $Q_\ell^{(S)}$.
>
> This also clarifies that DecodeShare is not primarily designed for knowledge-tracking-style token/source attribution; For more details, please refer discussion of W2/Q2.
>
> We agree the estimator is operational rather than theorem-level unique, and we will make this more explicit in revision. Our claim is modest: not that DecodeShare recovers a unique latent circuit, but that it identifies a compact candidate subspace with **stable, meaningful** causal conclusions. The paper supports this through strong null tests and a stable intermediate $\tau$-regime for **H1**, together with matched-control, decision-level, LOTO, and patchback results for **H2**.
>
> ---
>
> **W2/Q2.** We agree the paper should position DecodeShare more explicitly relative to referenced interpretability methods. **The key difference is the recovered object.** referenced methods—such as causal tracing and influence functions—typically recover instance-specific properties, whereas DecodeShare targets a **pooled cross-task shared decode-time subspace**. Thus, although our intervention locus is local to individual decode steps, it is used to test a global (task-level) structure rather than to recover run-specific saliency/readouts.
>
> **Table R7. DecodeShare vs. referenced methods: instance-specific vs. global scope.**
> |Method|Typical output|Mostly instance-focused?|
> |---|---|---|
> |Attention attribution|Token/head/path saliency for one output|Yes|
> |Gradient attribution|Token/feature saliency for one logit or loss|Yes|
> |Causal tracing|Important layers/head for one prediction|Usually yes|
> |Logit lens|Decoded Single Hidden States from Intermediate Layers|Usually yes|
> |Influence functions|Training-example influence on one test prediction/loss|Yes (test-example-centered)|
> |**DecodeShare (ours)**|**Pooled cross-task shared decode-time subspace**|**No for the recovered object; intervention is local**|
>
> ---
>
> **W3 / Q3.** As clarified in **W1–W2**, the reviewer’s suggested metrics are only partially aligned with DecodeShare’s target object. Quantitative metrics should match the recovered object: for DecodeShare, the key questions are not token-/source-level attribution accuracy, but whether the recovered subspace is **stable and causally necessary/specific** for decode-time decisions. Under these aligned criteria, the paper is **already quantitative rather than purely case-study based: H1** directly evaluates **stability/sharedness**, while **H2–H3** evaluate causal necessity/specificity and prefill-to-decode transfer under matched controls. So yes, quantitative metrics can be designed for DecodeShare, and our H1–H3 framework is exactly such an aligned design.
>
> Beyond stability, in the closest aligned sense of “accuracy,” namely whether the recovered basis isolates a meaningful rather than superficial component, our additional 32D split (Table R3) suggests the dominant effect lies in the 29D residual core rather than the 3D readout slice. We will make this evaluation boundary clearer in revision.
>
> ---
>
> **W4.** We believe this concern reflects a slight mismatch with our protocol. DecodeShare does not use attention weights or token-level interaction signals; it estimates a shared subspace from **decode-time hidden states** and intervenes at the same **decode-only** locus. Accordingly, the aligned concern is not attention noise per se, but whether the recovered activation-level subspace is stable and causally specific under the same serving regime. The paper already addresses this through strong null tests and LOTO/patchback specificity checks (detailed in Section 3.1-3.3). Together, these results argue against the effect being explained purely by noise or by a brittle local artifact. We will clarify this boundary in revision and leave finer module-level decomposition as future work.
>
> ---
>
> **W5. The practical overhead of DecodeShare is modest because it does not require per-token source tracing in generation.** Instead, the pipeline relies on one-time offline basis estimation, so the added online decode-step cost is only a lightweight projection that scales with $\mathcal{O}(d \times |S|)$ for the shared core. As detailed in our measurements for **Reviewer GcDp (Q2)**, this translates to ~20% throughput overhead and negligible memory footprint on an H200 GPU.

---

> > ### Author Rebuttal · Reviewer_8cSe · 2026-04-04
> >
> > Clarifying that "knowledge sharing" isn't a technical term just confirms the original framing was sloppy.  Claiming H1–H3 are the quantitative metrics is circular: you built the benchmark, you passed it.

---

> > > ### Author Response · Authors · 2026-04-07
> > >
> > > We thank the reviewer for the final remarks. We addressed the reviewer’s earlier questions in our prior rebuttal. In the present follow-up comment, the remaining explicit concerns center on the manuscript’s framing and the claim that the evaluation is circular, so we address those two points directly below.
> > >
> > > We respectfully disagree that clarifying the term “knowledge sharing” implies sloppiness in the manuscript’s technical framing. As a factual clarification, the phrase “knowledge sharing” does not appear in the manuscript. Our prior response regarding W1/Q1 addressed terminology only to avoid confusion; it did not alter the paper’s claim. **The relevant question is whether the manuscript itself clearly specifies its formal object and how that object is evaluated. It does:** the paper presents a consistent line from motivation to formalization to evaluation, grounding the problem in the brittleness of existing steering methods and prefill/decode mismatch, formalizing the target through a central hypothesis and three falsifiable checks (H1–H3), and evaluating those checks with quantitative measurements and statistical readouts detailed in Secs. 2.2–2.4 and 3. The manuscript’s framing is therefore defined by this consistent technical line. **Clarifying a colloquial label does not justify the conclusion that the manuscript’s framing is “sloppy.”**
> > >
> > > **Regarding the circularity concern, the key is to clearly separate three parts of the paper:** first, H1–H3 as falsifiable hypotheses rather than quantitative metrics; second, the evaluation of a candidate subspace rather than a restatement of the setup used to construct it; and third, a claim-aligned protocol rather than a self-fulfilling benchmark (*We assume the “benchmark” mentioned in acknowledgement refers to our hypotheses H1-H3*).
> > >
> > > **First, H1–H3 are not quantitative metrics; they are falsifiable hypotheses** about distinct claimed properties of the proposed decode-time shared subspace. The quantitative evaluation is carried out through the task readouts and statistical tests reported in the paper (e.g., FC-Acc, gen-acc/EM, pair-logprob, paired significance tests), as described in Secs. 2.4 and 3. The paper therefore does not define a metric and then satisfy it by construction. Rather, it specifies claims and evaluates them with quantitative evidence.
> > >
> > > **Second, the evaluation logic is not circular.** That characterization would apply only if the later validation stages merely restated what was already assumed in the initial setup. That is not the structure of the paper. The logic is summarized below.
> > >
> > > **Table R14. Logic Flow of DecodeShare’s Four Stages**
> > >
> > > | Stage | In the manuscript | Role |
> > > | --- | --- | --- |
> > > | Setup | Protocol and falsification criteria | Defines what is tested and how failure is identified |
> > > | Construction | Candidate shared subspace | Produces the object under study |
> > > | Checks | H1–H3 with quantitative readouts | Tests the candidate empirically |
> > > | Verdict | Overall empirical conclusion | Drawn from the checks |
> > >
> > > As the Table R14 shows, the paper proceeds in four stages. Setup defines the protocol and falsification criteria. Construction produces the candidate shared subspace. Checks then evaluate that candidate through H1–H3 with quantitative readouts. Finally, the verdict is drawn from those checks. The key point is that the setup determines what is tested and how failure is identified, **but it does not predetermine the verdict.** The empirical conclusion is therefore obtained from the later evaluation rather than fixed by the initial protocol.
> > >
> > > **Third, this also means the protocol is not a self-fulfilling benchmark, but a claim-aligned test.** DecodeShare is explicitly positioned as a protocol and evaluation contribution under KV-cached decoding: it estimates a candidate shared subspace from decode-time hidden states, intervenes on the same decode-time states to avoid estimator–intervention mismatch, and studies that candidate through H1–H3 using matched nulls, matched controls, decision-level readouts, and LOTO. These choices are tied to the object under study and to the confounds the paper is designed to rule out.
> > >
> > > Moreover, **the protocol was not designed to be an automatic pass.** Our experiment was based on hypothesis testing, a commonly used statistical tool, which outputs a statistical conclusion based on empirical evidences observed and does not guarantee to pass by design. As discussed in Table R1 and Reviewer hHrn Q2, there can be some layers with relatively weaker sharedness so some intervention are less stable, despite a consistent trend observed among all cases. This shows that the criteria do not operate as an automatic pass condition; the proposed shared subspace is required to demonstrate its role through separate task-level evaluation.

---

### Official Review · Reviewer_hHrn · 2026-03-12

**Soundness:** 4
**Presentation:** 3
**Significance:** 3
**Originality:** 4
**Overall Recommendation:** 5
**Confidence:** 3

**Summary:**

The paper investigates a very interesting, and to my knowledge, novel question of decode time vs prefil-time circuit distinctions. The authors encounter a decoding subspace that is practical for activation steering, unique from previous steering methods (that presumably target prefil activation subspaces). The resulting method DecodeShare rests on the hypothesized existence of the decision subspace S, and three sub-hypotheses (H1,.., H3), which are evaluated robustly, and scientifically.

**Compliance With Llm Reviewing Policy:**

Affirmed.

**Final Justification:**

The paper investigates a very interesting, and novel question of decode time vs prefil-time circuit distinctions. The authors encounter a decoding subspace that is practical for activation steering, unique from previous steering methods (that presumably target prefil activation subspaces). The resulting method DecodeShare rests on the hypothesized existence of the decision subspace S, and three sub-hypotheses (H1,.., H3), which appear to be evaluated robustly, and scientifically.

I find that the evaluation of these hypotheses is comprehensive, and rather convincing. Without getting lost in the details, it is difficult to argue that the paper does not make interesting discoveries. What remains unclear is the broader significance of these discoveries. I leave this up to the opinion of the other reviewers/ACs. In any case it is my view that the work deserves recognition.

**Key Questions For Authors:**

These questions mainly represent curiosities, rather than limitations. The main qualms with the paper are listed in the strengths/weaknesses section above. The authors are under no obligation to render a conclusive opinion on these questions.

Q1. THe method finds a shared subspace using pooled PCA and a sharedness threshold τ. How sensitive are the results to the exact estimation pipeline? i.e could multiple distinct subspaces (S1, S2, ..) produce similar causal effects to S? i.e Is the discovered S stable?

Q2. To my knowledge the performed interventions are primarily at layer ℓ = 10. How consistent are the discovered shared subspaces across layers, and do earlier or later layers show similar causal behavior?

Q3. The experiments intervene at decode-time states aggregated across steps. Have the authors analyzed whether the shared subspace becomes more important closer to the final answer token, or is it equally important earlier in decoding? (i.e does this inform decision formation vs decision readout in some sense?)

Q4. Most experiments appear to use ~7B-class models. Have the authors evaluated whether the same shared decode-time subspace phenomenon holds for larger models (e.g., 30B or 70B)? Do the authors expect these results (steerability etc.) to vary based on model scale/model aspect ratio etc.

**Limitations:**

Yes

**Strengths And Weaknesses:**

**Soundness**

(Strengths) The paper is very sound. The authors define explicit hypotheses H1 to H3 and falsification criteria, rather than just telling a post hoc story. They use matched null baselines for sharedness, paired bootstrap confidence intervals, paired randomization tests, decision-level forced-choice metrics to avoid formatting confounds, and LOTO estimation to address leakage.

The paper is disciplined about intervention alignment. The core idea is to specifically estimate the subspace S from decode-time KV-cached states and intervene at that same decode-time locus, directly testing the paper’s main claim about decision-time computation and cleanly motivates the H3 prefill-vs-decode comparison. The control design also stands out. The paper uses dimension-matched and energy-matched controls.

(Weaknesses) One weakness is that the subspace construction depends on several choices that are somewhat heuristic: the threshold τ, the all-task or m-task sharing rule, layer choice, and decode-step pooling. The authors do show some robustness to τ and ρ, but the discovered object is arguably a product of this pipeline rather than something uniquely identified. Granted the focus of the paper is highly empirical and MechInterp generally admits this property.

The paper also shows that the subspace is causally important, but it does not render an opinion about what computation that subspace represents. It may be a shared decision workspace, but it could also bundle together several entangled things like confidence calibration, and answer-formatting.

**Significance**

(Strengths) The results are strong. The shared subspace is small relative to width, yet removing it causes much larger accuracy drops than matched controls, and this persists across multiple models, tasks, and both generation and forced-choice style evaluations, which appears to be of great significance. The paper also shows strong prefill-decode mismatch, which will likely be of great interest to the community. Any qualms with the significance of this paper would likely rest on the significance/value of activation steering generally as an alignment method. The authors have done a good job in convincing the reader that this is the right way to do this form of steering.

**Originality**

(Strengths) The paper appears highly original. The paper combines decode-time alignment under KV cache, explicit causal testing of a cross-task shared subspace, and the demonstration that prefill-estimated bases can fail badly at decode time. To my knowledge these components have not been organized as such in previous literature.

**Presentation**

The presentation is very professional, and high quality. The only weakness may be that, on account of the highly comprehensive empirical results, the experiments and figures may border on dense and cluttered. The authors may consider simplifying some tables (i.e table2) and moving some extraneous results to the appendix.

---

> ### Author Rebuttal · Authors · 2026-03-31
>
> We sincerely thank the reviewer for the thoughtful and encouraging assessment. Since these points are closely related, we answer them in grouped form.
>
> ---
>
> **W1 Part 1/ Q1.** We agree that the shared subspace (S) is recovered by an operational pipeline, not as a theorem-level uniquely identified latent object. Our claim is therefore narrower: DecodeShare does not depend on one privileged parameterization of a hidden circuit; rather, it shows that a transparent, decode-aligned estimator can recover a compact candidate subspace whose **causal conclusions** remain stable under reasonable choices.
>
> For $\tau$, the relevant scale is the diffuse baseline $1/k$, not an absolute constant. As detailed in Table R5, local sweeps show a stable compact regime rather than a brittle threshold effect: at $\mathrm{pca\\_var}=0.95$, varying $\tau$ from $5\times10^{-4}$ to $10^{-3}$ to $2\times10^{-3}$ changes the shared ratio from 0.0903 to 0.0365 to 0.0164; at $\mathrm{pca\\_var}=0.97$, the corresponding values are 0.0735, 0.0300, and 0.0132. Furthermore, Table R6 demonstrates that the pooled basis also converges quickly as the task pool grows: mean overlap with the full-task reference basis is already about 0.92 with 2 tasks and reaches 1.00 by 8 tasks for both Llama-2-7B-Chat and Qwen2.5-7B-Instruct.
>
> More broadly, we agree that multiple nearby subspaces could in principle yield similar effects; our claim is **not uniqueness among all possible decompositions**. The relevant empirical question is whether the recovered object is stable enough to support reproducible causal conclusions. On that criterion, the evidence suggests yes.
>
> ---
>
> **W1 Part 2/ Q3.** We thank the reviewer for raising this point, because it goes directly to the scope of our claim. We agree that the current paper identifies the **causal role** of the shared subspace more directly than its uniquely disentangled functional components. Our claim is therefore about empirical robustness rather than mathematical uniqueness: the current evidence supports $S$ as a **decision-relevant decode-time workspace**, not a uniquely isolated single-function module.
>
> At the same time, the evidence goes beyond a pure answer-format account. In the paper, the shared–control gap persists under staged evaluation and under forced-choice decision-level readout, which was introduced specifically to separate decision quality from formatting/termination artifacts. Patchback further shows that restoring only the removed shared component rescues decisions, rather than merely repairing generic generation damage.
>
> We do **not** claim that $S$ is purely “reasoning” or fully disentangled from calibration / *answer-formatting* effects. The safer interpretation is that $S$ contains a dominant **decision-forming computational core** together with a smaller readout/format component. Our additional 32D split in discussion with reviewer 6RVe is consistent with this view: ablating the 3D readout slice is nearly inert, whereas ablating the 29D residual core reproduces most of the causal effect.
>
> To check if this is merely a final-token artifact, we ran a step-localized ablation on extended generation trajectories (GSM8K, L2-7b-chat). Ablating the shared subspace in only the early, middle, or late third of CoT steps drops accuracy from 25.0% to 14.1%, 6.2%, and 12.5%, respectively (full ablation: 0.0%). This confirms a distributed decode-time role unfolding during reasoning, rather than a single final-token switch. We will clarify this in the revision.
>
> ---
>
> **Q2/ Q4.** We agree that across-layer dynamics and scale deserve a clearer treatment. The main paper uses $\ell=10$ as a representative layer for the clearest causal exposition along with layer 4 and 24, as detailed in experiment setup (line 204-205), and the Appendix B jointly shows sharedness across layers and layer-dependent patchback behavior. In rebuttal, we extended the analysis beyond 7B. On Llama-2-13B-chat, H1 holds at all probed layers, and H2-style decode-only ablation yields non-positive task-level deltas at every probed layer, with the strongest effects in later layers. We also ran targeted late-layer checks on 70B and again observed the strongest degradation in late layers (e.g., layer 56, mean drop $\approx 20.3$ pts across four held-out tasks).
>
> Our intended claim is therefore modest: the current evidence suggests the decode-shared phenomenon is **not confined to 7B**, while its causal strength is **layer- and model-dependent rather than uniform across depth or scale**. We do not yet claim a full multi-scale sweep or a fixed scaling law, and the exact best layer / shared ratio / intervention strength may vary across model families.

---

> > ### Author Rebuttal · Reviewer_hHrn · 2026-04-02
> >
> > These new results are encouraging. I want to thank the authors for addressing my questions and (limited) concerns. I maintain that this work appears to be an **excellent**, and **interesting** piece of science worthy of acceptance.

---

> > > ### Author Response · Authors · 2026-04-03
> > >
> > > We sincerely thank the reviewer for the thoughtful evaluation and the very encouraging feedback. It is gratifying to hear that our rebuttal and the additional experiments successfully addressed your concerns. We truly appreciate your strong support and your recognition of our work as a "worthy of acceptance" piece of science.

---

### Official Review · Reviewer_6RVe · 2026-03-15

**Soundness:** 3
**Presentation:** 3
**Significance:** 3
**Originality:** 3
**Overall Recommendation:** 5
**Confidence:** 2

**Summary:**

In this paper, the authors propose DecodeShare, a protocol for identifying and causally testing cross-task shared subspaces in LLM hidden states, specifically during KV-cached decode time. The method collects decode-time hidden states across multiple tasks, applies pooled PCA to find a shared coordinate system, then identifies directions with consistently high relative variance across tasks via thresholding. The authors test three hypotheses: (H1) shared decode-time structure exists beyond chance, (H2) the shared subspace is causally necessary for decode-time decisions, and (H3) prefill-estimated subspaces fail to transfer to decode-time causal effects. A downstream application shows that projecting steering vectors away from the shared subspace can reduce template sensitivity. The authors perform experiments spanning 7B models (Llama-2, Llama-3.1, Qwen2.5, Falcon) across multiple benchmarks.

**Compliance With Llm Reviewing Policy:**

Affirmed.

**Final Justification:**

The rebuttal addressed all my concerns.

**Key Questions For Authors:**

1.  Have the authors probed the shared subspace through the unembedding matrix or linear classifiers to characterize what it encodes?
2. Is there a principled threshold selection method for τ ?

**Limitations:**

yes

**Strengths And Weaknesses:**

> **Strengths:**

1. The authors provide a rigorous causal methodology by pairing shared-subspace ablations with energy-matched controls (α-match, k-match), patchback specificity tests showing 100% targeted vs. 0% non-shared rescue, and LOTO re-estimation to rule out leakage.

2.  The paper demonstrates that prefill-estimated shared bases are near-orthogonal to decode-estimated ones and fail to produce causal effects at decode time (Table 3, Table 16). This finding can be directly relevant to existing activation steering work that estimates directions from prefill.

3. The authors perform a comprehensive evaluation across four model families, multiple layers, benchmarks, and multiple evaluation protocols (generation, forced-choice, staged).

> **Weaknesses:**

1. Currently, all the experiments use 7B models, it would be interesting to see results on larger models such as 13B.

---

> ### Author Rebuttal · Authors · 2026-03-31
>
> We thank the reviewer for the constructive feedback and for recognizing the rigor of our causal design. We address the scale concern below and clarify both what the recovered subspace appears to encode and how we select $\tau$.
>
> ---
>
> **W1.** We agree that evaluation beyond 7B is important. We therefore extended the rebuttal experiments to **Llama-2-13B-Chat and -70B-Chat (Table R8-9)**, testing H1, H2, and H3 under the same protocol. The 13B results match the 7B qualitative picture: **H1 holds at all three tested layers** under the default setting ($\tau=10^{-3}, 8$ shared tasks, n=128), with compact shared subspaces and both null tests passed; **H2 is directionally consistent across all 12 task-layer cell** with clear decode-time degradation after shared-subspace removal. H3 is also consistent with 7B: the decode basis drops accuracy (-32.0), while the near-orthogonal (81.2°) prefill basis remains inert. Together, these results suggest that DecodeShare’s main findings are **not specific to 7B-class models**.
>
> **Table R1. 13B evidence: geometric existence (H1) and causal effect (H2).** H2 reports mean $\Delta(\text{shared} - \text{baseline})$ in accuracy points over the 4-task multiple-choice slice (CSQA / OBQA / QASC / ARC). More negative means stronger causal degradation. Sig. is the number of tasks with p<0.05.
>
> | Layer | \|S\| | Ratio | H1 (p_perm, p_scram) | Mean Δ | Sig. Tasks |
> | --- | --- | --- | --- | --- | --- |
> | 10 | 88 | 1.72% | ≤0.015, ≤0.048 | -9.6 pts | 4/4 |
> | 28 | 132 | 2.58% | ≤0.015, ≤0.048 | -11.9 pts | 3/4 |
> | 39 | 87 | 1.70% | ≤0.015, ≤0.048 | -11.7 pts | 2/4 |
>
> **Table R2. 13B-H3: Prefill-Decode mismatch (Layer 10).**
>
> | **13B Mean (L10)** | **Baseline** | **Decode-est** | **Prefill-est** | **Random** | **Mean Angle** |
> | --- | --- | --- | --- | --- | --- |
> | **Accuracy (**$\Delta$**)** | 61.7 | 29.7 (-32.0) | 62.1 (+0.4) | 61.3 (-0.4) | 81.2° |
>
> ---
>
> **Q1.** Yes. On Llama-2-7B-Chat layer 10, held-out linear probes split the extracted 32D shared workspace into a separable **3D readout/output slice** $Q_{\mathrm{out}}$ (answer readout / option letter / newline) and a complementary **29D core subspace** $Q_{\mathrm{core}}$. As shown in Table R3, ablating the 29D core subspace nearly reproduces the effect of ablating the full shared subspace, whereas ablating the 3D readout/output slice alone produces little to no effect.
>
> **Table R3. Causal split of the shared basis (4 pooled forced-choice tasks; n=256; baseline_acc = 44.1).**
>
> | **Ablation** | **Accuracy** | **Δ vs. baseline** | **p-value** |
> | --- | --- | --- | --- |
> | Full shared (32D) | 28.5 | -15.6 | 0.0010 |
> | $Q_{\mathrm{out}}$ (3D) | 44.9 | +0.8 | 0.748 |
> | $Q_{\mathrm{core}}$ (29D) | 27.3 | -16.8 | 0.0005 |
>
> This pattern suggests that the dominant causal effect is not captured by a narrow readout/formatting slice. Held-out probes (Table R4) show that the 29D core is richer in reasoning-like, procedural, and symbolic markers than the 3D readout/output slice. Taken together, while we do not claim a cleanly isolated decision module, these results are **more consistent with a decision-relevant decode-time core and a smaller separable readout/output component** than with a superficial formatting channel.
>
> **Table R4. Held-out probe average precision (AP) for reasoning-related and symbolic markers.**
>
> |**Probe tag** | $Q_{\mathrm{core}}$ | $Q_{\mathrm{out}}$ |
> |---|---|---|
> |Reasoning markers (*because, therefore, thus*) | 0.564 | 0.041 |
> |Step markers (*step 1, step 2, next*) |0.169|0.029 |
> |Digits | 0.966 | 0.132 |
> |Equation symbols | 0.673 | 0.055 |
>
> ---
> **Q2.** Yes, in the sense of a scale-aware selection rule rather than a single universal constant. Our choice is to interpret $\tau$ relative to the diffuse baseline $1/k$, so the relevant quantity is $\tau k$. Operationally, we choose $\tau$ from a **stable intermediate regime where the shared set stays compact and changes smoothly**, rather than collapsing to near-empty or expanding to near-full. Table R5 shows this smooth behavior around the default setting. Table R6 shows basis stability as the task pool grows: mean overlap rises from about **0.92** at 2 tasks to **1.00** at 8 tasks for both Llama-2-7B-Chat and Qwen2.5-7B-Instruct. So the principle is **scale-aware compactness ($\tau k$) plus empirical stability diagnostics.**
>
> **Table R5. Local $\tau$ Sensitivity (Empirical Stability to Hyperparameters)**
>
> | pca_var | tau | cross_dim | shared_count | shared_ratio |
> | --- | --- | --- | --- | --- |
> | 0.95 | 0.0005 | 2989 | 270 | 0.0903 |
> | 0.95 | 0.0010 | 2989 | 109 | 0.0365 |
> | 0.95 | 0.0020 | 2989 | 49 | 0.0164 |
> | 0.97 | 0.0005 | 3332 | 245 | 0.0735 |
> | 0.97 | 0.0010 | 3332 | 100 | 0.0300 |
> | 0.97 | 0.0020 | 3332 | 44 | 0.0132 |
>
> **Table R6. Convergence With More Tasks (Geometric Convergence and Uniqueness)**
>
> | Model | overlap mean at 2 tasks | overlap mean at 8 tasks |
> | --- | --- | --- |
> | `Llama-2-7b-chat-hf` | 0.921 | 1.000 |
> | `Qwen2.5-7B-Instruct` | 0.914 | 1.000 |

---

> > ### Author Rebuttal · Reviewer_6RVe · 2026-04-04
> >
> > Thanks for the additional experiments. The rebuttal addresses all my concerns, and I have increased my score accordingly.

---

> > > ### Author Response · Authors · 2026-04-04
> > >
> > > We are very grateful to Reviewer 6RVe for the encouraging feedback and for increasing the score. It is gratifying to know that our response and the new results on model scaling successfully resolved your concerns. We sincerely appreciate your support and your recognition of our work's impact. Thank you for your time and thoughtful engagement throughout the review process!

---

### Decision · Program_Chairs · 2026-04-30

**Decision:**

Accept (spotlight)

**Comment:**

In this paper, the authors propose DecodeShare, which identifies the shared low-dimensional subspace among tasks using pooled PCA of the KV-cached decode-time hidden states. They find that the shared decode subspace is causally important for next-token prediction. They also show that pre-estimated subspaces fail to capture such causality. The distinction between activation steering and the previous steering method (typically from the prefill direction) is illustrated by templated sensitivity.

Most reviewers found the research question "very interesting," the hypothesis evaluations "convincing," and the findings "genuinely useful."
However, some have expressed concern that the formulated hypotheses may not be valuable.  This would affect the significance of the work. Nevertheless, since the conference widely accepts foundational work and since the research asks valuable questions to the community, I recommend acceptance.